

# Overview and evaluation of the Community Multiscale Air Quality (CMAQ) model version 5.1

K. Wyat Appel[1], Sergey L. Napelenok[1], Kristen M. Foley[1], Havala O. T. Pye[1], Christian Hogrefe[1], Deborah J. Luecken[1], Jesse O. Bash[1], Shawn J. Roselle[1], Jonathan E. Pleim[1], Hosein Foroutan[1], William T. Hutzell[1], George A. Pouliot[1], Golam Sarwar[1], Kathleen M. Fahey[1], Brett Gantt[3], Robert C. Gilliam[1], Daiwen Kang[1], Rohit Mathur[1], Donna B. Schwede[1], Tanya L. Spero[2], David C. Wong[1], Jeffrey O. Young[1]

[1]Computational Exposure Division, National Exposure Research Laboratory, Office of Research and Development, U.S. Environmental Protection Agency, RTP, NC
[2]Systems Exposure Division, National Exposure Research Laboratory, Office of Research and Development, U.S. Environmental Protection Agency, RTP, NC
[3]Air Quality Analysis Division, Office of Air Quality Planning and Standards, Office of Air and Radiation, U.S. Environmental Protection Agency, RTP, NC

*Correspondence to:* K.Wyat Appel (appel.wyat@epa.gov)

**Abstract.** The Community Multiscale Air Quality (CMAQ) model is a comprehensive multi-pollutant air quality modeling system developed and maintained by the U.S. Environmental Protection Agency's (EPA) Office of Research and Development (ORD). Recently, version 5.1 of the CMAQ model (v5.1) was released to the public which incorporates a large number of science updates and extended capabilities over the previous release version of the model (v5.0.2). These updates include improvements in the meteorological calculations in both CMAQ and the Weather Research and Forecast (WRF) model used to provide meteorological fields to CMAQ; updates to the gas and aerosol chemistry; revisions to the calculations of clouds and photolysis; and improvements to the dry and wet deposition in the model. Sensitivity simulations isolating several of the major updates to the modeling system show that changes to the meteorological calculations generally result in greater afternoon and early evening mixing in the model, times when the model historically underestimates mixing. The result is higher ozone ($O_3$) mixing ratios on average due to reduced NO titration and lower fine particulate matter ($PM_{2.5}$) concentrations due to greater dilution of primary pollutants (e.g. elemental and organic carbon). Updates to the clouds and photolysis calculations greatly improve consistency between the WRF and CMAQ models and result in generally higher $O_3$ mixing ratios, primarily due to reduced cloudiness and reduced attenuation of photolysis in the model. Updates to the aerosol chemistry results in higher secondary organic aerosol (SOA) concentrations in the summer, thereby reducing $PM_{2.5}$ bias, while updates to the gas chemistry result in generally increased $O_3$ in January and July (small) and slightly higher $PM_{2.5}$ concentrations on average in both January and July. Overall, seasonal variation in simulated $PM_{2.5}$ generally improves in the new model version, as concentrations decrease in the winter (when $PM_{2.5}$ is overestimated by CMAQ v5.0.2) and increase in the summer (when $PM_{2.5}$ is underestimated by CMAQ v5.0.2). Ozone mixing ratios are higher on average with v5.1 versus v5.0.2, resulting in higher $O_3$ mean bias, as $O_3$ tends to be overestimated by CMAQ throughout most of the year (especially at locations where the observed $O_3$ is low), however both the error and correlation are largely improved with v5.1. Sensitivity simulations for several hypothetical emission reduction scenarios showed that v5.1 tends to be slightly more responsive to reductions in $NO_X$ ($NO + NO_2$), VOC and $SO_X$ ($SO_2 + SO_4$) emissions than v5.0.2, representing an improvement as previous studies have shown CMAQ to underestimate the observed reduction in $O_3$ due to large, widespread reductions in observed emissions. Finally, the computational efficiency of the model was significantly improved in v5.1, which keeps runtimes similar to v5.0.2 despite the added complexity to the model.

**Keywords**





CMAQ, WRF, Air Quality Modeling, Model Evaluation, Ozone, Particulate Matter, $PM_{2.5}$

## 1 Introduction

Numerous Federal (e.g. United States Environmental Protection Agency (USEPA)), State and private entities rely on numerical model simulations of atmospheric chemistry, transport and deposition of airborne emissions and the resulting pollutants as part of their decision-making process for air quality management and mitigation (e.g. Scheffe et al., 2007). Chemical Transport Models (CTMs), such as the Community Multiscale Air Quality (CMAQ) model (Byun and Schere, 2006), are often employed to provide information about the potential effects of emission control strategies (e.g. Fann et al., 2009), climate change (e.g. Nolte et al., 2008), and provide next-day air quality forecasts (e.g. Eder et al., 2006) in order to inform and protect the public from potentially harmful air pollutants. Since these models are often used to inform the standard setting and implementation for criteria pollutants (e.g. ozone ($O_3$) and fine particulate matter ($PM_{2.5}$)), they must be maintained at the state-of-the-science. New versions of the CMAQ model have been released periodically over the past fifteen years, with each new version consisting of numerous updates to the scientific algorithms within the model, while also improving the quality of the input data used. Collectively, these updates are aimed at improving the underlying science of atmospheric dynamics and chemistry represented in the model, extending the capabilities for emerging applications, and reducing systematic biases in the modeling system. Every new release of the CMAQ model undergoes extensive evaluation in order to establish its credibility (e.g. Mebust et al., 2003; Appel et al., 2007, 2008, 2013; Foley et al., 2010) and documents its performance relative to previous versions. Most recently, the CMAQ modeling system version 5.1 (v5.1) has been tested and evaluated against observations and was publically released in December 2015 (http://www.cmaq-model.org/).

The scientific upgrades in v5.1 include changes to the photochemistry, new explicit treatment of secondary organic aerosol (SOA) formation from isoprene, alkenes and PAHs; updates to the photochemical cross sections and quantum yields for the Carbon Bond 2005 e51 (CB05e51) chemical mechanism; updates to inorganic and organic chemical reaction rates and products to ensure consistency with the International Union of Pure and Applied Chemistry (IUPAC); and additional representation of organic nitrate species in CB05e51. Significant changes were also made to the in-line calculation of photolysis rates. Additionally, two aerosol mechanisms are now available in v5.1, AERO6 and AERO6i (with isoprene extensions), which include updates to the SOA and ISORROPIA algorithms (Nenes et al., 1998; Nenes et al., 1999), while AERO5 has been deprecated and is no longer available. There were also major revisions to the Pleim-Xiu land-surface model (PX-LSM; Pleim and Xiu, 1995) and Asymmetric Convective Mixing version 2 (ACM2; Pleim, 2007ab) planetary boundary layer (PBL) model in the Weather Research and Forecast (WRF) model version 3.7 (Skamarock et al., 2008), which required revisions to the ACM2 scheme in CMAQ to maintain consistency. In addition, corrections were also made to the Monin-Obukhov length (MOL) calculation in v5.1 to make it consistent with the calculation in the WRF model. The changes to the PX-LSM, ACM2 and MOL calculations in CMAQ had significant impact on the mixing within both WRF and CMAQ, and hence large impacts on the pollutant concentrations in CMAQ.

Section 2 provides a brief description of the major scientific and structural improvements included in v5.1. The model configuration and observational data sets used in the model evaluation are provided in Section 3. The evaluation itself is then presented in two parts. Section 4 documents the evaluation of several specific changes that were isolated as part of the overall testing of the model: (a) changes to the PX-LSM and ACM2 schemes (b) updates to the MOL calculation (c) changes to the inline photolysis calculation and the representation of clouds within CMAQv5.1 and (d) updates for the CB05e51 chemical mechanism. These increments were



chosen as the focus of this paper because they represent a fundamental change from the previously released model version and had the propensity to impact model performance for criteria pollutants. The second portion of the evaluation, presented in Section 5, summarizes the overall change in $PM_{2.5}$ and $O_3$ model performance with v5.1 compared to the previously released version (CMAQ version 5.0.2 (v5.0.2)). Section 6 provides a discussion of the model response of $O_3$ and $PM_{2.5}$ to hypothetical reductions in

emissions. Section 7 discusses upcoming changes and future work for CMAQv5.2 (v5.2), followed by a summary discussion in Section 8.

## 2 Review of scientific improvements in CMAQ v5.1

Improvements to the v5.1 modeling system are the result of many years of scientific advancements derived from laboratory, field and numerical experiments and the efforts of a relatively small group of model developers that both investigate avenues for model

improvements and then update the model (i.e. write code). Given the large community of CMAQ model users, there are never sufficient resources to diagnose and address every issue in the modeling system that has been reported. As such, it is necessary to prioritize updates to the model based on many different factors, including results from evaluations of past model versions, existing and upcoming regulatory needs, emerging scientific issues, requests from the CMAQ user community, and the expertise within the model developer group to meet those needs/requests. The updates described herein represent the "major" updates made to the

CMAQ modeling system from the previous model version, and therefore does not constitute a fully comprehensive description of all the changes made to the system.

### 2.1 WRF and CMAQ meteorological and transport updates

The WRF and CMAQ models were updated to improve the representation of land-surface processes and vertical mixing. There were two changes made to the PX-LSM in WRF. First, the stomatal conductance function for photosynthetically active radiation

(PAR) was revised with a new function that yields a significantly lower magnitude when short-wave radiation is less than 350 $Wm^{-2}$. This in turn results in reduced latent heat flux and enhanced sensible heat flux, causing a delay in surface stabilization (prolongs mixing) during evening transitions hours (i.e. sunset). This reduces overestimations (reduced positive bias) in water vapor mixing ratios, which are common during the evening transition. Similarly, overestimation of concentrations of surface emitted species (e.g. NO, $NO_2$, CO and EC) are also reduced during the evening transition. This change was released in WRFv3.7 and further

revised in WRFv3.8. The second change made to the PX-LSM is a reduction in the heat capacity value for vegetation based on values suggested by Jayakshmy and Philip (2010), which reduces overestimations of minimum 2-meter temperature (i.e. warmer surface temperatures) during the early morning (dawn) hours while also reducing underestimations of 2-meter temperature during the post-dawn hours.

There were also two major revisions made to the ACM2 vertical mixing scheme in both WRF and CMAQ (Pleim et al., 2016). In WRF, the ACM2 was updated to estimate and apply different eddy diffusivities for momentum (Km) and heat (Kh) so that the Prandtl number (Pr) is no longer assumed to be unity (Pr = Km/Kh $\neq$ 1). The second major modification to ACM2 is the implementation of new stability functions for both heat and momentum for stable conditions, which allows for more mixing in the stable regimes, particularly moderately stable conditions that often occur in the early evening hours. CMAQ v5.1 has also been

modified to include the same stability functions that are used in WRF v3.7, and therefore, for consistency, WRF v3.7 (or newer) and CMAQ v5.1 should be used together.



The Monin-Obukhov length (MOL) values used in the ACM2 model in CMAQ were found to differ from the MOL values used in the ACM2 model in WRF. Specifically, the output from WRF was for a preliminary estimate of MOL that was computed in the surface layer model in WRF (module_sf_pxsfclay.F). The MOL was later re-computed in ACM2 in WRF but not loaded into the output array. This inconsistency has been fixed in v5.1 by recomputing the MOL in CMAQ exactly as it is computed in ACM2 in

WRF. However, starting with WRF v3.8, this re-computed MOL value will be available in the WRF output, and therefore it will be unnecessary to re-compute the MOL value in CMAQ. In addition, the estimated MOL value in the surface layer model in WRF will be improved such that there is little difference between the initial MOL estimate and the final re-calculated value.

Finally, previous evaluations of the ground-level coarse particle ($PM_{10}$) concentrations in CMAQ have shown that the model

significantly underestimated the total $PM_{10}$ concentrations (Appel et al., 2012). Contributing to this underestimation is the fact that CMAQ did not have a mechanism in place to allow coarse particles to settle from upper layers to lower layers (although coarse particles in layer one can settle to the surface). As a result, large particles that would normally settle to the lower layers could remain trapped in the layers in the model in which they are emitted or formed. To account for this deficiency in the model, the effects of gravitational settling of coarse aerosols from upper to lower layers has been added to v5.1 to more realistically simulate

the aerosol mass distribution. The net effect of this update is an increase in ground-level $PM_{10}$ concentrations in v5.1 compared to v5.0.2, particularly near coastal areas impacted by sea-spray (Nolte et al., 2015).

### 2.2 Scientific improvements in the CMAQ v5.1 aerosol treatment

CMAQ has historically underestimated SOA in both urban (Woody et al., 2016) and rural (Pye et al., 2015) locations. Thus, improvements to the representation of aerosol from anthropogenic and biogenic hydrocarbons were needed. The updates to SOA

formed from anthropogenic volatile organic compounds (VOC) focus on VOC compounds in existing emission inventories, such as the EPA National Emissions Inventory (NEI), that are likely to fall in the intermediate VOC (IVOC) range. These include long-chain alkanes such as heptadecane and polycyclic aromatic hydrocarbons (PAHs) such as naphthalene. Since these compounds are much less volatile than traditional VOCs, they readily form aerosol in high yields. Long-chain alkanes and PAHs were included in other VOC categories in CMAQ versions prior to v5.1, but were lumped with smaller, more-volatile

compounds that did not form SOA with the same efficiency. By separating long-chain alkanes and naphthalene at the emission processing step, CMAQ can better account for their higher yields. Work by Pye and Pouliot (2012) as well as Jathar et al. (2014) indicate that a large fraction of VOC emissions, particularly IVOC-type compounds, may not be characterized in emission inventories which limits how much SOA can be formed from anthropogenic VOCs in current chemical transport models.

Several new SOA species were introduced in v5.1 AERO6, specifically AALK1 and AALK2 (from long-chain alkanes) and APAH1, APAH2, and APAH3 (from naphthalene). CMAQ v5.1 predicted alkane SOA is responsible for ~20 to 50% of SOA from anthropogenic VOCs, with the largest absolute concentrations during summer in urban areas. Naphthalene oxidation is predicted to produce more modest amounts of SOA (Pye and Pouliot, 2012). Note that PAH SOA in v5.1 only considers naphthalene as the parent hydrocarbon, which about half of the PAHs is considered as SOA precursors in Pye and Pouliot (2012). This approach was

used since naphthalene is a high priority hazardous air pollutant (HAP) and necessary in the model for purposes other than SOA.

Later generation isoprene oxidation products formed under low-$NO_X$ conditions, specifically isoprene epoxydiols (IEPOX), are recognized as a significant source of SOA based on laboratory (Surratt et al. 2010), field (Hu et al. 2015), and modeling (McNeill et al. 2012, Pye et al. 2013, Marais et al. 2016) studies. This SOA is linked to sulfate and acidity and thus represents an





anthropogenically controlled source of biogenic SOA. CMAQv5.1 includes updates to present the IEPOX SOA resulting from aqueous reactions for most chemical mechanisms including CB05 and SAPRC07 as described in Pye et al. (2013).

In addition to the SOA updates for anthropogenic VOCs, AISO3 (acid catalyzed isoprene epoxide aerosol) was revised to represent
SOA from IEPOX. For the CB05tucl, CB05e51 and SAPRC07 chemical mechanisms with IEPOX formation in the gas-phase, heterogeneous uptake of IEPOX on acidic aerosol results in SOA (Pye et al. 2013). This IEPOX SOA replaces the $AISO_3$ treatment based on Carlton et al. (2010). The AISO3J species name is now retained for IEPOX SOA and represents the sum of IEPOX-derived organosulfates and 2-methyltetrols. Explicit isoprene SOA species including 2-methyltetrols, 2-methylglyceric acid, organosulfates, and oligomers/dimers are available in SAPRC07tic with AERO6i. See Table 1 for more information regarding
these new SOA species.

**2.3 Improvements to the CMAQv5.1 in-line photolysis and cloud model**

The in-line calculation of photolysis rates has undergone significant changes in three areas. First, the description of clouds has changed. In v5.0.2, a vertical column had a single cloud deck with a constant cloud fraction and water droplet mixing ratio. In v5.1, a vertical column can have multiple cloud decks with variable cloud fractions and multiple types of water condensates. The
new description is more consistent with the WRF meteorological model output typically used for CMAQ simulations. Second, the mixing model used to compute the refractive indices of aerosol modes, an internal-volume weighted average model, allows the refractive index of each aerosol component to depend on wavelength. Most importantly, the refractive index for elemental (black) carbon reflects the current scientific consensus (Bond and Bergstrom, 2006; Chang and Charalampopoulos, 1990; Segelstein, 1981; Hess et al., 1998) and increases its absorptive capacity. Additionally, estimating aerosol optical properties includes new options to
solve Mie scattering theory or to use the Core-Shell model with an elemental carbon core (Bohren and Huffman, 2004). Run-time options determine whether to solve Mie scattering or to use the Core-Shell model for the internal mixed aerosol modes (http://www.airqualitymodeling.org/cmaqwiki/index.php?title=CMAQv5.1_In-line_Calculation_of_Photolysis_Rates).          By default, the model uses approximate solutions to Mie scattering and the internal-volume weighted average model (Binkowski et al., 2007). Third, several new variables (e.g. resolved cloud fraction, sub-grid cloud fraction, resolved cloud water content) have
been added to the cloud diagnostic file that describe the optical properties of aerosol and clouds and their radiative effects.

Cloud albedo from NASA's Geostationary Operational Environmental Satellite Imager product
(GOES; http://satdas.nsstc.nasa.gov/) was used to evaluate the cloud parameterizations in WRF3.7 and in the photolysis calculations within CMAQ.  The GOES product has a 4km horizontal resolution and was re-gridded to the 12-km grid structure
used in the WRF and CMAQ simulations using the Spatial Allocator utility (https://www.cmascenter.org/sa-tools/).  The satellite data are available at 15 minutes prior to the top of the hour during daytime hours (11:45UTC – 23:45UTC) and were matched to model output at the top of the hour.  Figure 1 shows the average cloud albedo during daytime hours in July 2011 derived from (a) the GOES satellite product (b) WRF3.7 (c) CMAQv5.0.2 photolysis calculations (d) CMAQv5.1 photolysis calculations. Comparison of Figure 1(a) to (c) shows that the cloudiness parameterization in the photolysis module in v5.0.2, which was based
solely on relative humidity, produced far too many clouds relative to the satellite observations.  The new parameterization within v5.1 uses the resolved cloud fractions and water content from WRF and sub-grid cloud fractions and water content determined by the convective cloud model within CMAQ (acm_ae6; Pleim et al., 2005).  As a result, the model predicted clouds in v5.1 are now considerably more consistent with the WRF parameterization (compare Figure 1 (b) to (d)).

**2.4 Improvements in CMAQ v5.1 atmospheric chemistry**





Several changes were made to the CB05TUCL chemical mechanism in v5.1 (Whitten et al., 2010; Sarwar et al., 2012), which is now referred to as CB05e51. These changes include updates to reactions of oxidized nitrogen ($NO_y$) species; incorporation of new research on the atmospheric reactivity of isoprene photo-oxidation products; addition of several high priority HAPs to the standard CB05e51 mechanism (following the protocol in the multipollutant version of CMAQ); and other changes to update the mechanism

and make it compatible with updates to the aerosol chemistry. The objective was to limit modifications to those reactions that are most important, so that the core CB05 mechanism was not fundamentally changed. A more detailed explanation of the changes made in the CB05e51 mechanism is provided below.

### 2.4.1 $NO_y$ updates and additions

The most extensive changes consisted of updates and extensions of the $NO_y$ species, including peroxyacylnitrates, alkyl nitrates,

and $NO_x$ reactions with $HO_x$. The thermal formation and degradation of peroxyacetyl nitrate (PAN) were modified to correct the parameters that describe the rate constant pressure dependence in the fall-off region between the high-pressure limit and the low-pressure limit (i.e. using N=1.41 and Fc=0.3 instead of the CB05tucl defaults of N=1.0 and Fc=0.6) (Bridier et al., 1991). An additional species, MAPAN, was added to explicitly represent PANs from methacrolein because these are a possible contributor to SOA formation. The OH+$NO_2$ reaction rate was updated (Troe, 2012) and a small yield of $HNO_3$ (<1% at STP, varying with

temperature and pressure) was added to the reaction of $HO_2$+NO (Butkovskaya et al., 2007). The single alkyl nitrate species (NTR) in CB05 was replaced with seven species to better investigate the variety of chemical and physical fates of alkyl nitrates. The first-generation monofunctional alkylnitrates and difunctional hydroxy nitrates were assigned Henry's law constants of 6.5e-1 M and 6.5e3 M respectively, while second generation carbonyl nitrates were assigned 1.0e3 M and multifunctional hydroxynitrates were assigned a value of 1.7e4 M. Five species are predominantly from anthropogenic sources, with the relative distribution of mono-

functional (alkyl nitrates) and multi-functional (hydroxy, carbonyl, hydroxycarbonyl, and hydroperoxy) nitrate products determined based on the nitrates produced from the five alkanes and alkenes with the largest emissions as listed in the NEI (Simon et al., 2010). The other two nitrate species represent first generation and later generation nitrates from biogenic (isoprene and terpene) sources. Biogenic nitrate products were based on reaction products from Lee et al. (2014), with $NO_x$ recycling from secondary biogenic nitrate products (Jenkin et al., 2015) and photolysis rates with quantum yields of unity. Finally, a heterogeneous

hydrolysis rate of alkyl nitrates was added (Hildebrandt-Ruiz et al., 2013), with a six-hour lifetime on aerosol at high relative humidity (Liu et al., 2012; Rollins et al., 2013). Additional details can be found in the CMAQv5.1 release documentation (http://www.airqualitymodeling.org/cmaqwiki/index.php?title=CMAQ_version_5.1_(November_2015_release)_Technical_Documentation).

### 2.4.2 Other changes

The high $HO_x$ pathways for isoprene oxidation have been modified to explicitly account for production of isoprene epoxydiol (IEPOX), which can form SOA and modify the gas phase concentrations. The high $NO_x$ pathways have been modified to explicitly produce methacrolein PAN (MAPAN, described in Section 2.4.1) because it reacts faster with OH than other PAN species. Several high priority HAPs were added to the standard version of CB05e51 as either active species or reactive tracers, specifically acrolein, 1,3-butadiene (which produces acrolein), toluene, xylene isomers, α- and β-pinene, and naphthalene using reaction pathways and

rates as defined by IUPAC. Refer to the CMAQv5.1 release documentation for additional details on these updates.

Several other, smaller changes were made to the chemistry to either improve consistency with IUPAC, enhance the integration with heterogeneous chemistry, or for numerical consistency. These include the updates to the products of ethanol reaction with



OH using recommended yields from IUPAC (http://iupac.pole-ether.fr; accessed May 11, 2016); updates to the reactions of acylperoxy radicals with $HO_2$ to include a 44% yield of OH; the addition of a new species, SOAALK, to account for SOA formation from alkanes; and the addition gas-phase and heterogeneous nitryl chloride formation ($ClNO_2$) and $ClNO_2$ photolysis as described by Sarwar et al. (2012).

**2.5 Updates to air-surface exchange processes in CMAQ v5.1**

Meteorologically dependent emissions and deposition, hereafter referred to as air-surface exchange, were extensively updated in v5.1. A data module was developed to share meteorological and calculated atmospheric transport environmental variables between vertical diffusion, deposition, and meteorological dependent emissions to more consistently represent processes common to both deposition and emissions. Additionally, sea salt and biogenic emissions and dry deposition routines were updated.

**2.5.1 Sea salt aerosol emission**

The sea salt aerosol emissions module was updated to better reflect emissions estimates from recent field observations and to incorporate ocean thermodynamic impacts on emissions. The size distribution of sea salt aerosol was expanded to better reflect recent fine-scale aerosol measurements in laboratory and field studies (de Leeuw et al., 2011) by modifying the Ө parameter Gong (2003) from 30 to 8. A sea-surface temperature (SST) dependency to the sea-salt aerosol emissions following Jaeglé et al. (2011)

and Ovadnevaite et al. (2014) was also added, which increased accumulation and coarse mode sea-salt emissions in regions with high SSTs and reduced the emissions in regions with low SSTs. Finally, the surf-zone emissions of sea-salt aerosol were reduced by 50% assuming a decrease in the surf-zone width from 50 m to 25 m to address a systematic overestimation of near-shore coarse sea-salt aerosol concentrations (Gantt et al., 2015).

**2.5.2 Biogenic emissions (BEIS)**

There were also several updates to the calculation of non-methane biogenic voltile organic carbon (BVOC) emissions in v5.1. The Biogenic Emissions Inventory System (BEIS; https://www.epa.gov/air-emissions-modeling/biogenic-emission-inventory-system-beis) model was updated to include the implementation of a dynamic two-layer, sun and shaded, vegetation canopy model, while the PAR response function was integrated into the canopy model following Niinements et al. (2010) for each canopy layer. A leaf temperature algorithm was implemented that replaced the 2-meter temperature to be more consistent with emission factor

measurements. Finally the Biogenic Emission Land-use Data (BELD v4.0) and emission factors for herbaceous wetlands were updated to address overestimates of biogenic VOCs at coastal sites and updated BELD land-use and vegetation species with high-resolution satellite data and *in-situ* survey observations from 2002-2012 (Bash et al., 2016).

**2.5.3 Dry deposition**

Finally, there were two important updates to the dry deposition calculation in v5.1. First, the dry deposition of $O_3$ over oceans was

updated to include the additional sink due to interaction with iodide in the seawater (marine halogen chemistry), with the iodide concentrations estimated based on sea-surface temperature (Sarwar et al., 2015), which increased the $O_3$ deposition velocity over oceans. Second, over vegetative surfaces, the wet cuticular resistance was updated following Altimir et al. (2006), 385 s $m^{-1}$, and dry cuticular resistance was set to the value of Wesley (1989) for lush vegetation, 2000 s $m^{-1}$. These changes resulted in an approximately 2.0 ppbv reduction in the modeled $O_3$ mixing ratios, with the largest reductions, ~10%, occurring during the

nighttime and early morning hours, and approximately a 2% reduction in the modeled midday $O_3$ mixing ratio.



### 3 Modeling setup and observational data sets

The modeling setup for the evaluation of v5.1 utilizes a domain covering the entire contiguous United States (CONUS) and surrounding portions of northern Mexico and southern Canada, and the eastern Pacific and western Atlantic oceans. The modeling domain consists of 299 north-south by 459 east-west grid cells with 12-km by 12-km horizontal grid spacing and 35 vertical layers

with varying thickness extending from the surface to 50 hPa and an approximately 10 meter mid-point for the lowest (surface) model layer. The simulation time period covers the year 2011, which is a base year for the National Emission Inventory (NEI) and also a period during which specialized measurements from a variety of trace species are available from the Deriving Information on Surface Conditions from Column and Vertically Resolved Observations Relevant to Air Quality (DISCOVER-AQ; http://www.nasa.gov/mission_pages/discover-aq/index.html) campaign.

Several sets of CMAQ simulations were performed to help thoroughly evaluate both the overall change in model performance between v5.0.2 and v5.1 and to examine the individual impact of specific model process changes on the model performance metrics. As such, different input data sets were used/required for the v5.0.2 and v5.1 simulations. The base v5.0.2 simulation (CMAQv5.0.2_Base) utilized WRF v3.4 meteorological input data, while WRF v3.7 derived meteorological data were used for all

the v5.1 simulations presented here. Model ready meteorological input files were created using version 4.1.3 of the Meteorology-Chemistry Interface Processor (MCIP; Otte and Pleim, 2010) for the WRF v3.4 data and MCIP v4.2 (https://www.cmascenter.org/help/documentation.cfm?model=mcip&version=4.2) for the WRF v3.7 data.

Emission input data for the v5.0.2 simulation were based on version 1 (v1) of the 2011 modeling platform developed by the USEPA

from regulatory applications (https://www.epa.gov/sites/production/files/2015-08/documents/lite_finalversion_ver10.pdf), while the base v5.1 simulation utilized emission data based on version 2 (v2) of the 2011 modeling platform. The most significant changes in the emissions between v1 and v2 (based on https://www.epa.gov/sites/production/files/2015-10/documents/nei2011v2_tsd_14aug2015.pdf) are highlighted below.

For the oil and gas sector, there were 4 major changes: 1) better aligning the inputs and emission factors between the EPA's Office of Atmospheric Program (OAP) work on the Greenhouse Gas (GHG) Emissions Inventory (EI) / GHG Reporting Program and the NEI on condensate tanks, liquids unloading, pneumatic devices and well completions, 2) additional information from the Western Regional Air Partnership (WRAP) based on new survey data and studies, 3) improved resolution of data (to county level rather than basin), and 4) new SCCs, including the distinction between Coal Bed Methane (CBM) wells from other natural gas (NG)

wells. For other nonpoint sources, many states resubmitted data based on EPA or their own review of v1 (i.e. CA, CT, DC, DE, IA, ME, MI, MN, NC, NE, NY, OK, UT, VA, WA). Some tribes also submitted their data for the first time for the 2011 v2. For mobile sources, MOVES2014 was used in v2, while MOVES2010b was used for v1 (https://www3.epa.gov/otaq/models/moves/moves-docum.htm). Some commercial marine inventories were also updated.

With respect to fires in the NEI, wild-land and prescribed fire emissions were altered for NC and DE. NC submitted their own emissions in going from v1 to v2, resulting in an over 95% reduction in NC wildfire emissions. Nationally, this caused emissions to be about 30% lower in 2011 v2 vs 2011 v1. The Delaware fire emissions were reduced about 96%, however the effects nationwide were small. For agricultural fires, updates from v1 resulted in a reduction of between 95-99% of emissions for WI, MI, OH, MO, and IL. Cumulatively, these changes reduced emissions about 34% nationwide.





The raw emissions files were processed using versions 3.5 (v1 emissions) and 3.6.5 (v2 emissions) of the Sparse Matrix Operator Kernel Emissions (SMOKE; https://www.cmascenter.org/smoke/) to create gridded, speciated hourly model-ready input emission fields for input to CMAQ. Electric generating unit (EGU) emissions were obtained using data from EGUs equipped with Continuous Emission Monitoring System (CEMS). Plume rise for point and fire sources were calculated in-line for all simulations.

Biogenic emissions were generated in-line in CMAQ using BEIS versions 3.14 for v5.0.2 and 3.61 (Bash et al., 2016) for v5.1. All the simulations employed the bi-directional ammonia flux (bi-di) option for estimating the air-surface exchange of ammonia, as well as the in-line estimation of $NO_X$ emissions from lightning strikes.

Output from the various CMAQ simulations is paired in space and time with observed data using the Atmospheric Model
Evaluation Tool (AMET; Appel et al., 2011). There are several regional and national networks that provide routine observations of gas and particle species in the U.S. The national networks include the EPA's Air Quality System (AQS; 2086 sites; https://www.epa.gov/aqs) for hourly and daily gas and aerosol PM species; the Interagency Monitoring of PROtected Visual Environments (IMPROVE; 157 sites; http://vista.cira.colostate.edu/improve/) and Chemical Speciation Network (CSN; 171 sites; https://www3.epa.gov/ttnamti1/speciepg.html) for daily average (measurements typically made every third or sixth day) total
and speciated aerosol PM species; and the Clean Air Status and Trends NETwork (CASTNET; 82 sites; http://www.epa.gov/castnet/) for hourly $O_3$ and weekly aerosol PM species. In addition to these routinely available observations, the DISCOVER-AQ campaign (https://www.nasa.gov/mission_pages/discover-aq/) during July 2011 provides additional ground-based gas and aerosol PM measurements, along with unique aloft measurements made by aircraft, vertical profilers (e.g. Light Detection And Ranging (LiDAR) measurements), ozonesondes and tethered balloons.

**4 Evaluation of major scientific improvements**

In this section we evaluate the impact that several of the major scientific improvements in v5.1 have on the operational model performance. Unlike Foley et al. (2010), in which several individual major scientific improvements in CMAQ v4.7 were evaluated incrementally (e.g. each subsequent improvement is evaluated against the previous improvement), here we examine each scientific improvement by comparing simulations with the specific improvement removed (i.e. as it was in v5.0.2) to the base v5.1 simulation
(CMAQv5.1_Base) which includes all the updates. While this has the disadvantage of not showing the incremental change in model performance due to each improvement, it does limit the number of simulations that need to be performed. In addition, it allows for easier examination of the effect of nonlinear increments on total model performance, as some updates to the modeling system may be affected by updates to other parts of the model, the effects of which on model performance may not be captured in an incremental testing format. Note that while some attempt is made to broadly identify the processes involved that cause the
observed changes in model performance between v5.0.2 and v5.1, it would be too laborious (both to the reader and to the investigators) to comprehensively describe and investigate in-depth the processes involved that result in each observed difference in model performance described in this section. Where appropriate, the analyses presented in this section use the v5.0.2 base simulation (CMAQv5.0.2_Base) for comparison to the scientific improvement while for other improvements the v5.1 base simulation is used for comparison. In each case, the simulations being compared are noted. Table 2 provides a description of the
CMAQ model simulations referred to in the following sections.

**4.1 WRF and CMAQ meteorological updates**



As discussed in section 2.1, there were several significant corrections/improvements made to the meteorological calculations in both WRF and CMAQ. While the focus of this work is on updates to the CMAQ model, certain options within WRF and CMAQ are linked, and therefore it is necessary to discuss the WRF model updates alongside the corresponding CMAQ model updates.

5 Figure 2 shows the cumulative impact that all the meteorological changes in WRF and CMAQ (i.e. changes to ACM2 and MOL) had on $O_3$ and $PM_{2.5}$ in January and July by comparing the CMAQv5.0.2_Base simulation to a CMAQv5.0.2 simulation using WRFv3.7 (CMAQv5.0.2_WRFv3.7) which includes the ACM2 and MOL updates. The effect of the changes on $O_3$ in January is mixed, with some areas (e.g. Florida, Chicago and the Northwest) showing a relatively large (2.5 ppbv) increase in $O_3$, while other areas (e.g. Southwest and Texas panhandle) show a relatively large decrease (-2.5 ppbv) in $O_3$. For $PM_{2.5}$, the differences in 10 January are generally small and isolated, however there is a relatively large increase in $PM_{2.5}$ (>2.5 $\mu gm^{-3}$) in the San Joaquin Valley (SJV) of California due to the updates, which combined with the decrease in $O_3$ there as well, indicates a likely reduction in PBL height and mixing as the cause. There are also some relatively large decreases (1.5 – 2.0 $\mu gm^{-3}$) in $PM_{2.5}$ in the Northeast and around in the Great Lakes region (i.e. Chicago). Otherwise, most of the remaining impacts on $PM_{2.5}$ are relatively small (< 1.0 $\mu gm^{-3}$).

For July, the meteorological updates in WRF and CMAQ result in exclusively increased $O_3$ mixing ratios over land, which are considerably larger than the impacts in January. The largest increases (4.0 – 10.0 ppbv) occur in the eastern U.S., particularly in the Southeast. Smaller increases of 2.0-4.0 ppbv occur across much of the U.S., while in the Gulf of Mexico and the Caribbean $O_3$ mixing ratios decrease roughly 2.0 – 6.0 ppbv across a large area. The difference in $PM_{2.5}$ in July is similar to that in January, with 20 mostly small, isolated increases or decreases occurring in the eastern U.S. The largest increase (2.0 – 2.5 $\mu gm^{-3}$) occurs in the southern Ohio Valley (Kentucky and West Virginia), while the largest decreases (> 2.5 $\mu gm^{-3}$) occur in Louisiana and Texas (i.e. Houston).

It makes intuitive sense to see summertime $O_3$ mixing ratios increasing due to the meteorological changes in WRF and CMAQ, 25 since the net effect of the changes was to increase mixing, particularly in the late afternoon and early evening, which in turn decreases the amount of NO titration of $O_3$ that occurs in the model and ultimately results in higher $O_3$ mixing ratios on average. Conversely, $PM_{2.5}$ concentrations would be expected to decrease due to the increased mixing in the model, which would effectively decrease the concentrations of primary emitted pollutants (e.g. EC and OC), which was generally seen in areas with the largest emissions (i.e. urban areas). However, the spatial heterogeneity of $PM_{2.5}$ formation in the atmosphere results in both increases and 30 decreases in $PM_{2.5}$.

### 4.2 Aerosol updates

Several new SOA species from anthropogenic VOCs (i.e. AALK1, AALK2, APAH1, APAH2 and APAH3; Table 1) were added to AERO6 in v5.1 that are not present in v5.0.2. Figure 3 shows the difference in the monthly average sum total concentration of these five species for January and July 2011 between the CMAQv5.0.2_Base and CMAQv5.1_Base simulations. Since none of 35 these species were present in v5.0.2, the difference totals in Figure 3 represent the additional SOA mass that these five species contribute to the total $PM_{2.5}$ mass in v5.1. For both January and July, the monthly average concentration of these species is small, ranging between 0.0-0.1 $\mu gm^{-3}$, with the largest concentrations in the eastern half of the U.S., particularly in the upper Midwest. Since these species are not routinely observed and unique tracers have not been identified for alkene SOA, no comparison with observations is made here. Overall these new species represent a small addition to the total $PM_{2.5}$ concentration in the model.





Along with the introduction of the new SOA species above, the pathways for the formation of acid enhanced isoprene SOA were also updated. The bottom panels in Figure 3 show the monthly average difference in the sum of the species containing isoprene SOA (AISO1, AISO2, AISO3 and AOLGB) between v5.1 and v5.0.2 (v5.1 – v5.0.2). For January, the difference in the sum of

these species is relatively small, with minimum and maximum values peaking around ±0.5 µgm$^{-3}$ consistent with the fact that isoprene emissions are low in winter. For July the difference is always positive (v5.1 higher than v5.0.2) and much larger compared to January, with peak differences exceeding 2.5 µgm$^{-3}$, primarily in the areas with the highest aerosol $SO_4^{2-}$ concentrations (i.e. Ohio Valley). Therefore, the updated IEPOX-SOA formation pathways in v5.1 represent a potentially significant contribution to the total $PM_{2.5}$, particularly during the summer. Increased isoprene emissions in v5.1 with BEIS v3.61 compared to v5.0.2 with

BEIS v3.14 also contribute to the larger contribution of isoprene SOA in v5.1.

### 4.3 Cloud model and in-line photolysis updates

Changes in the photolysis/cloud model treatment in v5.1 have potentially significant impacts on the $O_3$ and $PM_{2.5}$ estimates from the model. Figure 4 shows the difference in $O_3$ and $PM_{2.5}$ for the CMAQv5.1_Base simulation and the CMAQv5.1_RetroPhot simulation (see Table 2 for simulation description). The CMAQv5.1_RetroPhot simulation is the same as the CMAQv5.1_Base

simulation except it employs the same (old) photolysis/cloud model treatment as in v5.0.2. For January, $O_3$ mixing ratios (Figure 4a) and $PM_{2.5}$ concentrations (Figure 4c) are both higher across the Southeast and portions of California in the v5.1 simulation, indicating that v5.1 has much less photolysis attenuation due to the updates in the representation of cloud effects on photolysis.

The impact of the updated photolysis in v5.1 is considerably larger in July (when there is more convection) than in January. Peak

$O_3$ differences in January were around 2.0 ppbv, whereas in July peak differences of greater than 5.0 ppbv occur over the Great Lakes (where low PBL heights can enhance the impact of changes in $O_3$). However, in general the difference in $O_3$ mixing ratios is larger in both magnitude and spatial coverage in July compared to January, indicating that the updated photolysis/cloud model treatment in v5.1 increases $O_3$ to a greater extent in July compared to January, as expected due to increased photolysis rates in the summer compared to winter. Overall, differences in $O_3$ in July range on average from 1.0 to 3.0 ppbv, with larger differences

occurring in the major urban areas (e.g. Atlanta, Charlotte and Los Angeles) and off the coast of the Northeast corridor. The change in $PM_{2.5}$ is also larger (both in magnitude and spatial coverage) in July than January, and is primarily confined to the eastern U.S. and results in a roughly 0.1 to 0.5 µgm$^{-3}$ increase in $PM_{2.5}$ in v5.1, with the maximum increase located over the Great Lakes region and areas to the south, the result of increased SOA and gas-phase production of $SO_4^{2-}$ due to greater $OH^-$ concentrations in v5.1.

Two notable issues remain with the v5.1 modeled cloud parametrization. The photolysis cloud parameterization in v5.1 produces more clouds over water compared to the WRF parameterization, which is itself biased high for some parts of the Atlantic Ocean compared to GOES. This issue will be addressed by science updates planned for the CMAQ system and evaluation results are expected to improve in the next CMAQ release (See Section 7.4). A more significant issue, from an air quality perspective, is the under-prediction of clouds over much of the Eastern and West Central US in the WRF predicted

clouds, which is now directly passed along to CMAQ. This misclassification of modeled clear sky conditions can contribute to an over prediction of $O_3$ in these regions. Resolving this issue will require changes to the WRF cloud parameterization. Future research will also include changing the sub-grid cloud treatment currently used in the CMAQ system to be consistent with the sub-grid parameterization used in WRF. Section S.1 in the supplemental material provides a table with additional evaluation



metrics of the modeled clouds over oceans versus over land and also describes how cloud albedo was calculated for the three model simulations.

### 4.4 Atmospheric chemistry updates

As detailed in section 2.4, numerous updates were implemented in the representation of atmospheric chemistry in v5.1. It would

be extremely cumbersome to attempt to isolate the impact of each chemistry update individually. Instead, in order to assess the overall impact that the combined chemistry changes have on the model results, model comparisons are conducted using the CMAQv5.1_Base simulation, which employs the CB05e51 chemical mechanism (the v5.1 default chemical mechanism) and the CMAQv5.1_TUCL simulation (see Table 2 for description). The CMAQv5.1_TUCL simulation is the same as the CMAQv5.1_Base simulation except that it employs the CB05tucl chemical mechanism (Whitten et al., 2010; Sarwar et al., 2012),

the default mechanism in v5.0.2. Note that the aerosol updates discussed in section 4.2 were incorporated into the CB05e51 chemical mechanism (in the past that portion of the aerosol chemistry was separate from the gas-phase chemical mechanism). As such, differences between the CMAQv5.1_TUCL and CMAQv5.1_Base simulations include impacts from those changes (i.e. Figure 2). In order to isolate primarily just the effect on $PM_{2.5}$ from the atmospheric chemistry changes, the organic matter (AOMIJ; See S.2 and S.3 for species definition descriptions) mass has been removed from the comparisons of total $PM_{2.5}$ mass discussed

below.

Figure 5 shows the difference in monthly average $O_3$ and $PM_{2.5}$ for January and July between the CMAQv5.1_Base and CMAQv5.1_TUCL simulations. For January, $O_3$ mixing ratios are higher in the CMAQv5.1_Base simulation versus the CMAQv5.1_TUCL simulation, indicating generally higher $O_3$ during winter due to the updates in the CB05e51 mechanism.

However, the overall impact of CB05e51 on $O_3$ is generally small, with maximum differences of around 1.0 ppbv, primarily along the southern coastal areas of the U.S. $PM_{2.5}$ is also higher in January in the CMAQv5.1_Base simulation, indicating higher $PM_{2.5}$ with the CB05e51 mechanism versus CB05tucl. Changes in $PM_{2.5}$ primarily occur in the eastern U.S., with differences ranging between 0.0 to -0.4 $\mu gm^{-3}$, with a notable larger isolated difference ($> 1.0\ \mu gm^{-3}$) in the SJV.

For July, $O_3$ mixing ratios are higher across most areas in the CMAQv5.1_Base simulation (due to the CB05e51 mechanism), primarily across northern portions of the U.S., the Great Lakes region and in California (i.e. Los Angeles and SJV). Most increases in $O_3$ in the CMAQv5.1_Base simulation range between 0.6 and 1.2 ppbv, however larger increases of over 3.0 ppbv occur in southern California and over Lake Michigan (likely influenced by low PBL heights). A small area of lower $O_3$ mixing ratios occurs off the eastern coast of the U.S. For $PM_{2.5}$ in July, the difference in $PM_{2.5}$ due to the CB05e51 chemical mechanism is relatively

small, with differences in concentrations generally ranging from $\pm 0.50\ \mu gm^{-3}$ in the eastern U.S.

### 5 Evaluation of CMAQv5.1

In this section, comparisons are made of the operational performance of the CMAQv5.0.2_Base and CMAQv5.1_Base simulations by initially comparing the simulations to each other (model to model) and then evaluating them against a wide variety of available air quality measurements (see section 3). Several common measurements of statistical performance are used, namely mean bias

(MB), mean error (ME), root mean square error (RMSE) and correlation. Note that representativeness (incommensurability) issues are present whenever gridded values from a deterministic model such as CMAQ are compared to observed data at a particular point in time and space, as deterministic models calculate the average outcome over a grid for a certain set of given conditions,



while the stochastic component (e.g. sub-grid variations) embedded within the observations cannot be accounted for in the model (Swall and Foley, 2009). These issues are somewhat mitigated for networks that observe for longer durations, for example the CSN and IMPROVE networks which are daily averages and the CASTNET observations which are weekly averages. The longer temporal averaging helps reduce the impact of stochastic processes, which can have a large impact on shorter (e.g. hourly) periods
of observation (Appel et al., 2008).

There are several important differences to keep in mind between the comparison of the CMAQv5.0.2_Base and CMAQv5.1_Base simulations beyond the obvious changes to the model process representations discussed in the previous sections. First, the simulations use different versions of WRF (as discussed in Sections 2.2 and 4.1). This was intentional, as it was determined that
the changes made from WRF v3.4 (used in the CMAQv5.0.2_Base simulation) to WRF v3.7 (used in the CMAQv5.1_Base simulation) and subsequent required changes made to the CMAQ code represent a change to the overall WRF-CMAQ modeling system and therefore should be evaluated together. Second, the emission inventories for the two base runs are slightly different, as discussed in Section 3. While the changes between the emission inventories are generally minor, they do represent another difference between the simulations, and where possible the effect of the different inventories on the model performance is noted.
Finally, it should also be noted that the windblown dust treatment was employed in the CMAQv5.0.2_Base simulation but not in the CMAQv5.1_Base simulation. This was due to issues with the implementation of the updated windblown dust treatment in v5.1 that were not discovered until after the model was released and the CMAQv5.0.2_Base simulation was completed. However, the contribution of windblown dust to total $PM_{2.5}$ in v5.0.2 tends to be small and episodic and therefore should not constitute a significant impact to the performance differences between v5.0.2 and v5.1, especially for the monthly averages generally shown
here.

### 5.1 $PM_{2.5}$

Figure 6 shows the seasonal average difference in model simulated $PM_{2.5}$ between v5.0.2 and v5.1 (CMAQv5.1_Base – CMAQv5.0.2_Base), with cool colors indicating a decrease in $PM_{2.5}$ in v5.1 (versus v5.0.2) and warm colors indicating an increase in $PM_{2.5}$. Figure 7 shows the seasonal mean bias (MB) for $PM_{2.5}$ for the CMAQv5.1_Base simulation, while Figure 8 shows the
change in the absolute value of the seasonal mean bias (|MB|) in $PM_{2.5}$ between the CMAQv5.0.2_Base and CMAQv5.1_Base simulations. Cool colors indicate smaller $PM_{2.5}$ |MB| in the CMAQv5.1_Base simulation (versus the CMAQv5.0.2_Base simulation), while warm colors indicate larger |MB| in the CMAQv5.1_Base simulation.

During winter, v5.1 predicts lower $PM_{2.5}$ concentrations in the eastern U.S. and portions of western Canada compared to v5.0.2,
but higher $PM_{2.5}$ concentrations in the SJV and isolated portions of Mexico and Alabama (due to emissions changes affecting that state) (Figure 6). $PM_{2.5}$ is largely overestimated in the eastern U. S. and underestimated in the western U.S. in the winter in CMAQv5.1_Base simulation (Figure 7a). The change in |MB| between v5.0.2 and v5.1 is negative (reduced MB in v5.1) across the majority of the sites, with relatively large reductions (3-5 $\mu gm^{-3}$) in |MB| in the Northeast, upper Midwest (i.e. Great Lakes region), SJV, and portions of the mid-Atlantic (e.g. NC) (Figure 8a). Alabama is a notable exception, with the MB increasing in
the v5.1 simulation due to changes in the emissions inventory. Figure S1 presents a histogram of the change in $PM_{2.5}$ |MB| using the same data and color scale as in Figure 8. It's clear from the histogram the large percentage (69.2%) of sites where the |MB| decreases in the v5.1 simulation in the winter (Figure S1a), demonstrating a significant improvement in the $PM_{2.5}$ performance for v5.1 versus v5.0.2.

The diurnal profile of PM$_{2.5}$ for winter (Figure S2) shows a relatively large decrease in MB throughout most of the day in the v5.1 versus v5.0.2, particularly during the overnight, morning and late afternoon hours. A similar improvement is seen in the RMSE, while the correlation also improves for all hours. Finally, Figure 9 shows seasonal and regional stacked bar plots of PM$_{2.5}$ composition (SO$_4^{2-}$, NO$_3^-$, NH$_4^+$, EC, OC, Soil, NaCl, NCOM, and PM Other), where Soil is based on the IMPROVE soil equation

and contains both primary and secondary sources of soil (Appel et al., 2013), and PM Other represents the unspeciated PM mass in the inventory (see Appel et al., 2008) The five regions shown in Figure 9 are Northeast (ME, NH, VT, MA, NY, NJ, MD, DE, CT, RI, PA, D.C., VA and WV), Great Lakes (OH, MI, IN, IL and WI), Atlantic (NC, SC, GA and FL), South (KY, TN, MS, AL, LA, MO, OK and AR) and West (CA, OR, WA, AZ, NV, NM). These regions are derived from principle component analysis to group states with similar PM$_{2.5}$ source regions together. For winter, the total PM$_{2.5}$ high bias is reduced across all five regions,

with most of the improvement coming from primary emitted species such as EC and OC, non-carbon organic matter (NCOM; Table S1) and PM Other, indicating that improvements in the representation of mixing under stable conditions helped in reducing the high bias. Still, a large bias remains for OC, which may be due to an overestimation of the residential wood combustion in the NEI.

For spring, the changes in PM$_{2.5}$ are much more isolated than in winter, with the largest decreases occurring around Montreal (Canada), NC, and portions of the desert Southwest (lack of wind-blown dust in v5.1 likely contributes to this decrease in the desert Southwest). A small increase is again noted in Alabama due to emissions changes between the v1 and v2 emissions. The MB for PM$_{2.5}$ in the spring is relatively small, with most sites showing an underestimation of 1-3 µgm$^{-3}$, while some overestimations of PM$_{2.5}$ occur in the Northeast, Great Lakes and Northwest coast (Figure 7b). As expected with the relatively

small changes in PM$_{2.5}$ concentrations in spring, the |MB| does not change significantly between v5.0.2 and v5.1, with most changes in |MB| less than ±1.0 µgm$^{-3}$ (Figure 8b). Some slightly larger decreases in |MB| occur in the Northeast and AL, while some larger increases in |MB| occur in the desert Southwest. About half (49.8%) of the sites show an improvement in |MB| (Figure S1b). The diurnal profile of PM$_{2.5}$ for spring (Figure S3) shows a consistent underestimation of PM$_{2.5}$ throughout most the day in the v5.0.2 simulation, which becomes larger in the v5.1 simulation with the overall decrease in PM$_{2.5}$ in the spring. However, the RMSE is

lower during the overnight, morning and afternoon hours in the v5.1 simulation, while the correlation also improves throughout most of the day (exception being 1pm to 4pm LST). Total PM$_{2.5}$ MB improves in three of the five regions shown in Figure 9, with most of the improvement again coming from reductions in the primary emitted species.

In the summer, PM$_{2.5}$ is considerably higher (> 5.0 µgm$^{-3}$) across the eastern U.S. in the CMAQv5.1_Base simulation, particularly

in MS, AL, GA and portions of the Ohio Valley, while PM$_{2.5}$ is lower in isolated areas in eastern NC, Montreal, Canada and small areas in the southwest U.S. and Mexico. The increase in PM$_{2.5}$ is primarily due to the updates to the IEPOX-SOA chemistry in v5.1 (Figure 2), updates to BVOC emissions in BEIS v3.61 (approximately 1.0 µgm$^{-3}$ increase PM$_{2.5}$ in the southwest U.S.), and ACM2/MOL updates in WRF and CMAQ (Figure 1), with smaller contributions from the updates in CB05e51 chemical mechanism (Figure 5) and updates to the clouds/photolysis (Figure 3). Despite the increase in PM$_{2.5}$ with v5.1, PM$_{2.5}$ remains

largely underestimated in the summer, with largest underestimations in the southeast U.S. and California (Figure 7c). However, the result of the widespread increase in PM$_{2.5}$ in the v5.1 simulation is a similar large, widespread reduction in the |MB| across the eastern U.S., particularly in the Southeast (except eastern NC and FL) and the Ohio Valley, where reductions in |MB| range from 3.0 - 5.0 µgm$^{-3}$ (Figure 8c). Small increases in the |MB| (typically less than 2.0 µgm$^{-3}$) occur in eastern NC and FL, and isolated areas in the western U.S. Of all the sites, 60.4% showed an improvement in |MB|, with a large number of sites showing reductions

in |MB| greater than 5.0 µgm$^{-3}$ (Figure S1c). PM$_{2.5}$ is underestimated throughout the day in both v5.0.2 and v5.1 (Figure S4) in





summer, with the underestimation improving slightly in v5.1, particularly during the afternoon and overnight hours. RMSE improves during the daytime hours in v5.1, while correlation is significantly higher in v5.1 than v5.0.2 throughout the entire day. Total PM$_{2.5}$ is underestimated by the model in four of the five regions (West region being the exception), but improves in three of those four regions with v5.1, with small increases in SO$_4^{2-}$ and NH$_4^+$, and larger increases in OC and NCOM contributing to the

improvement (Figure 9).

For the fall, the difference in PM$_{2.5}$ between v5.0.2 and v5.1 is again small (very similar to the spring), with the largest increases occurring in AL and western Canada, and the largest decreases occurring in Montreal, Mexico and isolated areas in the eastern and Midwest U.S. (Figure 6). The overall pattern in MB is somewhat similar to that of the spring (Figure 7d), with relatively small

MBs in the Eastern U.S. ($\pm2.0$ µgm$^{-3}$) and larger MBs along the west coast (underestimated in California and overestimated in the Northwest). As expected, the change in the |MB| between v5.0.2 and v5.1 is also relatively small in the fall, with the majority of the sites having a change in |MB| of less than $\pm2.0$ µgm$^{-3}$ (Figure 8d), while 65.3% of the sites show a reduction in |MB| (Figure S1d). The average diurnal profile of PM$_{2.5}$ in the fall is similar to the spring, with improved MB in v5.1 during the overnight, morning and late afternoon/evening hours and lower RMSE and higher correlation throughout the entire day (Figure S5). Total

PM$_{2.5}$ is overestimated in all five regions in the fall (Figure 9), but improves in v5.1 in four of those regions (exception being the South), with decreases in the primary emitted species responsible for most of the improvement.

### 5.2 Ozone

For the winter, O$_3$ widely decreases in the CMAQv5.1_Base simulation versus the CMAQv5.0.2_Base simulation across the

western U.S., with the seasonal average decreases ranging between 1.0 – 3.0 ppbv, and several areas where decreases exceed 3.0 ppbv, primarily over the oceans (Figure 10). In the eastern U.S., the change in O$_3$ is relatively small and isolated, the exception being along the coast of LA and a small portion of FL, where increases in O$_3$ exceed 5.0 ppbv. Ozone is underestimated at most sites across the northern portion of the U.S., with the largest underprediction occurring in Colorado, Wyoming and Utah. Despite the decreases in O$_3$ with v5.1, O$_3$ is still overestimated in Florida, along the Gulf Coast of Mexico, the Southwest U.S. and in

California (Figure 11a). There is a widespread reduction in the O$_3$ |MB| in California and increased |MB| in the upper Midwest with v5.1, while across the rest of the domain the change in |MB| is relatively small and mixed in direction (Figure 12a). The majority of the change in O$_3$ falls between $\pm5.0$ ppbv, with slightly more sites (55.8%) showing a reduction than increase in |MB| (Figure S6a). The average diurnal profile of O$_3$ in the winter (Figure S7) shows lower MB and RMSE, and higher correlation throughout the day with v5.1 versus v5.0.2. The NO$_X$ also generally improves throughout the day in winter, with decreases in MB

and RMSE in the afternoon/early evening and increased correlation throughout the day (Figure S8).

The pattern of change in O$_3$ between v5.0.2 and v5.1 in spring is similar to winter, with lower O$_3$ mixing ratios in the western U.S. and higher mixing ratios in the eastern U.S. in v5.1 compared to v5.0.2 (Figure 10b). Decreases in O$_3$ mixing ratios in the western U.S. in v5.1 range from roughly 1.0 – 3.0 ppbv (similar to winter), while in the eastern U.S. the increases generally range from 1.0

– 2.0 ppbv, with isolated areas of larger increases. The MB of O$_3$ for the v5.1 simulation primarily ranges from slightly over- to slightly underestimated across most the sites, with larger overestimations along the Gulf Coast and larger underestimations in the western U.S. (Figure 11b). The change in |MB| between v5.0.2 and v5.1 shows mixed results (Figure 12b), with slight increases and decreases across much of the eastern U.S. and a relatively large increase in |MB| in the Midwest (Colorado and Wyoming). The |MB| mostly improved across the Gulf Coast and in California due to reduced O$_3$ mixing ratios from the new marine halogen





chemistry and enhanced $O_3$ deposition to ocean surfaces. Roughly half (49.4%) of the sites showed a reduction in |MB| (Figure S6b) with v5.1. The diurnal profile of $O_3$ for spring (Figure S9) shows a large improvement in MB in v5.1 in the late afternoon and evening (4pm to 10pm LST), with similar improvements in RMSE and slightly higher correlations in the afternoon and evening hours. The $NO_X$ diurnal profile also shows a large decrease in the late afternoon and early evening MB and RMSE, improved

representation of the morning rush-hour peak, and correlation improves throughout the day as well (Figure S10).

For the summer, the pattern of change in $O_3$ is markedly different from the winter and spring, with widespread, large increases in $O_3$ mixing ratios across the eastern U.S. and decreases in $O_3$ mixing ratios in the Gulf of Mexico, southern FL and over the eastern Atlantic (Figure 10c) ocean. Increases in $O_3$ in v5.1 in the eastern U.S. range from 2.0 – 10.0 ppbv, with isolated areas of larger

increases in the major urban areas (e.g. Chicago and Atlanta). Smaller increases in $O_3$ occur in the western U.S., particularly southern California and the SJV. Decreases in $O_3$ over the oceans are also large as a result of the introduction of a chemical sink for $O_3$ due to halogen chemistry in v5.1 (Sarwar et al., 2015), with some decreases exceeding 10.0 ppbv. The MB of $O_3$ for the v5.1 simulation shows widespread overestimations in the eastern U.S., particularly along the Gulf of Mexico, while in the western U.S. the MB is mixed, with the largest overestimations occurring along the California coast (Figure 11c). As expected, the

consequence of the widespread increase in $O_3$ in the eastern U.S. in v5.1 is a corresponding widespread increase in the |MB| compared to v5.0.2, particularly in the Mid-Atlantic and Southeast (Figure 12). Ozone |MB| decreases along the coast of FL and along the Gulf of Mexico, likely the result of decreased $O_3$ over the water from changes in the deposition of $O_3$ over water and the inclusion of the halogen chemistry update. The change in |MB| in the western U.S. is mixed, with some areas showing improved |MB| (e.g. SJV), while others show increased |MB| (e.g. southern California). And while the diurnal profiles of $O_3$ show that MB

increases throughout much of the day in v5.1 (exception being 12am – 5am), RMSE decreases substantially during the overnight hours and the correlation improves throughout the entire day (Figure S11). The $NO_X$ concentrations are lower throughout the day in v5.1 compared to v5.0.2, which results in large improvements in the MB in the morning and afternoon periods and slightly increased MB in the middle of the day, while RMSE and correlation improve throughout the day (Figure S12).

For the fall, the pattern of change in $O_3$ for v5.1 versus v5.0.2 is nearly identical to spring (Figure 10), with widespread decreases in $O_3$ in the western U.S. (possibly due to reduced cloud mixing and entrainment from the free troposphere) and mostly small increases in $O_3$ in the eastern U.S., with the exception of larger increases in several of the major urban areas (e.g. St. Louis and Atlanta). The changes are generally small, between ±2.0 ppbv, with isolated areas of larger increases or decreases. Ozone is also lower over the Pacific and Atlantic oceans and the Gulf of Mexico. While the change in $O_3$ between v5.0.2 and v5.1 very similar

to the spring, the MB pattern for v5.1 is not. Unlike the spring where $O_3$ was largely underestimated, in the fall $O_3$ continues to be overestimated across most of the sites much like the summer (Figure 11d). The Midwest shows lowest overall MB, while the east and west coasts have large overestimations of $O_3$. The increased $O_3$ in the eastern U.S. with v5.1 results in generally higher |MB| compared to v5.0.2, while in the western U.S. the result is slightly lower |MB| on average, the exception being southern California (Figure 12d). As was the case in the spring, slightly less than half the sites (48.9%) showed a reduction in |MB|, with the majority

of the change falling between ±5.0 ppbv (Figure S6d). The diurnal profile of $O_3$ in the fall shows increased MB in v5.1 compared to v5.0.2 throughout most of the day, but again lower RMSE and higher correlation throughout the entire day (Figure S13). Similar to the other seasons, the diurnal profile of $NO_X$ in the fall shows lower MB in the afternoon and lower RMSE and higher correlation throughout the entire day in v5.1 (Figure S14).

**5.2 Comparisons to Aircraft Measurements**





In addition to the routine ground-based measurements, the DISCOVER-AQ (https://www.nasa.gov/mission_pages/discover-aq/) campaign that took place over the Baltimore, MD and Washington, D.C. area in July 2011 provides a unique measurement dataset containing both ground-based and upper-air (i.e. aircraft) measurements. Not only does this allow evaluation of the model performance throughout the PBL, the unique measurements also allow evaluation against species that are not routinely observed,

specifically peroxy nitrates (PNs) and alkyl nitrates (ANs), both of which are important species in $O_3$ chemistry. The National Oceanic and Atmospheric Administration (NOAA) P3B aircraft performed measurement flights on a number of days during the DISCOVER-AQ campaign. Those flights included vertical spirals over several locations, one of which was Edgewood, MD, a site that often measures very high $O_3$, and in recent years has measured some of the highest $O_3$ in the eastern U.S. Figure 13 shows vertical profiles of observed and CMAQ (v5.0.2 and v5.1) simulated $O_3$, $NO_2$, $NO_y$, ANs, PNs and $HNO_3$ for the Edgewood site

on July 5, 2011. While $O_3$ is underestimated throughout the PBL by both versions of the model on that day, the underestimation is significantly improved in the v5.1 simulation. $NO_2$ and $NO_y$ are overestimated throughout the PBL by both versions of the model, but again, the overestimation is greatly improved in the v5.1 simulation. The PNs, ANs, and $HNO_3$ show mixed results, with the ANs performance improving, the PNs performance degrading and the $HNO_3$ performance relatively unchanged with v5.1. Note that there has been an update in the recommended PAN formation and degradation equilibrium constant (http://iupac.pole-

ether.fr) which lowers the predicted PAN concentrations in CMAQ and is currently being examined for its impact on other species. On this particular day, v5.1 generally shows a large improvement in performance over v5.0.2.

## 6 Modeled Response to Emission Changes

One of the primary applications of air quality models is to determine the impact that changes (e.g. reductions from abatement

strategies) in emissions have on ambient air quality. Examples of this type of application include Federal rules and State Implementation Plans (SIPs). In this type of application, the air quality model is run using both baseline (often current year) and future year emissions (when emissions are typically lower due to state and national regulatory efforts) and then the change in criteria pollutant (e.g. $O_3$ and $PM_{2.5}$) concentrations between the two simulations is quantified in order to assess the impact (benefit) that emission reductions will have on future ambient air quality. As such, it is important to establish the ability of the model to

accurately simulate the future ambient air quality given a known change in emissions, which here is referred to as the model responsiveness (to emission changes).

Some previous analyses comparing observed changes in ambient air quality (over periods witnessing large reductions in emissions) to CMAQ estimated changes in ambient air quality (with estimated reductions in emissions) during the same period have shown

that the model tends to underestimate the observed change in ambient $O_3$, suggesting the model may be under-responsive to the emission reductions impacting $O_3$ (Gilliland et al., 2008; Foley et al., 2015). The over/under responsiveness of the model to emission projections can have implications in the planning process for determining the extent to which emissions must be reduced in order to meet future air quality standards. In the following sections, we examine the model responsiveness to emission reductions in CMAQ v5.0.2 and v5.1 by computing the ratio of maximum daily 8-hr average (MDA8) $O_3$ mixing ratios and total $PM_{2.5}$ (and

select $PM_{2.5}$ component species) between simulations using the base emissions inventories and those employing 50% reductions in $NO_x$, VOC and $SO_x$ emissions in order to estimate a model responsiveness to the emission reductions for each version of the model. The model responsiveness for v5.1 is then compared to that of v5.0.2 to determine whether the model responsiveness increased, decreased or was unchanged in the new version of the model.



## 6.1 O$_3$

Figure 14 shows the difference in the ratio (emission cut simulation / base simulation) of MDA8 O$_3$ for the 50% cut in anthropogenic NO$_x$ and VOC scenarios, binned by model MDA8 O$_3$ mixing ratio. Values greater than zero indicate v5.1 is more responsive to the NO$_x$ or VOC cut than v5.0.2, while values less than zero indicate v5.1 is less responsive than v5.0.2. For both

January and July, the median difference in ratio values for all bins for the 50% NO$_x$ cut scenario are greater than zero, indicating that v5.1 is more responsive than v5.0.2 to the cut in NO$_x$. For the 50% cut in VOC emissions the difference in the ratio values is mixed across the two months and the different bins. For January, all of the bins indicate that v5.0.2 is more responsive than v5.1 to the 50% VOC cut, with the greatest difference occurring for MDA8 O$_3$ mixing ratios greater than 65 ppbv. For July, v5.1 is slightly more responsive to the VOC cut for MDA8 O$_3$ mixing ratios less than 75 ppbv and less responsive for MDA8 O$_3$ mixing

ratios greater than 85 ppbv.

## 6.2 PM$_{2.5}$

Figure 15 shows the ratio (emission cut simulation / base simulation) of PM$_{2.5}$ and select PM$_{2.5}$ component species between v5.0.2 (blue) and v5.1 (red) for January (top) and July (bottom) for a 50% cut in anthropogenic emissions of NO$_x$, VOC and SO$_x$. For January, the overall response of modeled PM$_{2.5}$ (PMIJ) to a 50% reduction in NO$_x$ is primarily driven by a decrease in nitrate and

its associated ammonium. CMAQ v5.1 PM$_{2.5}$ is slightly less responsive to NO$_x$ reductions compared to v5.0.2, but is still overall quite similar. The VOC cut shows greater response with v5.1 than v5.0.2 in January in ANCOMIJ (non-carbon organic matter attached to primary organic carbon; Simon and Bhave, 2012), AUNSPECIJ (unspeciated PM), AOMIJ (all organic matter), AORGAJ (SOA from anthropogenic VOCs) and AORGBJ (SOA from biogenic VOCs) and total PM$_{2.5}$ (see species definition files in supplemental material). Note that the letters I and J after the species name indicate which CMAQ modal distributions are

being included in the total species mass, with I indicating the Aitken mode and J indicating the Accumulation mode. Since NCOMIJ is nonvolatile, its change reflects how reducing VOCs changes oxidants such as OH. In general, the model PM$_{2.5}$ is not very sensitive to VOC cuts in January. And finally for the 50% SO$_x$ cut scenario, PM$_{2.5}$ is only slightly less responsive, with all the species being similarly responsive to the SO$_x$ cut using v5.1 compared to v5.0.2.

For July, the NO$_x$ cut scenario with v5.1 shows greater responsiveness for the ASO4IJ (sulfate), ANH4IJ (nitrate), AECIJ (elemental carbon), APOAIJ (primary organic aerosol), AORGCJ (SOA from glyoxal and methylglyoxal processing in clouds) species and total PM$_{2.5}$ versus v5.0.2. For the VOC cut scenario, the AORGAJ species show increased responsiveness with v5.1. CMAQ v5.1 alkane SOA is not dependent on NO$_x$ levels or HO$_2$:NO ratios, so the decrease in VOC precursors have a more direct effect than for the aromatic systems (the only AORGAJ in v5.0.2), where decreasing the VOC precursors can also modify the

HO$_2$:NO ratio and thus yields. CMAQ v5.1 PMIJ becomes slightly more responsive to SO$_x$ as a result of an increased sensitivity of biogenic SOA to sulfur containing compounds. This link results from the IEPOX acid-catalyzed SOA in the model which has been shown to be correlated with sulfate (Pye et al. in prep).

## 7 Discussion

A new version of the CMAQ model (v5.1) containing numerous scientific updates has been released and evaluated in terms of

operational performance and response to changes in inputs (i.e. emissions). Specifically, updates were made to the ACM2 scheme in both WRF and CMAQ to improve the vertical mixing in both models, along with updates to the MOL calculation, which also directly impacted the vertical mixing in the WRF-CMAQ system. The overall net effect of these updates was to increase the





ventilation in the model, particularly during the transition periods (morning and evening), which in turn reduced the concentration of primary emitted species (e.g. $NO_X$ and OC) and consequently increased simulated $O_3$ (a result of reduced $NO_x$ titration) and decreased $PM_{2.5}$ concentrations due to greater dilution. Several new SOA formation pathways and species were added to v5.1, resulting in increased SOA, particularly in the southeast U.S., and improved $PM_{2.5}$ performance in the summer as $PM_{2.5}$ is often

underestimated in CMAQ during the summer.

The in-line photolysis model within CMAQ was updated in v5.1. Cloud cover for the photolysis model in v5.0.2 used a single cloud deck with a constant cloud fraction and water droplet mixing ratio. In v5.1, multiple cloud decks with variable cloud fractions and multiple types of water condensates are used in the photolysis model to be more consistent with the WRF meteorological

model and the CMAQ cloud model. The net effect of this change was to decrease the amount of sub-grid in the photolysis calculation in v5.1, which in turn results in higher photolysis rates and thus higher predicted $O_3$ mixing ratios on average. In addition to the change to the photolysis model, the refractive indices for aerosol species are now both wavelength and composition dependent. Changes in aerosol scattering and extinction also introduce run-time options for how to calculate their optical properties and allow the user to specify which aerosol mixing model and method to use to solve the Mie scattering theory. The atmospheric

chemistry in the model has also been updated from CB05tucl to CB05e51 in v5.1, which includes, among other things, updates to the $NO_y$ reactions, additional isoprene extensions, explicit representation of several HAPs, and a simple parameterization of the effects of halogens on $O_3$ in marine environments. The net effect of going from CB05tucl to CB05e51 was to increase $O_3$ in the winter and summer, while increasing $PM_{2.5}$ slightly in the winter and increasing/decreasing $PM_{2.5}$ slightly in the summer.

Overall, the scientific updates in v5.1 resulted in relatively dramatic improvements in model performance for $PM_{2.5}$ in the winter and summer and small overall changes in performance during spring and fall. $PM_{2.5}$ concentrations decreased significantly in v5.1 in the winter when $PM_{2.5}$ is typically overestimated by CMAQ over the U.S., and increased significantly in the summer when $PM_{2.5}$ is typically underestimated by the model. The change in $O_3$ mixing ratios in v5.1 results in mixed improvement in MB, both spatially and temporally, with the summer showing the largest increase in MB. However, RMSE largely improves regardless of

season and shows a larger improvement spatially across the sites than MB, and the correlation is almost always higher with v5.1. Comparisons of vertical profiles of several species taken over Edgewood, MD on July 5, 2011 during the DISCOVER-AQ campaign showed improved performance in v5.1 throughout the PBL for $O_3$, $NO_2$, $NO_Y$, ANs and CO, with the PNs being the only species to show degraded performance on that day. And while the complexity of the model increased (e.g. additional species and reactions), the computational time required to complete a v5.1 simulation remained similar to v5.0.2 due to several

improvements made to the model code to increase computational efficiency.

The response of the model to changes in emission inputs was examined by comparing the ratio of the base v5.0.2 and v5.1 simulations to sensitivity simulations with 50% cuts each to anthropogenic $NO_x$, VOC and $SO_x$ emissions. CMAQv5.1 simulated MDA8 $O_3$ exhibited more responsiveness (greater reduction) to the 50% $NO_x$ cut in January and July than v5.0.2, which is

considered an improvement as previous studies suggested CMAQ $O_3$ to be under-responsive to large changes in emissions. The responsiveness of $PM_{2.5}$ to the emission cuts is more complicated than for $O_3$ since there are many more species comprising $PM_{2.5}$ and some of those have greater or smaller response with v5.1. However, the new pathways of formation for several $PM_{2.5}$ components in v5.1 generally result in greater responsiveness in v5.1 compared to v5.0.2 for the various emission cut scenarios.



Finally, a number of important science updates are in development and will be available in the next release of CMAQ (v5.2), which update or correct known issues in v5.1, and improve upon the existing science in the model. These updates include a new version of the windblown dust treatment (Foroutan et al., 2016), the Carbon-Bond 6 (CB6) chemical mechanism (Ramboll Environ, 2016), enhancements to the calculation of semi-volatile Primary Organic Aerosol (POA) and SOA from combustion sources in CMAQ

(Pye et al., 2016), and additional updates to the calculation of clouds. In addition to the model updates, a number of instrumented versions of the model (e.g. decoupled direct method, sulfur tracking) will also be released with v5.2. These updates represent potentially significant improvements over the current options in v5.1 (specifically the updated windblown dust treatment) and therefore are being made available to the community more quickly than they might have in the past.

**Code availability**

CMAQ model documentation and released versions of the source code are available at www.cmaq-model.org. The updates described here, as well as model post-processing scripts, are available upon request

**Data availability**

The raw observation data used are available from the sources identified in Section 3, while the post-processed observation data are available upon request. The CMAQ model data utilized are available upon request as well.

**Disclaimer**

The views expressed in this article are those of the author[s] and do not necessarily represent the views or policies of the U.S. Environmental Protection Agency.

**Acknowledgements**

The authors would like to think CSRA for creating the WRF meteorological inputs and emissions data used in the various model

simulations. The authors would also like to thank the DISCOVER-AQ project (DOI: 10.5067/Aircraft/DISCOVER-AQ/Aerosol-TraceGas) for providing some of the data used in this work.

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



**Table 1: New/revised SOA species in the CMAQ v5.1 AERO6 mechanism.**

| Aerosol Species | Change since v5.0.2 | Applicable Mechanism | Description of Modification |
|---|---|---|---|
| AH3OP | added | all | Hydronium ion (predicted by ISORROPIA for I+J modes), used for IEPOX uptake |
| APAH1,2 | added | cb05e51, saprc07tb, saprc07tc, saprc07tic, racm | Naphthalene aerosol from $RO_2+NO$ reactions |
| APAH3 | added | cb05e51, saprc07tb, saprc07tc, saprc07tic, racm | Naphthalene aerosol from $RO_2+HO_2$ reactions |
| AISO1,2 | updated | cb05e51, saprc07tb, saprc07tc*, racm | Aerosol from isoprene reactions $NO_3$ added to existing OH (all yields follow the OH pathway) |
| AISO3 | updated | cb05e51, saprc07tb, saprc07tc*, racm | Aerosol from reactive uptake of IEPOX on aqueous aerosol particles. Specifically intended to be the sum of 2-methyltetrols and IEPOX-derived organosulfates. |
| AALK1,2 | added | cb05e51, saprc07tb, saprc07tc, saprc07tic, racm | Alkane aerosol |
| AALK | removed | all | deprecated alkane aerosol |

*AERO6i does not include SOA from isoprene+NO3 in AISO1,2 (it is included in AISOPNNJ). AERO6i does not include IEPOX SOA in AISO3 (it is included in AITETJ, AIEOSJ, AIDIMJ, etc). AISO3 is approximately zero in AERO6i.



**Table 2: Description of the CMAQ model simulations utilized.**

| CMAQ Simulation Name | Simulation Description |
|---|---|
| CMAQv5.0.2_Base | Annual (2011) CMAQ simulation utilizing CMAQ version 5.0.2 and WRF version 3.4; NEI version 1 emissions; CB05TUCL chemical mechanism; AERO6 |
| CMAQv5.0.2_WRFv3.7 | January and July (2011) simulation utilizing CMAQ version 5.0.2 and WRF version 3.7; NEI version 1 emissions; CB05TUCL chemical mechanism; AERO6 |
| CMAQv5.1_Base | Annual (2011) CMAQ simulation utilizing CMAQ version 5.1 and WRF version 3.7; NEI version 2 emissions; CB05e51 chemical mechanism; AERO6 |
| CMAQv5.1_RetroPhot | January and July (2011) CMAQ simulation utilizing the same configuration as the CMAQv5.1_Base simulation except with the same photolysis scheme as was implemented in CMAQ version 5.0.2 |
| CMAQv5.1_TUCL | January and July (2011) CMAQ simulation utilizing the same configuration as the CMAQv5.1_Base simulation except with the TUCL chemical mechanism in place of the CB05e51 chemical mechanism |





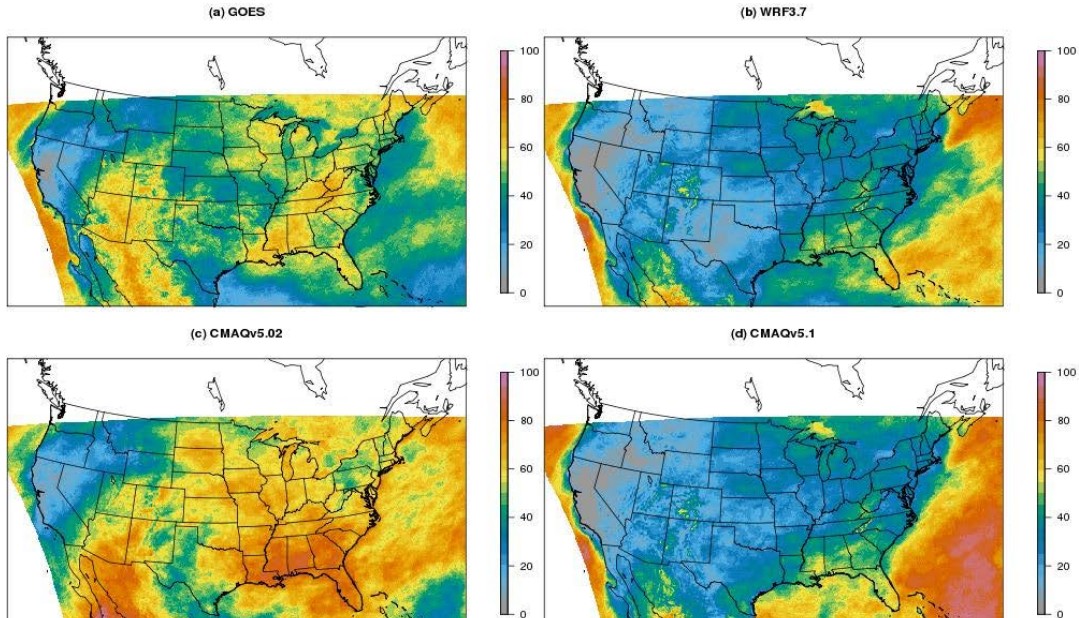

**Figure 1: Percentage of daytime hours in July 2011 that were cloudy based on cloud albedo derived from (a) GOES satellite (b) WRFv3.7 (c) CMAQv5.0.2 photolysis calculations (d) CMAQv5.1 photolysis calculations. Warmer colors indicate grid cells that are cloudy more than 50% of the time during this time period.**





January O₃ diff (CMAQv5.0.2_WRFv3.7 – CMAQv5.0.2_Base)

July O₃ diff (CMAQv5.0.2_WRFv3.7 – CMAQv5.0.2_Base)

January PM₂.₅ diff (CMAQv5.0.2_WRFv3.7 - CMAQv5.0.2_Base)

July PM₂.₅ diff (CMAQv5.0.2_WRFv3.7 – CMAQv5.0.2_Base)

**Figure 2: Monthly average difference in O₃ (ppbv) for a) January and b) July and PM₂.₅ (µgm⁻³) for c) January and d) July between CMAQv5.0.2 using WRFv3.4 (CMAQv5.0.2_Base) and CMAQv5.0.2 using WRFv3.7 (CMAQv5.0.2_WRFv3.7) (CMAQv5.0.2_WRFv3.7 – CMAQv5.0.2_Base). Note that the scales for each plot can vary.**



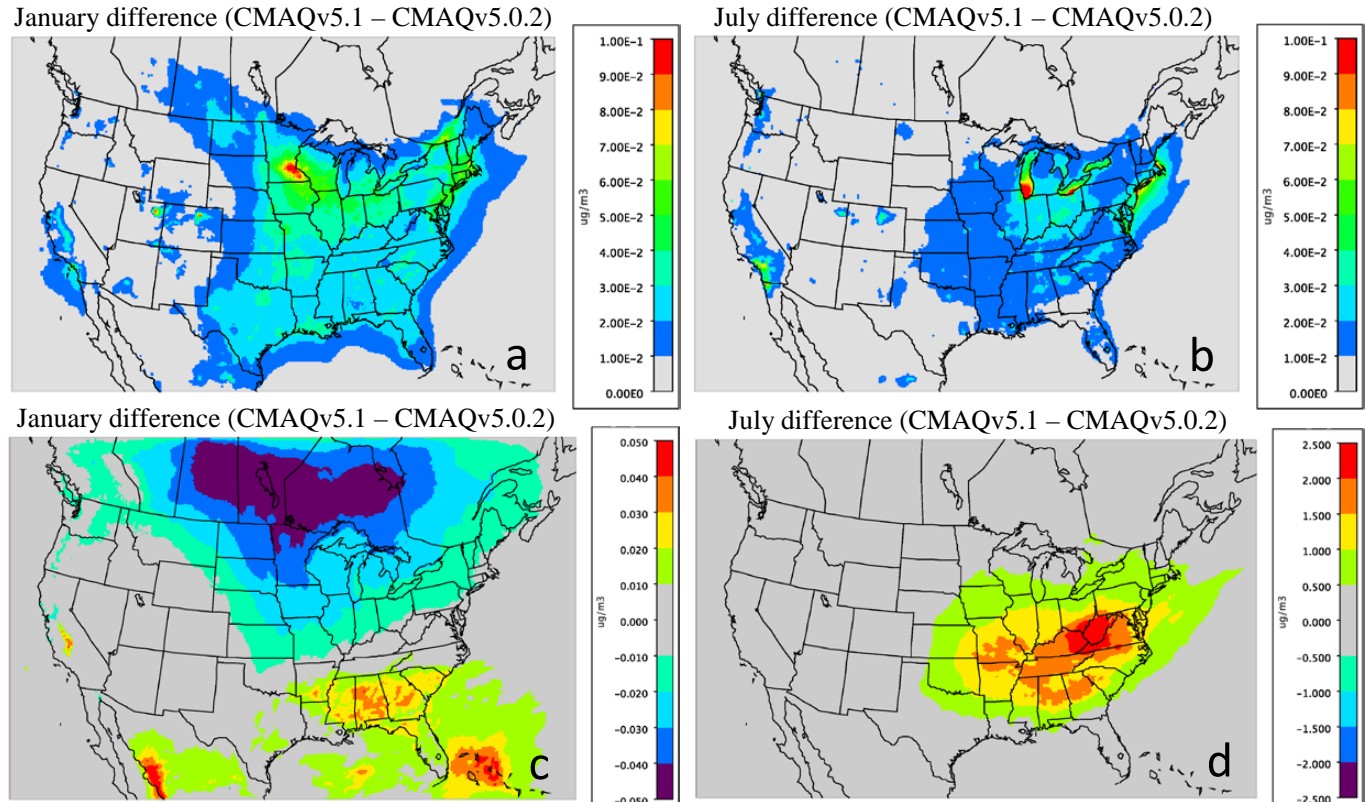

**Figure 3: Monthly average sum total of AALK1, AALK2, APAH1, APAH2 and APAH3 for a) January and b) July (upper right) and the monthly average difference is the sum total of AISO1, AISO2, AISO3 and AOLGB for c) January and d) July between CMAQ v5.0.2 and v5.1 (v5.1 – v5.0.2). All plots are in units of µgm⁻³. Note that the scales for each plot can vary.**





**Figure 4: Difference in the monthly average O₃ for a) January and b) July and PM₂.₅ for c) January and d) July between CMAQ v5.1_base and v5.1_RetroPhot (v5.1_Base – v5.1_RetroPhot). O₃ plots are in units of ppb and PM₂.₅ plots are in units of μgm⁻³. Note that the scales for each plot can vary.**





**Figure 5: Difference in the monthly average O₃ for a) January and b) July and PM₂.₅ (with organic matter mass removed) for c) January and d) July between CMAQ v5.1_Base and v5.1_TUCL (CMAQv5.1_Base – CMAQv5.1_TUCL). O₃ plots are in units of ppb and PM₂.₅ plots are in units of μgm⁻³. Note that the scales for each plot can vary.**





**Figure 6: Difference in the seasonal average PM$_{2.5}$ for a) winter (DJF) b) spring (MAM) c) summer (JJA) and d) fall (SON) between CMAQ v5.0.2_Base and v5.1_Base (CMAQv5.1_Base – CMAQv5.0.2_Base). All plots are in units of μgm$^{-3}$.**





**Figure 7: Seasonal average PM$_{2.5}$ mean bias (µgm$^{-3}$) at IMPROVE (circles), CSN (triangles), AQS Hourly (squares) and AQS Daily (diamonds) sites for a) winter (DJF) b) spring (MAM) c) summer (JJA) and d) fall (SON) for the CMAQ v5.1_Base simulation.**



**Figure 8: Difference in the absolute value of seasonal average PM$_{2.5}$ mean bias for a) winter (DJF) b) spring (MAM) c) summer (JJA) and d) fall (SON) between CMAQ v5.0.2_Base and v5.1_Base (CMAQv5.1_Base – CMAQv5.0.2_Base). All plots are in units of μgm$^{-3}$. Cool colors indicate a reduction in PM$_{2.5}$ mean bias in v5.1 while warm color indicate an increase in PM$_{2.5}$ mean bias v5.1.**





**Figure 9: Regional and seasonal stacked bar plots of PM₂.₅ composition at CSN sites. In order from top to bottom are spring, summer, fall and winter seasons and left to right the Northeast, Great Lakes, Atlantic, South and West regions. The individual PM₂.₅ components (in order from bottom to top) are SO₄²⁻ (yellow), NO₃⁻ (red), NH₄⁺ (orange), EC (black), OC (light gray), Soil (brown), NaCl (green), NCOM (pink), other (white), blank adjustment (dark gray) and H₂O/FRM adjustment (blue).**





**Figure 10:** Difference in the monthly average hourly O₃ (ppbv) for winter (DJF; top left), spring (MAM; top right), summer (JJA; bottom left) and fall (SON; bottom right) between CMAQ v5.0.2_Base and v5.1_Base (CMAQv5.1_Base – CMAQv5.0.2_Base). Note that the scales for each plot can vary.



**Figure 11: Seasonal average hourly O₃ mean bias at AQS sites for a) winter (DJF) b) spring (MAM) c) summer (JJA) and d) fall (SON) for the CMAQ v5.1_Base simulation. All plots are in units of ppbV.**





**Figure 12: Difference in the absolute value of monthly average O₃ mean bias for a) winter (DJF) b) spring (MAM) c) summer (JJA) and d) fall (SON) between CMAQ v5.0.2_Base and v5.1_Base (CMAQv5.1_Base – CMAQv5.0.2_Base). All plots are in units of ppbV. Cool colors indicate a reduction in O₃ mean bias in v5.1 while warm color indicate an increase in O₃ mean bias v5.1.**





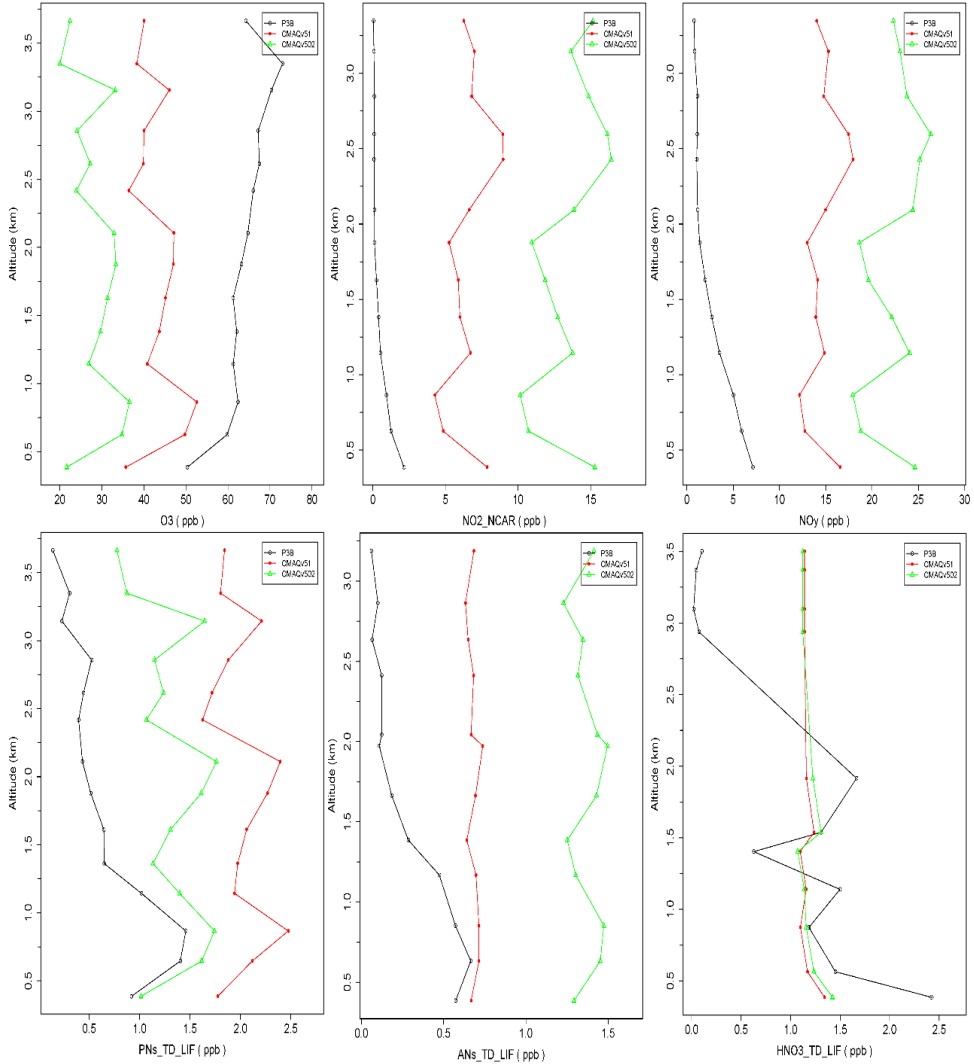

**Figure 13: Observed and CMAQ simulated vertical profiles of O₃ (upper left), NO₂ (upper middle), NO_Y (upper right), peroxy nitrates (PNs; lower left), alkyl nitrates (ANs; lower middle) and HNO₃ (lower right) for the Edgewood site in Baltimore, MD on July 5, 2011. CMAQv502_Base simulation profiles are shown in green and CMAQv51_Base simulation profiles are shown in red. Altitude (km) is given on the y-axis, while mixing ratio (ppbv) is given on the x-axis.**







**Figure 14: Difference in MDA8 O$_3$ RRFs for CMAQv502 and v51 (v502 – v51) for a 50% cut in anthropogenic NO$_x$ (top) and VOC (bottom) for January (left) and July (right) binned by the modeled MDA8 O$_3$ mixing ratio (ppbV). Values greater than one indicate v51 is more responsive than v502 to the emissions cut, while values less than one indicate v502 is more responsive.**





**Figure 15: Box plots of monthly average ratio values (Cut /Base) of PMIJ (total PM₂.₅), ASO4IJ, ANO3IJ, ANH4IJ, AECIJ, ANCOMIJ, AUNSPECIJ, AOMIJ, APOAIJ, AORGAJ, AORGBJ, and AORGCJ for v502 (blue) and v51 (red) for a 50% cut in anthropogenic NOₓ (left), VOC (middle) and SOₓ (right) for January (top) and July (bottom).**