# Peer review of "Description and evaluation of the Community Multiscale Air Quality (CMAQ) modeling system version 5.1"

_Geoscientific Model Development, 2016_

## Referee Comment (RC1) · Anonymous Referee #1 · 7 Oct 2016

This paper describes recent updates to CMAQ, a chemical transport model used for regulatory and research purposes. The topic of the paper is certainly suitable to GMD and will likely be useful to users of CMAQ. However, the manuscript needs to be improved to better communicate the changes in the code and remove apparent contradictions. In particular, I found the discussion for some on the updates to be too general and did not include sufficient citations justifying those updates.

Major Comments:

1) In general, I found the discussion for some on the updates to be too general and did not include sufficient citations justifying those updates as indicated by some of my specific comments below. In addition, the end of Section 1 and Section 2 need to be reordered to have common steps to improve the clarity of the text. I also have concerns

regarding how the two versions of CMAQ are compared given other differences in WRF and the emissions. I understand that there are often complicating factors that make a more fair comparison possible. Some discussion is included to state why those differences in the configuration arise, but those points could have been made more clearly.

2) Section 5.2: I like evaluating the models using profiles from the DISCOVER-AQ data, since observations at the surface only provide a small slice of the atmoshere. However, Section 5.2 seems rather brief and overly simplistic. Extensive measurements were collected during the campaign I presume, yet only one profile is shown. It does illustrate the differences between models, but only for one case. The authors needs to either delete the section, provide a more extensive evaluation, or justify why only one profile is needed. One way to summarize the aircraft data is to showing percentiles of both model and observations as a function of height. In addition, why not use the NASA lidar data to illustrate differences in PM? The authors describe changes in aerosol optical properties but do not evaluate this part even though data is available to evaluate the optical properties.

Specific Comments:

Page 1, line 27: Does "thereby reducing the PM2.5 bias" refer to the previous bias? Please be specific.

Page 1, lines 28-29: The text in these to lines seem to contradict one another in terms of the PM2.5 bias.

Page 1, lines 31-32: Line 31 says v5.1 has a higher O3 bias, but in next line says error is better. I can understand that the correlation could be better in v5.1 even though the bias is worse, but the authors are not clear what they mean here.

Page 1, line 36: What does "significantly" really mean? This is not specific.

Page 2, lines 20-24: Consider making this text a bulleted list.

Page 2, lines 18-29: I was trying to relate the changes described in this paragraph, with Section 2. But upon first reading the paper, it was not clear to me that these two parts were necessarily referring to the same changes. The text could be improved if lines 18-29 were written to be parallel to Section 2, or visa versa.

Page 3, lines 18-24: Please include a reference justifying the revised stomatal conductance. As written, it seems the modification is simply a tuning parameter that improves some quantities in the predictions. There could be easily other changes in the model that could improve the quantities that were mentioned.

Page 3, line 25: Mention values of the heat capacity used in the old and new versions.

Page 3, line 32: So what is the Pr now? The authors say they changed it, but are not specific about this parameter.

Page 3, line 32-36: Similarly, what are the new stability functions and are there some published results to describe them?

Page 4 line 7, This line is confusing. I am missing some details since the phrase "little difference between the initial MOL estimate and the final re-calculated value" is basically saying the code will do nothing. Then what is the point of the code? To me the test associated with MOL assumes the reader is already somewhat familiar with this subject, but I do not understand the logic here.

Page 4, lines 9-16: The text on gravitational settling seems out of place in this section.

Page 4, line 26: There are many studies, not just the two cited, that indicate IVOCs are missing in the emission inventory. Suggest changing text to include "e.g." or "for example" or something similar.

Section 2.2: In general, I found the text in this section to be confusing in terms of what is actually new in v5.1 compared to older versions. The level of detail is rather minimal.

Page 5, line 15: What is "more consistent" mean? Why isn't it completely consistent?

[Figure]

Page 5, lines 20-21: What does "run time options" mean? I assume the authors mean the user has the ability to choose these options. "run time options" sounds like unnecessary jargon.

Page 5, lines 24-25: This sentence does not describe how "cloud fraction, sub-grid cloud fraction, resolved cloud water content" are actually used.

Page 5, line 27: The satellite data can be used to evaluate clouds, but it cannot be used to directly evaluate photolysis calculations. The authors need to be more specific here. I think the authors mean that the clouds indirectly determine where photolysis rates may be high or low, but the satellite does not provide any quantitative estimate of photolysis.

Page 5, line 34: Do you mean photolysis rates at the surface? Please be specific. Surface values will differ from those aloft.

Page 5, lines 37-38: This statement is about the clouds, but c) and d) are about photoloysis rates. I understand the photoloysis rates reflect the cloud distributions, it is just strange the way the sentence is stated. As I said before, the use of "more consistent" leads me to wonder in what ways the clouds in WRF and CMAQ still differ. What are those ways?

Page 6, line 6: How was "most important" determined? More important than what? It seems that the modifications are being added based on recent research activities, but it is not clear why these are more important than other new pathways that may have been reported in the literature. Please explain.

Page 6, line 12: N and Fc need to be defined.

Page 7, lines 6-9: This text is really only saying that some updates have been made, but gives no real specifics on what those updates actually are. How will this help users?

Page 7, line 26: The text mentions overestimates of biogenic VOCs at coastal sites, but this sentence seems to require a reference to know what study pointed that out and

how.

Page 8, Section 3: What is missing from this section is a list of parameterizations used in WRF.

Page 8, line 14: I am not sure why the same version of WRF was not used to drive the two versions of CMAQ. I assume it is to have their older treatments in the land-surface and PBL parameterizations; however, there are likely other changes in the model as well that could cause differences. Please comment, and I think it is worthwhile to reiterate at this point why the two versions of WRF are used.

Page 8, lines 19-39: I am also confused why different emission inventories are used. This will drive differences in the v5.0 and v5.1 simulations that are beyond just the changes to the parameterizations.

Page 10, lines 27-28: While I cannot disagree with these sentence, I think the explanation is rather simplistic. SOA depends on photochemistry and has been shown to be correlated with O3. So if O3 increases, one could expect increases in SOA and therefore an increase in PM2.5.

Page 10, line 39: The authors note that the total concentration of the new SOA species are small. One could conclude here that why were they included in the first place? It would be useful to reiterate that the PAH species are for health reasons and will need to be evaluated in the future. I am less sure about the ALK species. In section 2, the authors not that only the "most important" changes are made, but it is not clear why this is important.

Page 11, lines 1-10: Was the temperature the same between the two versions? Since biogenic emissions are temperature dependent, I am wondering how much difference here is due to meteorological effects versus the changes in chemistry.

Page 11, lines 30-31: Here is a first mention that CMAQ produces more clouds than WRF. The reason for the differences would be useful to describe in Section 2. I still do

not understand why CMAQ would have a different representation of clouds, which can only complicate interpretation of the effects of clouds on chemistry. Later in lines 37-38, they mention differences in sub-cloud treatments. Again this should be stated more upfront in the text. Why is it difficult to have consistent treatment of clouds between the models?

Page 11, line 36: The authors mention "WRF cloud parameterization" but they should specifically state in their model set up which microphysics and cumulus parameterization they used. The way the text is stated, it implies WRF has only one when in fact there are many options. It is not clear that the underprediction in clouds they have could have been fixed or improved using another choice of microphysics or cumulus parameterization.

Page 12, line 19: The second phrase of this sentence is redundant with the first phrase and adds no new information; therefore, it should be deleted. For the same reason, the second phrase in the sentence in lines 20-22 should be deleted.

Page 12, line 28, the authors mention low (I assume lower) PBL heights. So the difference in O3 are driven by the differences in meteorology and it would be useful to quantify this difference in PBL height. If the difference in PBL is on the order of 10's of meters, how does that compare to the vertical grid spacing of the model to actually make a difference?

Figure 5, Perhaps it would be more useful to use percentage changes instead?

Section 4.4: The differences described in this section seem small, so how does this demonstrate a major update of the code? Does this mean the code changes are important theoretically, but they do not make big difference in the predictions.

Page 12, line 32: I am thrown a bit by the phrase "operational performance". "operational" may mean different things to different communities. Section 4 had a comparison of the models, which is repeated here but now include observations. Maybe just say

the performance is evaluated by comparing the models with one another and observations?

Page 13, line 37: Is the change statistically significant?

Page 14, lines 10-11: The authors note an improvement in certain aerosol species, yet emission are different between the simulations. On the next page on line 16, they mention the differences are due to emissions. Why is this then important in terms of the code changes in CMAQ?

Page 15, line 22: Abbreviations are used for states here, but not elsewhere so there is an inconsistent use. I suggest writing out all state names since international readers will not necessarily know what the state abbreviations are.

Page 18, line 32: References to papers in preparation should not be included. Are there other references that can be used?

Page 18, line 33: This section is titled "discussion" but this section contains little new discussion regarding the model results. It reads more like a summary section.
* * *

---

## Referee Comment (RC2) · Anonymous Referee #2 · 17 Oct 2016

Review of the manuscript "Overview and evaluation of the Community Multiscale Air Quality (CMAQ) model version 5.1

This article describe the changes that have been brought to the CMAQ model from version 5.0.2 (released in April 2014 according to the CMAQ website) and version 5.1, released in Dec. 2015, including comparison of model performance between these two model versions, mostly for ozone and PM2.5 over the continental United States. Five different simulations have been performed for the year 2011 (or two months in this year) to evaluate the changes in model performance due to the global and simultaneous upgrade of WRF and CMAQ version as well as of the emission database, but also to separate the contribution of different changes in the models. The authors show in a convincing way that the improvement between model v. 5.0.2 and v5.1 is substantial, even though it is unclear whether this improvement could be due in part or totally to the change of emission datasets. The sensitivity of versions 5.0.2 and version 5.1 of this model to emission reduction scenarios is also evaluated in terms of RRF (relative response factor) for ozone, and of emission cut simulation / base simulation (for PM25).

This article is definitely within the scope of GMD. The improvements in CMAQ that are presented seem substantial even though for some of them the detailed explanation of what has actually been done and why lacks detail. A considerable work of validation has been performed.

Many aspects of the manuscript need to be improved before final publication can be considered. These aspects include the traceability / reproducibility of results (exact description of model), and providing a real and complete model overview. Also, the possibility that the described improvements in model performance are due totally or partly to the use of a new emission dataset must be ruled out.

Less importantly, a clarification of the links between WRF and CMAQ is needed (all these points are developed below in points GC1 to GC6 of my review which I think should definitely be adressed).

This article has the potential to be read and cited by many researchers or operational modellers that use CMAQ for their studies, operational previsions, and/or contribute to its development. It also reflects a huge amount of work by the CMAQ development team. This is why, **on the one hand, I recommend that this study shall be published in GMD, but I think that some important aspects of the paper definitely need to be improved before publication in GMD, including the detailed description of the model parameterizations that are changed (traceability) and a complete overview of CMAQ v5.1.**

The article is clearly written and the language level is good as far as I can say.

**General comments**

**GC1 Sensitivity to emissions**

It is appreciable that sensitivity simulations are performed to evaluate the impact of the various changes between the v5.0.2 setup with WRF 3.4 and NEIv1 emissions and the v5.1 setup with WRF 3.7 and NEIv2 emissions.

However, I have the feeling that, if the idea is to test the sensitivity of the results to the various improvements performed, then a simulation with the v5.1 setup but with NEIv1 emission dataset should be provided. In the present version, all simulations with v5.1 are performed with NEIv2 emissions and all simulations with v5.0.2 are performed with NEIv1 emissions. So the improvement between both model versions, which is shown in a convincing way, could after all be due in part or totally to the improved emission dataset. A sensitivity experiment to emissions is in my opinion needed to rule this hypothesis out and show in a direct way that the improved results are really due to the improvements in WRF and CMAQ and not to better emission input datasets.

*I feel that a convincing answer to this caveat needs to be brought before publication. Otherwise, the reader has no proof at all that the described improvements are not just an effect of changing the input emission dataset*

**GC2 : reproducibility/traceability, precise description of parameterizations**
In some occasions, statements are done that dome parameterizations have been "improved", or "changed", but failing to described exactly what has been changed in which way, questioning the reproducibility of the results (see below comments C7, C10, C14). Details, references and, if necessary equations need to be brought so that developers of other models are able to test similar changes in their models.
Point 6 in the GMD review criteria states : *"In the case of model description papers, it should in theory be possible for an independent scientist to construct a model that, while not necessarily numerically identical, will produce scientifically equivalent results. Model development papers should be similarly reproducible. For MIP and benchmarking papers, it should be possible for the protocol to be precisely reproduced for an independent model. Descriptions of numerical advances should be precisely reproducible".*
-> *The present manuscript clearly fails to meet this criteria. This is not a reason for rejection because the authors will easily be able to correct this caveat in the review process, but this should definitely be done before final publication.*

**GC3 : Need for a real model overview**
The title of the paper is "Overview and evaluation of the Community multiscale Air Quality (CMAQ) model version 5.1" but in my opinion the paper clearly lacks an overview of the model. I think that a Section "Overview of CMAQ 5.1" or similar should be introduced between the Introduction and the section about the scientific improvements.

This section should include at least the basic information one would expect to find about a chemistry-transport model: since when has this model been developed ? What kind of grid does it use ? With what type of transport scheme (horizontal and vertical transport), is the transport Eulerian, Lagrangian, mixed ? What is the recommended range of use in terms of resolution, domain size, regions of use, vertical extension ? What are the inputs that are needed and the variables that are provided as an output ? Are the aerosols treated in a sectional or modal way, and which physico-chemical processes are included regarding the aerosols ? For example, we read that gravitational sedimentation is now included but we do not know if and how processes such as evaporation, coagulation, dry and wet deposition etc. are treated.

*Maybe the authors consider that the focus of this article shall be uniquely the increment from version 5.0.2 to version 5.1 instead of a real model overview, but in this case the title would have to be changed accordingly (and the interest of the paper would be greatly lessened in my opinion, questioning the*

*interest of the publication). In the present state, the paper does not reflect the title, because it does not include an overview of v5.1, only a set of sensitivity studies between v5.0.2 and v5.1.*

**GC4 Need for a general presentation of model outputs**

I feel that the reader lacks a spatialized vision of the model outputs and their characteristics compared to the known features of atmospheric composition over north America. It would be very helpful to provide a map of simulated ozone for the months of January and July, as well as simulated NOx, PM2.5 for these months with v5.1, possibly superposed with the measured average where station data is available, or any other way to give a spatialized vision of model outputs in comparison with state-of-the-art knowledge of the atmospheric composition over the continental US.

**GC5 Better description of modelling setup**
The modelling setup (Section 3) should be described more carefully. Very little space is dedicated to describing the CMAQ configuration for the main simulation, and this section mostly describes the changes between NEI emissions v1 and v2. I think that some information is clearly missing: what kind of initial and boundary conditions are used for the main species, what advection schemes, and the main user options that have been chosen for the simulations. This is particularly the case as CMAQ is a very modular model in which many choices are left to the user, as stated on other CMAQ-related documents.

The same applies to the WRF configuration, particularly as WRF seems very intricated with CMAQ (it seems that some of the updates need to be performed simultaneously in WRF and CMAQ). WRF also has many configuration options, and some of them are very critical for chemistry-transport modelling, such as for example the PBL scheme, but also the convection parameterization (if used), the eventual damping options to avoid instability over steep terrain (which is important since the continental US include major montain rages), etc. Some information is given later in the text but should be given in Section 3 as well. It should also be mentioned what initial and boundary conditions have been used for WRF, and if some nudging has been applied.

It should also be mentioned explicitly if a spinup period has been performed (and discarded) for each simulation. This is particularly the case for the july and january simulations, which last only one month.

**GC6: Precisions about the WRF model and its links with CMAQ**

In many parts of the CMAQ and WRF seem very intricated (as soon as the abstract, which states "Version 5.1 of the CMAQ model was realeased to the public which incorporates a large number of science updates (...) These updates include improvements in the meteorological calculations **in both CMAQ and WRF**". Also on p. 13, l. 8-12 (as well as in the conclusion), we find a sentence that tends to indicate that WRF-CMAQ would even have to be considered as a single "WRF-CMAQ modeling system", that "therefore should be evaluated together". This raises some questions :
* Can CMAQ be used with other meteorological models than WRF (such as reanalyses or ouputs from national meteorological centers) ? Is it recommended by the CMAQ developers ?
* If these models are so intricated together, would it not be relevant to change the title to include this concept of "WRF-CMAQ modeling system" ?

**Minor general comments**
- Some parts of the article are not friendly for a non-US reader. For example, the 2-letter codes for US states are not well-known to the international public. Either the authors should provide a map of these codes, also including the five zones defined p. 14, l. 6-8, or give the complete names of the states.
- Many long URLs are given between parenthesis in the text (e.g. p. 6, l. 27-28). They should probably be given as footnotes.

**Specific comments :**

**Section 2**

C1 : p. 5, l. 31 : Is this time interval valid for all the domain ? Days in July should last much longer than 12 hours at least in the north of the domain, and the daytime interval must be very different from the west to the east of the simulation domain (about 5000 km, which is about 4 hours time lag in the solar time). Using points from 11:45 to 23:45 UTC from west to east would result to using data points from mid-morning to the sunset at the eastern part of the domain, and from dawn to mid-afternoon in the west of the domain, which is critical as cloudiness often has a strong diurnal cycle.
I recommend that all the available daytime data points shall be used for this comparison.

C2 : p. 5, l. 32 : the description of Fig. 1 does not fit that in the caption of Fig. 1 (the latter one seems to be more relevant). The average cloud albedo seems not to be shown.
This should be clarified.

C3 p. 5, l. 36-371 : The authors should explain why it is needed to have a convective cloud model within CMAQ and not use cloud fractions and water content provided by WRF, particularly if CMAQ and WRF almost form a single modeling system as stated elsewhere.

C4 Fig. 1 : The methodology to produce these maps should be precised. Is a threshold placed on cloud albedo to decide tat a particular hour is or is not cloudy ? Is this threshold the same in the models and for the GOES data ?

C5 p. 5, l. 37-38 : the statement that the model cloudiness is "more consistent with the WRF parameterization" is possibly correct but not shown by the figure, since WRF3.4 cloudiness is not provided for comparison with CMAQ v5.0.2 cloudiness. Also, it is hardly a surprise, since CMAQ v5.1 clouds are produced from WRF 3.7 cloud data it would be alarming if the results are very different.

Actually, this paragraph seems to me a bit titled towards suggesting that CMAQv5.1 cloudiness is better that CMAQ v5.0.2, but the figure does not allow to make such a statement, since CMAQ 5.1 cloudiness is compared to the cloudiness of WRF 3.7, which is almost the same data, while CMAQ v5.0.2 is compared to the actual satellite data, which is more of a challenge. Actually, in my eyes, visual comparison between Figs 3c and 3d to Fig. 3a shows that, above most of the continental US and the surrounding oceans (except maybe the center-north of the US), CMAQ 5.0.2 cloudiness is in much better agreement than CMAQ 5.1 with the observed cloudiness.

*The authors invite to compare Fig. 3c to 3a and 3d to 3b, but I recommend that they also invite the reader to compare Fig. 3d to 3a and explicitly comment the comparison between CMAQ v 5.1*

*cloudiness to the Goes data and comment why the agreement does not seem as good as with CMAQ v5.0.2 over many areas.*

Also, it would be interesting to have the same 4 figures shown and commented for the month of January (eventually as a supplement).

C6 p. 6, l. 11-12 : "the rate constant (...) low-pressure limit" : please clarify, and define what are N and Fc in the subsequent parenthesis

C7 p. 7, l. 11-19
The modification to the sea-salt emission schemes is described in a rather vague way. I recommend that the following information is added :
- Give the equations that control the sea-salt emission processes in CMAQ in open ocean and in the surf-zone and the reference from which they were inferred. It would also be helpful to the reader to also show the size distribution of sea-salts in the former and in the new version ("the size distribution (...) was expanded to better reflect ..." seems a bit to vague to me, not permitting reproducibility.

C8 subsection 2.1, p. 3, seems to adress issues about vertical mixing and air-surface exchanges rather than explicit transport. I think it would fit better in the 2.5 subsection.

C9 p. 7, l. 23 : Niinements to Niinemets ?

C10 p. 7, l. 23-24 : "A leaf temperature algorithm was implemented that replaced the 2m-temperature..."
I think there is not enaugh detail to guarantee reproducibility, or the possible use of this algorithm by other modellers. I recommend that the idea of this algorithm and its equations are given, and if possible that the interested reader shall be oriented to a publication decsribing this algorithm.

C11 p. 7, l. 24-25 : it is unclear to me how 2m-temperature can be consistent with emission factor measurements (2m temperature should be consistent with 2m-temperature measurements...). Please reformulate or clarify this sentence.

C11 p. 7, 25 : please define BELD

**Section 3**

**C12 : p. 8, l. 5 :** it is interesting that the model is run up to 50 hPa, well into the stratosphere. I think it would be interesting to have a glimpse of the model outputs in the stratosphere (or throughout the whole atmospheric column) and their validity, particularly regarding ozone. For upper troposphere/lower stratosphere, comparison with either satellite sata or aircraft data such as MOZAIC (http://www.iagos.fr/web/) would, I think bring something useful to this study. Also, it would be interesting to know whether additional reactions are needed in CMAQ to take into account the lower stratospheric chemistry, which is different from tropospheric chemistry.

**C13 : Table 2** . This table is useful but not easy to read. I suggest that it is converted into a table with several columns :

| Simulation name | CMAQ version | WRF version | NEI version | Photolysis scheme | Chemical scheme | Simulation period |
|---|---|---|---|---|---|---|
| CMAQ_5.0.2_Ba | | | | | | |

| se | | | | | | |
|---|---|---|---|---|---|---|
| .... | | | | | | |

**C14 : p. 9, l. 4** : there are many ways to simulate plume-rise. The scheme that is being used and the underlying data (chimney height, flow speed and temperature if relevant etc.) should be described, or the reader should be referred to a previous publication describing the plume-rise strategy in CMAQ, to permit reproducibility.

**Section 4**

**C15 : p. 9, l. 24-30 :** I think the effect of the changes in the treatment of clouds should also be considered at that point. In July, the increase in O3 between both versions of WRF is strong in the SW United States (from Louisiana to Virginia). If one goes back to Fig. 1, we can see that this area was represented as cloudy by the CMAQv5.0.2 version with WRF 3.4, but almost cloud-free by WRF 3.7. It appears to me that this reduction of cloudiness between WRF 3.4 and WRF 3.7 is also a very plausible explication for O3 increase in summertime over this area, particlarly as the same is observed in western Mexico and the states of Arizona, Colorado, Utah and New-Mexico.

**Section 5**

**C15 : p.13, l. 1-5 :** I think the word "stochastic" is not appropriate. I very much prefer "subgrid variations", which is also used. It is very much a modeller vision to consider that everything below grid resolution is essentially stochastic, by the fact that a station close to a highway measures higher contamination levels than a station 3 miles away in a forest is perfectly deterministic.

**C16, p. 13, l. 6-20 :** This part describes the increments between both model versions, and would probably be more at its place in Section 4 than in Section 5. section 5 is about the validation of v5.1 so it is confusing that v.5.0.2 is mentioned so much at this point.

**C17, p. 13, l. 15-20 :** This is a significant caveat that should be mentioned in the model description much earlier than that. The fact that windblown dust treatment was not available for the article simulations and neither for teh public release is not anecdotic in my opinion. Also, it would be appreciated that the statement that dust contributions are "small and episodic" is made more quantitative, for example by providing a map of average dust concentrations (and variability) in v5.0.2.

**C18, Section 5.1 :**
This section essentially describes the differences between v5.1 and v5.0.2 . While this is pertinent for the study, it does not seem to fit within Section 5, which is about evaluation of v5.1 . Only the parts referring to Fig. 9 and to Figs. S2-S5, such as p. 14, l. 3-14, l. 26,27, etc. partly treat of the evaluation of v.5.1 . In my opinion, the parts describing differences between v5.1 and v5.0.2 should be moved to Section 4

**C19 :** figs S2 to S5 should be moved into the main manuscript (because **they do present material permitting an objective evaluation towards the observations in absolute terms, which is lacking in most of the manuscript**). These figures (S2 to S5) also show a rather spectacular improvement from v5.0.2 to v5.1, except maybe for springtime. I think it would be fair that the authors insist more on this very strong improvement of their model results.

There are some formatting problems in these Figs. S2-S5 : the simulation name in the caption do not fit exactly the ones given in Tab. 2 , legend of the vertical axis of the top middle panels fail to state that it is the bias which is plotted.

If it takes too much space to bring all figures S2-S5 into the manuscript, the authors should maybe consider providing a table with the average observed and modelled values for PM25 as well as the RMSE and correlations for both model versions, and for the four seasons.

**C20, Section 5.2 :**
I would make essentially the same general recommandations than for Section 5.1 : that the parts treating of increments from v5.0.2 to v5.1 be moved into Section 4, and that more focus is put on figures S7-S14, bringing some of them into the manuscript, and/or providing a table with the most relevant statistical parameters for both ozone and Nox. The title of the section should probably include Nox as well as ozone.

**C21, p. 16, l. 6-12 :** This difference in summertime ozone concentrations over the eastern US is rather significant and in my opinion can be attributed to the change in meteorology between WRF 3.4 to 3.7 (Fig. 2b): the similarity between Fig. 2b and 10c is striking and the numbers and patterns correspond quite well. I think the authors should comment that, and also the fact that the model bias for ozone in summertime over these regions is increased in v5.1, corresponding to the fact that cloud cover is underestimated in these regions in v5.1 (Fig. 1).

Section 5.2 ("comparison to aircraft measurements")

**C22-1:** A general comment about this part is that comparing with a single vertical profile is not enaugh to state an improvement or a deterioration in a model's performance. The analysis of other profiles should be included, either from the same campaign, or from routine MOZAIC measurements, which are abundant above the continental US.

**C22** Note, there is a problem in numbering, because previous section is numbered 5.2 as well.
I find it very interesting to give some comparison with aircraft measurements, even though it would be great to have it at upper altitude as well, either from this measurement campaign, or from the routine MOZAIC (or equivalent) measurements.

**C23** Please provide the coordinates and altitude of "Edgewood, MD", as well as the hour (and duration, if relevant) of the considered flight, because PBL structure and the behaviour of the real and modelled atmosphere depends a lot on the time of day, and if possible more meteorological context (was it a clear-sky, cloudy, rainy day at that place ?). This would help a lot the reader to analyze the figures.

**C24** Fig. 13 : please increase the size of fonts in the panels, it is hard to read in printed version.

**C25** Do the authors have an idea why the Nox (and Noy) values in Fig. 13 are reduced so drastically between both model versions (about 50% for Nox)? Model simulations describe a rather young air mass with Nox/Noy ratio around 60%, while the Nox to Noy ratio in the measurement is about 30%, typical of a much more aged (and clean) air mass, suggesting different trajectories in the model than in reality. Nox level being very dependent on anthropogenic emissions, is it possible that this drastical reduction is due at least in part to the emission update ? These changes between model versions seem more dramatic than the smooth statistical changes that appear in the statistical scores vetween v. 5.0.2

and v. 5.1.

**Section 6**

**C26 p. 17, l. 21 :** these notions are not necessarily familiar to the reader. I think it should be precised that these notions apply to the United States of America, and possibly add a reference that explains what are the SIPs and "Federal rules".

**C27 p. 18, l. 2 :** Text and figure caption of Fig. 14 announce that a ratio (emission cut simulation / base simulation) will be shown, but the panels show that what is shown is a "RRF", a notion which is not defined. If RRF is to be actually used, then it should be defined, and possibly, some clues shall be given about the use of this indicator, which is not known to the entire modelling community.

**C28 Figure 14 :** it is not clear to me what kind of samples populates the box plots. Are the samples made from model grid cells, model time series at given locations ? Also, the sample size for of each bin should be precised (for example, the appearance of the rightmost box plot for the case of January suggests that the sample size may be very small. A bit more methodological precisions for this plot (as well as Fig. 15) would be welcome. I would also suggest that the format of Fig. 15 is applied to Fig. 14 as well, which avoids introducing the RRF, and permits the reader to evaluate the reduction obtained in v5.0.2, the reduction obtained in v5.1, and visualize and evaluate the difference between the responsiveness in both versions. I think Fig. 14 does not allow as to know if the difference in responsiveness between both model versions amounts to 2% or 50% of the expected model response, while Fig. 15 does.

**Section 7**
**C29 p. 18, l. 35 :** I do not agree that the model has been "evaluated in terms of operational performance" since the evaluation has been performed for year 2011, so more like a reanalysis than an operational forecast model. I suggest to replace by "evaluated by comparison of a simulation of year 2011 to routine measurements of ozone, Nox and PM25 from xxx ground stations" (or something equivalent)

**C30 p. 19, l. 10 :** "to decrease the amount of sub-grid in the photolysis calculation" : please clarify, come words seem to be missing here

**C31 p. 19, l. 10-13 :** it also seems that switching from WRF3.4 to WRF3.7 had a strong effect in reducing model cloudiness over the continental US (Fig. 1), in turn increasing summertime ozone levels over the concerned areas (Fig. 2b), even in the absense of update in the photolysis scheme. Therefore, I find this part of the conclusion (from "The net effect..." to "on average") a bit questionable.

**C32 p. 19, l. 13-14 :** if I am not wrong, these options are not really been described  in the main development, neither which one of these options was chosen to obtain the results described here.

**C33 p. 20, l. 2 :** I think the authors should state explicitly which known issues they are referring to (because this may be of interest to model users)

**C34 p. 20, l. 10-11 :** I think the WRF website should be referred to as well since extensive use of WRF has been made and it seems critical that users use CMAQ with a recent WRF version.

---

## Author Response (AR1)

This paper describes recent updates to CMAQ, a chemical transport model used for regulatory and research purposes. The topic of the paper is certainly suitable to GMD and will likely be useful to users of CMAQ. However, the manuscript needs to be im- proved to better communicate the changes in the code and remove apparent contra- dictions. In particular, I found the discussion for some on the updates to be too general and did not include sufficient citations justifying those updates.

Major Comments:

1) In general, I found the discussion for some on the updates to be too general and did not include sufficient citations justifying those updates as indicated by some of my specific comments below. In addition, the end of Section 1 and Section 2 need to be reordered to have common steps to improve the clarity of the text. I also have concerns

regarding how the two versions of CMAQ are compared given other differences in WRF and the emissions. I understand that there are often complicating factors that make a more fair comparison possible. Some discussion is included to state why those differences in the configuration arise, but those points could have been made more clearly.

**Response: Hopefully we've addressed most of this concern by addressing the specific comments below. In general however, we tried to include enough detail so that the reader understood what basic changes were made and why. If the reader wishes to get more detailed specifics of the changes made, they are referred to the technical documentation for the model release.**

2) Section 5.2: I like evaluating the models using profiles from the DISCOVER-AQ data, since observations at the surface only provide a small slice of the atmosphere. However, Section 5.2 seems rather brief and overly simplistic. Extensive measurements were collected during the campaign I presume, yet only one profile is shown. It does illustrate the differences between models, but only for one case. The authors needs to either delete the section, provide a more extensive evaluation, or justify why only one profile is needed. One way to summarize the aircraft data is to showing percentiles of both model and observations as a function of height. In addition, why not use the NASA lidar data to illustrate differences in PM? The authors describe changes in aerosol optical properties but do not evaluate this part even though data is available to evaluate the optical properties.

**Response: Since the objective of this section is to evaluate the change in model performance for NOY, AN and PNs, the section has been retitled to reflect its purpose and not suggest to the reader that this section will be a comprehensive evaluation against aircraft measurements. Several statistical metrics of NOY performance have also now been included in the section to help expand the analysis provided and highlight the greatly improved performance of NOY in CMAQv5.1. While it would be nice to be able to show additional profiles from other days and include measurements from other networks, the point of the section was to simply inform the reader of the large improvement in NOY performance and an example of the change in ANs and PNs mixing ratios that can be expected in the new model. Future evaluations of CMAQ will focus specifically on the DISCOVER-AQ time period and utilize the measurements made to a much greater extent.**

Specific Comments:

Page 1, line 27: Does "thereby reducing the PM2.5 bias" refer to the previous bias? Please be specific.

**Response: Added a statement indicating underestimation of PM2.5 by CMAQ in the summer to clarify what bias is being reduced.**

Page 1, lines 28-29: The text in these to lines seem to contradict one another in terms of the PM2.5 bias.

**Response: Clarified that this refers to the consideration of the effect from all the changes made to the model and not just a single update as previous referred to.**

Page 1, lines 31-32: Line 31 says v5.1 has a higher O3 bias, but in next line says error is better. I can understand that the correlation could be better in v5.1 even though the bias is worse, but the authors are not clear what they mean here.

**Response: Corrected text to read that only the correlation improved and not the error.**

Page 1, line 36: What does "significantly" really mean? This is not specific. Page 2, lines 20-24: Consider making

this text a bulleted list.

**Response: Removed the word significantly since it is subjective. Opted to avoid introducing bulleted text into the manuscript.**

Page 2, lines 18-29: I was trying to relate the changes described in this paragraph, with Section 2. But upon first reading the paper, it was not clear to me that these two parts were necessarily referring to the same changes. The text could be improved if lines 18-29 were written to be parallel to Section 2, or visa versa.

**Response: Reordered this paragraph to make it consistent with the order in which the model updates were presented in section 2.**

Page 3, lines 18-24: Please include a reference justifying the revised stomatal conductance. As written, it seems the modification is simply a tuning parameter that improves some quantities in the predictions. There could be easily other changes in the model that could improve the quantities that were mentioned.

**Response: A reference was added regarding the origin of the stomatal conductance values.**

Page 3, line 25: Mention values of the heat capacity used in the old and new versions.

**Response: These values were added to the text.**

Page 3, line 32: So what is the Pr now? The authors say they changed it, but are not specific about this parameter.

**Response: The Pr is (and was) a function of the eddy diffusivity values of momentum and heat. Previous, these values were the same and there Pr was always equal to unity. That is no longer the case.**

Page 3, line 32-36: Similarly, what are the new stability functions and are there some published results to describe them?

**Response: The stability functions are described in Pleim et al. 2016 which is provided as a reference in this section.**

Page 4 line 7, This line is confusing. I am missing some details since the phrase "little difference between the initial MOL estimate and the final re-calculated value" is basically saying the code will do nothing. Then what is the point of the code? To me the test associated with MOL assumes the reader is already somewhat familiar with this subject, but I do not understand the logic here.

**Response: This statement was removed from the text as it was unnecessary and requires a greater understanding of the use of the MOL value in CMAQ and WRF that described in the text.**

Page 4, lines 9-16: The text on gravitational settling seems out of place in this section.

**Response: This description has been moved to the end of Section 2.5.**

Page 4, line 26: There are many studies, not just the two cited, that indicate IVOCs are missing in the emission inventory. Suggest changing text to include "e.g." or "for example" or something similar.

**Response: Changed the text to indicated the provided references are examples.**

Section 2.2: In general, I found the text in this section to be confusing in terms of what is actually new in v5.1 compared to older versions. The level of detail is rather minimal.

**Response: The goal of this section is to provide an overall understanding of what was updated and why. It is not intended to document in detail every change made to the aerosol code in CMAQ. Text was added to the beginning of the section that points the reader to the CMAQv5.1 technical documentation which includes in detail all the changes made to the model.**

Page 5, line 15: What is "more consistent" mean? Why isn't it completely consistent?

**Response: We have revised the paragraph to better define what is and is not consistent with the meteorological model. The updates in photolysis calculations in CMAQ v5.1 related to clouds were intended to ensure internal consistency between cloud mixing, aqueous chemistry and photolysis. The reason cloud treatment in CMAQ is not currently "completely consistent" with WRF is the way that sub-**

grid convective clouds are handled. The sub-grid convective cloud scheme in CMAQ, which is responsible for convective transport of chemical species, aqueous chemistry, and wet scavenging, is a simple bulk scheme based on the convective cloud model in the Regional Acid Deposition Model (RADM; Chang et al., 1987) but with convective transport based on the Asymmetric Convective Model (Pleim and Chang, 1992). Since the CMAQ cloud scheme uses the convective precipitation rate to diagnose sub-grid mass fluxes, the location and timing of precipitating convective clouds are consistent with WRF. A new convective cloud scheme for CMAQ based on the Kain-Fritsch scheme in WRF is currently being tested to improve consistency across chemical and meteorological components of the system.

Page 5, lines 20-21: What does "run time options" mean? I assume the authors mean the user has the ability to choose these options. "run time options" sounds like unnecessary jargon.

**Response: We removed the term and added text on how a user may use either of the two options.**

Page 5, lines 24-25: This sentence does not describe how "cloud fraction, sub-grid cloud fraction, resolved cloud water content" are actually used.

**Response: These parameters are simply provided in the text as examples of new diagnostic values that are available as output in the new version of the model in case a user wishes to examine them. They are not new variables used in the CMAQ model. We also added text to the paragraph briefly describing the calculation method for photolysis rates and how the clouds contribute to the calculation.**

Page 5, line 27: The satellite data can be used to evaluate clouds, but it cannot be used to directly evaluate photolysis calculations. The authors need to be more specific here. I think the authors mean that the clouds indirectly determine where photolysis rates may be high or low, but the satellite does not provide any quantitative estimate of photolysis.

**Response: We agree and revised the paragraph to state this point. Note that this paragraph was also moved to section 4.3 since the focus is on model evaluation. The revised sentences now state:**

**"Additional diagnostic evaluation of photolysis/cloud model treatment in CMAQ was conducted based on the model predicted cloud albedo at the top of the atmosphere. The predicted cloud albedo from WRF3.7, CMAQv5.0.2 and CMAQv5.1 were evaluated against cloud albedo from NASA's Geostationary Operational Environmental Satellite Imager product (GOES; http://satdas.nsstc.nasa.gov/). This evaluation was used to qualitatively determine if one CMAQ version better considers how clouds affect calculated photolysis rates."**

Page 5, line 34: Do you mean photolysis rates at the surface? Please be specific. Surface values will differ from those aloft.

**Response: The figure shows what the cloud parameterization between version 5.0.2 and 5.1 implies about the cloud albedo or reflectivity at the top of the atmosphere. Changes in photolysis rate are integrated over the vertical column so the paragraph does not discuss photolysis rates at a specific altitude. Also, the revised text attempts to better explain the analysis and displayed results.**

Page 5, lines 37-38: This statement is about the clouds, but c) and d) are about photolysis rates. I understand the photolysis rates reflect the cloud distributions, it is just strange the way the sentence is stated. As I said before, the use of "more consistent" leads me to wonder in what ways the clouds in WRF and CMAQ still differ. What are those ways?

**Response: We revised the paragraph in section 2.3 to better explain how WRF and CMAQ differ in the cloud description, specifically the sub-grid or convective clouds (please see response to previous question for more details). The paragraph describing the figure has been moved to section 4.3. The figure is now referred to as Figure X and the statement in question has been reworded to more accurately describe that what is being plotted is based on the cloud parameterization in the CMAQ system, not just within the photolysis module:**

**"Figure X shows the average cloud albedo or reflectivity at the top of the atmosphere during daytime hours in July 2011 derived from the GOES satellite product (5a), and the cloud parameterizations within: (5b) WRF3.7, (5c) CMAQv5.1_RetroPhot and (5d) CMAQv5.1_Base."**

**(Note that the "CMAQv5.1_RetroPhot" and "CMAQv5.1_Base" abbreviations have been defined at the beginning of section 4.3.)**

Page 6, line 6: How was "most important" determined? More important than what? It seems that the modifications are being added based on recent research activities, but it is not clear why these are more important than other new pathways that may have been reported in the literature. Please explain.

**Response: Since the term "most important" is inherently subjective, we opted to remove that statement since it was not important to the discussion regarding the mechanism updates.**

Page 6, line 12: N and Fc need to be defined.

**Response: The statement referencing these values has been removed since it's not critical at all to discuss these variables and they are described in the citation provided.**

Page 7, lines 6-9: This text is really only saying that some updates have been made, but gives no real specifics on what those updates actually are. How will this help users?

**Response: The specifics of these updates are providing in the technical documentation available through the CMAS website. A link to the documentation is provided in the beginning of the section. Here we're making the reader aware of the changes and why they were made.**

Page 7, line 26: The text mentions overestimates of biogenic VOCs at coastal sites, but this sentence seems to require a reference to know what study pointed that out and how.

**Response: The text has been modified to better explain how the overestimation was determined and addressed, and now also includes a reference.**

Page 8, Section 3: What is missing from this section is a list of parameterizations used in WRF.

**Response: Modified the text to include the specific parameterizations that were employed in the WRF simulations. In addition, now include the WRF namelists in the supplemental material.**

Page 8, line 14: I am not sure why the same version of WRF was not used to drive the two versions of CMAQ. I assume it is to have their older treatments in the land-surface and PBL parameterizations; however, there are likely other changes in the model as well that could cause differences. Please comment, and I think it is worthwhile to reiterate at this point why the two versions of WRF are used.

**Response: The text was modified to explain why different versions of WRF were used. In short, because the updates made in WRFv3.7 were tied to similar updates made in CMAQv5.1, those two version of the models need to be used together (without modifications to the MCIP code). Similarly, WRFv3.4 is tied to CMAQv5.0.2 without additional modifications to the MCIP preprocessor. Hopefully this is now clear to the reader in the main text.**

Page 8, lines 19-39: I am also confused why different emission inventories are used. This will drive differences in the v5.0 and v5.1 simulations that are beyond just the changes to the parameterizations.

**Response: A new emissions platform became available after the v5.0.2 simulations were complete. It was felt that in order to obtain the best model results the latest emissions platform should be used and therefore was used for the v5.1 simulations. Sensitivity tests were performed to assess the impact that the changes in the emissions platform had on the model results and the impacts were determined to be small. A figure showing the impact of the emissions platform change on ozone and PM2.5 in January and July has been added to the text to quantify to the reader the impact from the emissions platform**

**change.**

Page 10, lines 27-28: While I cannot disagree with these sentence, I think the explanation is rather simplistic. SOA depends on photochemistry and has been shown to be correlated with O3. So if O3 increases, one could expect increases in SOA and therefore an increase in PM2.5.

**Response: Added a statement regarding how the change in oxidant concentration could impact the formation of SOA and therefore PM2.5 concentrations.**

Page 10, line 39: The authors note that the total concentration of the new SOA species are small. One could conclude here that why were they included in the first place? It would be useful to reiterate that the PAH species are for health reasons and will need to be evaluated in the future. I am less sure about the ALK species. In section 2, the authors not that only the "most important" changes are made, but it is not clear why this is important.

**Response: Statements were added to indicate that while the overall monthly difference in concentration of these species is small, the episodic isolated concentration can be higher. Also indicated the importance of these species in health related studies.**

Page 11, lines 1-10: Was the temperature the same between the two versions? Since biogenic emissions are temperature dependent, I am wondering how much difference here is due to meteorological effects versus the changes in chemistry.

**Response: The temperature difference between the two simulations is very small due to the use of four-dimensional data assimilation in the WRF simulations and does not affect the biogenic emissions significantly.**

Page 11, lines 30-31: Here is a first mention that CMAQ produces more clouds than WRF. The reason for the differences would be useful to describe in Section 2. I still do not understand why CMAQ would have a different representation of clouds, which can only complicate interpretation of the effects of clouds on chemistry. Later in lines 37- 38, they mention differences in sub-cloud treatments. Again this should be stated more upfront in the text. Why is it difficult to have consistent treatment of clouds between the models?

**Response: Moved this paragraph to Section 2.3 as it seemed more appropriate there. Greater effort has been made in that section to explain why the clouds in WRF differ from the clouds in CMAQ.**

Page 11, line 36: The authors mention "WRF cloud parameterization" but they should specifically state in their model set up which microphysics and cumulus parameterization they used. The way the text is stated, it implies

WRF has only one when in fact there are many options. It is not clear that the underprediction in clouds they have could have been fixed or improved using another choice of microphysics or cumulus parameterization.

**Response: The text now includes which WRF parametrizations were used in the simulations. In addition, the WRF namelists used have been added to the supplemental material.**

Page 12, line 19: The second phrase of this sentence is redundant with the first phrase and adds no new information; therefore, it should be deleted. For the same reason, the second phrase in the sentence in lines 20-22 should be deleted.

**Response: The redundant lines were removed.**

Page 12, line 28, the authors mention low (I assume lower) PBL heights. So the difference in O3 are driven by the differences in meteorology and it would be useful to quantify this difference in PBL height. If the difference in PBL is on the order of 10's of meters, how does that compare to the vertical grid spacing of the model to actually make a difference?

**Response: This was to note the relatively low PBL heights that typically occur over water versus land and as a result the change in ozone precursors can amplify the change in ozone due over water to low PBL heights. This is not a statement of the actual difference in PBL height between the two WRF simulations.**

Figure 5, Perhaps it would be more useful to use percentage changes instead?

**Response: As a compromise, added the approximate percent change in O3 and PM2.5 to the text describing Figure 5.**

Section 4.4: The differences described in this section seem small, so how does this demonstrate a major update of the code? Does this mean the code changes are important theoretically, but they do not make big difference in the predictions.

**Response: Some of the differences are actually quite large for monthly averages. However, the changes made to the atmospheric chemistry are incremental in nature.**

Page 12, line 32: I am thrown a bit by the phrase "operational performance". "operational" may mean different things to different communities. Section 4 had a comparison of the models, which is repeated here but now include observations. Maybe just say the performance is evaluated by comparing the models with one another and observations?

**Response: Removed the word "operational" as this was a source of unnecessary confusion without the**

**appropriate context.**

Page 13, line 37: Is the change statistically significant?

**Response: Statistical significance doesn't apply in this case since we're comparing two model simulations against the exact same set of observations. So, any change is by default statistically significant. However, it would remain in the hands of the reader to determine whether the change is significant to their application.**

Page 14, lines 10-11: The authors note an improvement in certain aerosol species, yet emission are different between the simulations. On the next page on line 16, they mention the differences are due to emissions. Why is this then important in terms of the code changes in CMAQ?

**Response: I don't actually see where in line 16 it is mentioned that the differences are due to emissions. It is stated that the difference are primarily the result in differences in the concentration of primary emitted species, but that it likely the result of changes in the meteorology (lower or higher PBL heights) and changes in the emissions.**

Page 15, line 22: Abbreviations are used for states here, but not elsewhere so there is an inconsistent use. I suggest writing out all state names since international readers will not necessarily know what the state abbreviations are.

**Response: Abbreviations are no longer used for the state names.**

Page 18, line 32: References to papers in preparation should not be included. Are there other references that can be used?

**Response: This paper is actually referenceable as it is online and includes a doi. The text has been changed to no longer indicate the paper as in-preparation.**

Page 18, line 33: This section is titled "discussion" but this section contains little new discussion regarding the model results. It reads more like a summary section.

**Response: The title of this section has been changed from "Discussion" to "Summary" as it does constitute a summary of the work and not a new discussion.**

Review of the manuscript "Overview and evaluation of the Community Multiscale Air Quality (CMAQ) model version 5.1

This article describes the changes that have been brought to the CMAQ model from version 5.0.2 (released in April 2014 according to the CMAQ website) and version 5.1, released in Dec. 2015, including comparison of model performance between these two model versions, mostly for ozone and PM2.5 over the continental United States. Five different simulations have been performed for the year 2011 (or two months in this year) to evaluate the changes in model performance due to the global and simultaneous upgrade of WRF and CMAQ version as well as of the emission database, but also to separate the contribution of different changes in the models. The authors show in a convincing way that the improvement between model v. 5.0.2 and v5.1 is substantial, even though it is unclear whether this improvement could be due in part or totally to the change of emission datasets. The sensitivity of versions 5.0.2 and version 5.1 of this model to emission reduction scenarios is also evaluated in terms of RRF (relative response factor) for ozone, and of emission cut simulation / base simulation (for PM25).

This article is definitely within the scope of GMD. The improvements in CMAQ that are presented seem substantial even though for some of them the detailed explanation of what has actually been done and why lacks detail. A considerable work of validation has been performed.

Many aspects of the manuscript need to be improved before final publication can be considered. These aspects include the traceability / reproducibility of results (exact description of model), and providing a real and complete model overview. Also, the possibility that the described improvements in model performance are due totally or partly to the use of a new emission dataset must be ruled out.

Less importantly, a clarification of the links between WRF and CMAQ is needed (all these points are developed below in points GC1 to GC6 of my review which I think should definitely be addressed).

This article has the potential to be read and cited by many researchers or operational modellers that use CMAQ for their studies, operational previsions, and/or contribute to its development. It also reflects a huge amount of work by the CMAQ development team. This is why, **on the one hand, I recommend that this study shall be published in GMD, but I think that some important aspects of the paper definitely need to be improved before publication in GMD, including the detailed description of the model parameterizations that are changed (traceability) and a complete overview of CMAQ v5.1.**

The article is clearly written and the language level is good as far as I can say.

**General comments**

**GC1 Sensitivity to emissions**

It is appreciable that sensitivity simulations are performed to evaluate the impact of the various changes between the v5.0.2 setup with WRF 3.4 and NEIv1 emissions and the v5.1 setup with WRF 3.7 and NEIv2 emissions.

However, I have the feeling that, if the idea is to test the sensitivity of the results to the various improvements performed, then a simulation with the v5.1 setup but with NEIv1 emission dataset should be provided. In the present version, all simulations with v5.1 are performed with NEIv2 emissions and all simulations with v5.0.2 are performed with NEIv1 emissions. So the improvement between both model versions, which is shown in a convincing way, could after all be due in part or totally to the improved emission dataset. A sensitivity experiment to emissions is in my opinion needed to rule this hypothesis out and show in a direct way that the improved results are really due to the improvements in WRF and CMAQ and not to better emission input datasets.

*I feel that a convincing answer to this caveat needs to be brought before publication. Otherwise, the reader has no proof at all that the described improvements are not just an effect of changing the input emission dataset*

**Response: A new emissions platform became available after the v5.0.2 simulations were complete. It was felt that in order to obtain the best model results the latest emissions platform should be used and therefore was used for the v5.1 simulations. Sensitivity tests were performed to assess the impact that the changes in the emissions platform had on the model results and the impacts were determined to be small. Obviously it was not made clear in the manuscript that the overall impact from the emission platform change was small. Hopefully this is now made clear in the text. In addition, a figure showing the impact of the emissions platform change on ozone and PM2.5 in January and July has been added to the text to quantify to the reader the impact from the emissions platform change.**

**GC2: reproducibility/traceability, precise description of parameterizations**
In some occasions, statements are done that dome parameterizations have been "improved", or "changed", but failing to described exactly what has been changed in which way, questioning the reproducibility of the results (see below comments C7, C10, C14). Details, references and, if necessary equations need to be brought so that developers of other models are able to test similar changes in their models.
Point 6 in the GMD review criteria states: "*In the case of model description papers, it should in theory be possible for an independent scientist to construct a model that, while not necessarily numerically identical, will produce scientifically equivalent results. Model development papers should be similarly reproducible. For MIP and benchmarking papers, it should be possible for the protocol to be precisely reproduced for an independent model. Descriptions of numerical advances should be precisely reproducible*".
*-> The present manuscript clearly fails to meet this criteria. This is not a reason for rejection because the authors will easily be able to correct this caveat in the review process, but this should definitely be done before final publication.*

**Response: The details regarding the parameterizations and options employed in both the WRF and CMAQ simulations has been expanded in Section 3. In addition, the namelists for the WRF simulations have been included in the supplemental material. As for the CMAQ model, the model code for all the versions of the model presented here is available for download through the CMAS Center website. The code is open source and freely available. The input data, including the emission and MCIP (WRF) data are all available upon request from the corresponding author. Even the output from any/all the model simulations performed here are available upon request as well.**

**GC3: Need for a real model overview**
The title of the paper is "Overview and evaluation of the Community multiscale Air Quality (CMAQ) model version 5.1" but in my opinion the paper clearly lacks an overview of the model. I think that a Section "Overview of CMAQ 5.1" or similar should be introduced between the Introduction and the

section about the scientific improvements.

This section should include at least the basic information one would expect to find about a chemistry-transport model: since when has this model been developed? What kind of grid does it use? With what type of transport scheme (horizontal and vertical transport), is the transport Eulerian, Lagrangian, mixed? What is the recommended range of use in terms of resolution, domain size, regions of use, vertical extension? What are the inputs that are needed and the variables that are provided as an output? Are the aerosols treated in a sectional or modal way, and which physico-chemical processes are included regarding the aerosols? For example, we read that gravitational sedimentation is now   included but we do not know if and how processes such as evaporation, coagulation, dry and wet  deposition etc. are treated.

*Maybe the authors consider that the focus of this article shall be uniquely the increment from version 5.0.2 to version 5.1 instead of a real model overview, but in this case the title would have to be changed accordingly (and the interest of the paper would be greatly lessened in my opinion, questioning the interest of the publication). In the present state, the paper does not reflect the title, because it does not   include an overview of v5.1, only a set of sensitivity studies between v5.0.2 and v5.1.*

**Response: It's true that this manuscript is intended to provide a description of the changes that were made between CMAQ versions 5.0.2 and 5.1, along with a comparative evaluation of the results from the previous version of the model and the latest version. It is not intended to be a true "overview" of the modeling system. There have been several papers published that present a general overview of the CMAQ model from a physics and functionality perspective (Byun and Schere, 2006 for example, which is referenced in Section 1). The usefulness of presenting results from one model version to another can in some cases be questionable, however in this case it was felt that the scope of the updates made to the modeling system warranted informing the user community of the model updates and providing evaluation results that users of the model can use to determine whether or not that want to use the new model for their applications. As for the title of the manuscript, it was changed to better reflect that this article is a description of the changes in and evaluation of the CMAQv5.1 model, and not an overview of the CMAQ modeling system.**

**GC4 Need for a general presentation of model outputs**
I feel that the reader lacks a spatialized vision of the model outputs and their characteristics compared   to the known features of atmospheric composition over north America. It would be very helpful to   provide a map of simulated ozone for the months of January and July, as well as simulated NOx, PM2.5 for these months with v5.1, possibly superposed with the measured average where station data   is available, or any other way to give a spatialized vision of model outputs in comparison with state-of-   the-art knowledge of the atmospheric composition over the continental US.

**Response: This is a good suggestion. At the risk of inundating the manuscript with too many figures, we opted to provide seasonal plots of PM2.5 and O3 from CMAQv5.1 in the supplemental material. This should suffice to give the reader the reference that the reviewer is suggesting.**

**GC5 Better description of modelling setup**
The modelling setup (Section 3) should be described more carefully. Very little space is dedicated to describing the CMAQ configuration for the main simulation, and this section mostly describes the changes between NEI emissions v1 and v2. I think that some information is clearly missing: what kind of initial and boundary conditions are used for the main species, what advection schemes, and the main user options that have been chosen for the simulations. This is particularly the case as CMAQ is a very modular model in which many choices are left to the user, as stated on other CMAQ-related documents.

The same applies to the WRF configuration, particularly as WRF seems very intricate with CMAQ (it

seems that some of the updates need to be performed simultaneously in WRF and CMAQ). WRF also has many configuration options, and some of them are very critical for chemistry-transport modelling, such as for example the PBL scheme, but also the convection parameterization (if used), the eventual damping options to avoid instability over steep terrain (which is important since the continental US include major mountain rages), etc. Some information is given later in the text but should be given in Section 3 as well. It should also be mentioned what initial and boundary conditions have been used for WRF, and if some nudging has been applied.

It should also be mentioned explicitly if a spinup period has been performed (and discarded) for each simulation. This is particularly the case for the July and January simulations, which last only one month.

**Response: The model setup for both CMAQ and WRF has been expanded to include many of the options employed in both models. The WRF namelists are now provided in the supplemental material as well. Information regarding the spin-up periods used has also been added to Section 3.**

**GC6: Precisions about the WRF model and its links with CMAQ**

In many parts of the CMAQ and WRF seem very intricate (as soon as the abstract, which states "Version 5.1 of the CMAQ model was released to the public which incorporates a large number of science updates (...) These updates include improvements in the meteorological calculations **in both CMAQ and WRF**". Also on p. 13, l. 8-12 (as well as in the conclusion), we find a sentence that tends to indicate that WRF-CMAQ would even have to be considered as a single "WRF-CMAQ modeling system", that "therefore should be evaluated together". This raises some questions:
\*   Can CMAQ be used with other meteorological models than WRF (such as reanalysis or outputs from national meteorological centers)? Is it recommended by the CMAQ developers?
\*   If these models are so intricate together, would it not be relevant to change the title to include this concept of "WRF-CMAQ modeling system"?

**Response: This is good point to be made. While CMAQ can be used with other meteorological models (MM5 for example), it does have strong ties to the WRF model through consistent mixing schemes and land-surface treatments. So, while it's not truly correct to call it exclusively the WRF-CMAQ modeling system, what was tested and evaluated here was the WRF-CMAQ modeling system.**

**Minor general comments**
-   Some parts of the article are not friendly for a non-US reader. For example, the 2-letter codes for US states are not well-known to the international public. Either the authors should provide a map of these codes, also including the five zones defined p. 14, l. 6-8, or give the complete names of the states.

**Response: All state names are now spelled out.**

-   Many long URLs are given between parenthesis in the text (e.g. p. 6, l. 27-28). They should probably be given as footnotes.

**Specific comments:**

**Section2**

C1 p. 5, l. 31: Is this time interval valid for all the domain? Days in July should last much longer  than

12 hours at least in the north of the domain, and the daytime interval must be very different from the west to the east of the simulation domain (about 5000 km, which is about 4 hours time lag in the solar time). Using points from 11:45 to 23:45 UTC from west to east would result to using data points from mid-morning to the sunset at the eastern part of the domain, and from dawn to mid-afternoon in the west of the domain, which is critical as cloudiness often has a strong diurnal cycle.
I recommend that all the available daytime data points shall be used for this comparison.

**Response**: **All available data from the satellite product are used in the average in Figure 1a (Note this Figure has been moved and is now referred to as FigureXa). The figure in the Supplemental Information section S1 shows the number of daytime hours (11:45UTC – 23:45UTC) with available GOES cloud albedo data during July 1 – July 31, 2011 for the modeling domain. Regions in the eastern half of the US have a larger number of available satellite observations (on the order of 390hrs) compared to the western coast which has < 340hrs. Since the reference to the time window of 11:45UTC-23:45UTC caused unnecessary confusion we have removed this from the main text. We now point readers to the Supplemental Information for further description of the hours of available satellite data: "The satellite data are available at 15 minutes prior to the top of the hour during daytime hours and were matched to model output at the top of the hour (see section S.1 in the supplemental material for further information)."**

C2 p. 5, l. 32: the description of Fig. 1 does not fit that in the caption of Fig. 1 (the latter one seems to be more relevant). The average cloud albedo seems not to be shown.
This should be clarified.

**Response: The reviewer is correct. The wrong Figure 1 was included with the original submission of the manuscript. In the revised version of the manuscript this Figure (now called Figure 4) does show the average cloud albedo, consistent with the description in the text. The Figure caption has also been changed to say:**
**"Figure 4. The average cloud albedo during daytime hours in July 2011 derived from (a) the GOES satellite product (b) WRF3.7 (c) CMAQv5.1 with photolysis/cloud model treatment from v5.0.2 and WRF3.7 inputs (CMAQv5.1_RetroPhot) (d) CMAQv5.1 using WRF3.7 inputs (CMAQv5.1_Base)."**

C3 p. 5, l. 36-371: The authors should explain why it is needed to have a convective cloud model within CMAQ and not use cloud fractions and water content provided by WRF, particularly if CMAQ and WRF almost form a single modeling system as stated elsewhere.

**Response: We revised the preceding paragraph to explain why CMAQ cannot not use convective cloud predicted by WRF. The updates in photolysis calculations CMAQ v5.2 related to clouds were intended to ensure internal consistency between cloud mixing, aqueous chemistry and photolysis. The reason cloud treatment in CMAQ is not currently "completely consistent" with WRF is the way that sub-grid convective clouds are handled. The sub-grid convective cloud scheme in CMAQ, which is responsible for convective transport of chemical species, aqueous chemistry, and wet scavenging, is a simple bulk scheme based on the convective cloud model in the Regional Acid Deposition Model (RADM; Chang et al., 1987) but with convective transport based on the Asymmetric Convective Model (Pleim and Chang, 1992). Since the CMAQ cloud scheme uses the convective precipitation rate to diagnose sub-grid mass fluxes, the location and timing of precipitating convective clouds are consistent with WRF. A new convective cloud scheme for CMAQ based on the Kain-Fritsch scheme in WRF is currently being tested to improve consistency across the chemical and meteorological components of the system. This future model update will allow sub-grid cloud fraction and water content information from WRF to be used within the sub-**

**grid cloud-related processes within the CMAQ system.**

C4 Fig. 1: The methodology to produce these maps should be precised. Is a threshold placed on cloud albedo to decide that a particular hour is or is not cloudy? Is this threshold the same in the models and for the GOES data?

**Response: The wrong Figure 1 was included with the original submission of the manuscript.  In the revised version of the manuscript this Figure (now referred to as Figure 4) shows the average cloud albedo (consistent with the description in the text).   There is no threshold used or needed with the new figure.**

C5 p. 5, l. 37-38: the statement that the model cloudiness is "more consistent with the WRF parameterization" is possibly correct but not shown by the figure, since WRF3.4 cloudiness is not provided for comparison with CMAQ v5.0.2 cloudiness. Also, it is hardly a surprise, since CMAQ v5.1 clouds are produced from WRF 3.7 cloud data it would be alarming if the results are very different.

Actually, this paragraph seems to me a bit titled towards suggesting that CMAQv5.1 cloudiness is better that CMAQ v5.0.2, but the figure does not allow to make such a statement, since CMAQ 5.1 cloudiness is compared to the cloudiness of WRF 3.7, which is almost the same data, while CMAQ v5.0.2 is compared to the actual satellite data, which is more of a challenge. Actually, in my eyes, visual comparison between Figs 3c and 3d to Fig. 3a shows that, above most of the continental US and the surrounding oceans (except maybe the center-north of the US), CMAQ 5.0.2 cloudiness is in much better agreement than CMAQ 5.1 with the observed cloudiness.

**Response: We revised the paragraph in section 2.3 to better explain how WRF and CMAQ differ in the cloud description, specifically the sub-grid or convective clouds.  The motivation for the updates in the cloud treatment in CMAQv5.1 is the improved consistency with the WRF parameterizations and this Figure is intended to emphasize this improvement.  (Note that the figure in question has been moved to section 4.3 and is now referred to as Figure 4).  The cloud albedo in Figure 4c that represents the cloud treatment from CMAQv5.0.2 was based on inputs from WRF3.7, not WRF3.4.  This is now made more explicit in the text and the Figure caption to avoid any confusion.**

*The authors invite to compare Fig. 3c to 3a and 3d to 3b, but I recommend that they also invite the reader to compare Fig. 3d to 3a and explicitly comment the comparison between CMAQ v 5.1 cloudiness to the Goes data and comment why the agreement does not seem as good as with CMAQ v5.0.2 over many areas.*

**Response: The readers are provided information on the differences between the CMAQv5.1 cloudiness and the GOES data in the following paragraph in section 4.3:**
**"Two notable issues remain with the v5.1 modeled cloud parametrization.  The photolysis cloud parameterization in v5.1 produces more clouds over water compared to the WRF parameterization, which is itself biased high for some parts of the Atlantic Ocean compared to GOES.  This issue will be addressed by science updates planned for the CMAQ system and evaluation results are expected to improve in the next CMAQ release (See Section 7.4).  A more significant issue, from an air quality perspective, is the under-prediction of clouds over much of the Eastern and West Central US in the WRF predicted clouds, which is now directly passed along to CMAQ.  This misclassification of modeled clear sky conditions can contribute to an over prediction of O3 in these regions.  Resolving this issue will require changes to the WRF cloud parameterization.  Future research will also include changing the sub-grid cloud treatment currently used in the CMAQ system to be consistent with the sub-grid parameterization used in WRF.  Section S.1 in the supplemental material provides a table with additional evaluation metrics**

**of the modeled clouds over oceans versus over land and also describes how cloud albedo was calculated for the three model simulations."**

Also, it would be interesting to have the same 4 figures shown and commented for the month of January (eventually as a supplement).

**Response: We did attempt to make this figure for January but found that the satellite retrieval method for cloud albedo gave unreliable diagnostic information for locations with snow cover.**

C6 p. 6, l. 11-12: "the rate constant (...) low-pressure limit": please clarify, and define what are N and Fc in the subsequent parenthesis.

**Response: We opted to remove the statement containing the values for N and Fc, as they are the same as those reported by Bridier et al. (1991) which is referenced in the section. Including the updated values seemed unnecessary given the that they are available in both the cited paper and the CMAQv5.1 technical documentation, and would actually be quite laborious to define here.**

C7 p. 7, l. 11-19
The modification to the sea-salt emission schemes is described in a rather vague way. I recommend that the following information is added:
- Give the equations that control the sea-salt emission processes in CMAQ in open ocean and in the surf-zone and the reference from which they were inferred. It would also be helpful to the reader to also show the size distribution of sea-salts in the former and in the new version ("the size distribution (...) was expanded to better reflect ..." seems a bit to vague to me, not permitting reproducibility.

**Response: Numerous references were included in this section that contain the information requested above. In addition, the CMAQv5.1 technical documentation also contains detailed information regarding this update to the model. It seems excessive to rehash here the specific equations and values implemented in the new model when they are easily obtained through the citations provided. The combination of the citations provided and CMAQ technical documentation should be provide information needed for reproducibility. In addition, the CMAQ code itself containing the updates is available as well.**

C8 subsection 2.1, p. 3, seems to address issues about vertical mixing and air-surface exchanges rather than explicit transport. I think it would fit better in the 2.5 subsection.

**Response: This section has been moved to the end of Section 2.5.**

C9 p. 7, l. 23: Niinements to Niinemets?

**Response: The reference name has been corrected and added to the list of references.**

C10 p. 7, l. 23-24: "A leaf temperature algorithm was implemented that replaced the 2m-temperature...". I think there is not enough detail to guarantee reproducibility, or the possible use of this algorithm by other modellers. I recommend that the idea of this algorithm and its equations are given, and if possible that the interested reader shall be oriented to a publication describing this algorithm.

**Response: A detailed description of the updated algorithm is available in the CMAQv5.1 technical documentation, which is now provided as a link in the text and the reader is now encouraged to reference for additional details.**

C11 p. 7, l. 24-25: it is unclear to me how 2m-temperature can be consistent with emission factor

measurements (2m temperature should be consistent with 2m-temperature measurements...). Please reformulate or clarify this sentence.

**Response: This statement has been expanded to make it more clear what exactly was changed to make the calculation in the CMAQv5.1 more consistent regarding the 2-meter temperature and emission factor in BEIS.**

C11 p. 7, 25: please define BELD

**Response: BELD has already defined in the text as "Biogenic Emission Land-use Data".**

**Section 3**

**C12, p. 8, l. 5:** it is interesting that the model is run up to 50 hPa, well into the stratosphere. I think it would be interesting to have a glimpse of the model outputs in the stratosphere (or throughout the whole atmospheric column) and their validity, particularly regarding ozone. For upper troposphere/lower stratosphere, comparison with either satellite sata or aircraft data such as MOZAIC (http://www.iagos.fr/web/) would, I think bring something useful to this study. Also, it would be interesting to know whether additional reactions are needed in CMAQ to take into account the lower stratospheric chemistry, which is different from tropospheric chemistry.

**Response: It's really beyond of the scope of what this paper is trying to accomplish to get into the details of the treatment of the stratosphere in CMAQ. There are papers however, both published and in development, that discuss stratospheric treatment in CMAQ in terms of hemispheric CMAQ model simulations. Those papers do take advantage of the some of the measurement data referred to by the reviewer. In short however, there are checks within the CMAQ model to deal with stratospheric ozone, utilizing climatological ozone profiles and comparing those against model simulated upper-level ozone values for consistency. If the values are found to be inconsistent, then the user is warned and in some cases the model simulation is stopped.**

**C13, Table 2**. This table is useful but not easy to read. I suggest that it is converted into a table with several columns:

| Simulation name | CMAQ version | WRF version | NEI version | Photolysis scheme | | Chemical scheme | Simulation period |
|---|---|---|---|---|---|---|---|
| CMAQ_5.0.2_Ba | | | | | | | |
| se | | | | | | | |
| .... | | | | | | | |

**Response: The layout of Table 2 has been updated accordingly.**

**C14, p. 9, l. 4**: There are many ways to simulate plume-rise. The scheme that is being used and the underlying data (chimney height, flow speed and temperature if relevant etc.) should be described, or the reader should be referred to a previous publication describing the plume-rise strategy in CMAQ, to permit reproducibility.

**Response: The CMAQ plume rise has been described in previous publications (e.g. Foley et al.) and follows the same implementation as SMOKE. A link was added that provides details on the**

**plume rise calculation used in CMAQ.**

**Section 4**

**C15, p. 9, l. 24-30:** I think the effect of the changes in the treatment of clouds should also be considered at that point. In July, the increase in O3 between both versions of WRF is strong in the SW United States (from Louisiana to Virginia). If one goes back to Fig. 1, we can see that this area was represented as cloudy by the CMAQv5.0.2 version with WRF 3.4, but almost cloud-free by WRF 3.7. It appears to me that this reduction of cloudiness between WRF 3.4 and WRF 3.7 is also a very plausible explication for O3 increase in summertime over this area, particularly as the same is observed in western Mexico and the states of Arizona, Colorado, Utah and New-Mexico.

**Response: Agreed that the effect of the change in clouds between the two versions of the model is an important driver in the difference in ozone between the two simulations. The effect of the change in clouds is specifically addressed in section 4.3 that compares the different cloud treatments used in CMAQv5.0.2 and CMAQv5.1. Inherent in this discussion in the effect that the reduced cloudiness in v5.1 has on the ozone mixing ratios.**

**Section 5**

**C15, p.13, l. 1-5:** I think the word "stochastic" is not appropriate. I very much prefer "subgrid variations", which is also used. It is very much a modeller vision to consider that everything below grid resolution is essentially stochastic, by the fact that a station close to a highway measures higher contamination levels than a station 3 miles away in a forest is perfectly deterministic.

**Response: The term stochastic has been widely used and applied to these subgrid variations, and is the terminology used in the referenced article. We're just being consistent with the terminology used in that article.**

**C16, p. 13, l. 6-20:** This part describes the increments between both model versions, and would probably be more at its place in Section 4 than in Section 5. section 5 is about the validation of v5.1 so it is confusing that v.5.0.2 is mentioned so much at this point.

**Response: Section 4 is intended to only focus on a few of the major updates and the impact those specific updates had on the model performance. Section 5 is intended to show the overall performance of the model for ozone and PM2.5. While it would be possible to only show the results of the v5.1 simulation here, it seems useful to also include the v5.0.2 results alongside the v5.1 results so that the reader can quickly identify how they might expect ozone and PM2.5 to change between a v5.0.2 and v5.1 simulation. For that reason, we opted to keep results from both model simulations in this section. This is a fairly common way to present results of a model update such as this.**

**C17, p. 13, l. 15-20:** This is a significant caveat that should be mentioned in the model description much earlier than that. The fact that windblown dust treatment was not available for the article simulations and neither for the public release is not anecdotic in my opinion. Also, it would be appreciated that the statement that dust contributions are "small and episodic" is made more quantitative, for example by providing a map of average dust concentrations (and variability) in v5.0.2.

**Response: Granted the effect of windblown dust can be large in isolated areas, generally a short amount of time. In the United States, windblown dust concentrations are maximized in the springtime in the desert southwest. We've added quantitative values of the springtime seasonal average concentration of soil from CMAQv5.0.2. Overall the seasonal average concentration is**

**very small compared to the overall PM2.5 seasonal average concentration.**

**C18, Section 5.1:**
This section essentially describes the differences between v5.1 and v5.0.2 . While this is pertinent for the study, it does not seem to fit within Section 5, which is about evaluation of v5.1 . Only the parts referring to Fig. 9 and to Figs. S2-S5, such as p. 14, l. 3-14, l. 26,27, etc. partly treat of the evaluation of v.5.1. In my opinion, the parts describing differences between v5.1 and v5.0.2 should be moved to Section 4.

**Response: This section was intended to serve two purposes. First, it is intended to show the performance of CMAQv5.1 versus observations, which of course is useful to users. Secondly, it is also intended to show the change in performance of ozone and PM2.5 between CMAQv5.0.2 and CMAQv5.1 so that users can assess for themselves whether the change in model performance is significant for their application or not. To accomplish this, we present the evaluation results against observations for both versions of the model. Section 4 was intended to only show the impact of the individual model updates, while Section 5 shows the performance of the entire CMAQv5.0.2 and CMAQv5.1 modeling systems. Hopefully the reviewer can agree that this is an acceptable way to accomplish these goals.**

**C19:** figs S2 to S5 should be moved into the main manuscript (because **they do present material permitting an objective evaluation towards the observations in absolute terms, which is lacking  in most of the manuscript**). These figures (S2 to S5) also show a rather spectacular improvement from v5.0.2 to v5.1, except maybe for springtime. I think it would be fair that the authors insist more on this very strong improvement of their model results. There are some formatting problems in these Figs. S2-S5 : the simulation name in the caption do not fit  exactly the ones given in Tab. 2 , legend of the vertical axis of the top middle panels fail to state that it  is the bias which is plotted. If it takes too much space to bring all figures S2-S5 into the manuscript, the authors should maybe  consider providing a table with the average observed and modelled values for PM25 as well as the RMSE and correlations for both model versions, and for the four seasons.

**Response: Ideally we would like to include all the figures in the main text, as they all present relevant material. However, it would create an extremely lengthy article with all those figures from the supplemental material included. As a comprise, three new figures have been added that include the observed and modeled seasonal diurnal profiles for PM2.5, O3 and NOx. These figures do not include the MB, RMSE and correlation, and the reader is referred to the supplemental figures for that information.**

**C20, Section 5.2:**
I would make essentially the same general recommendations than for Section 5.1: that the parts  treating of increments from v5.0.2 to v5.1 be moved into Section 4, and that more focus is put on  figures S7-S14, bringing some of them into the manuscript, and/or providing a table with the most  relevant statistical parameters for both ozone and Nox. The title of the section should probably include   Nox as well as ozone.

**Response: See responses to C18 and C19.**

**C21, p. 16, l. 6-12:** This difference in summertime ozone concentrations over the eastern US is rather significant and in my opinion can be attributed to the change in meteorology between WRF 3.4 to 3.7 (Fig. 2b): the similarity between Fig. 2b and 10c is striking and the numbers and patterns correspond quite well. I think the authors should comment that, and also the fact that the model bias for ozone in summertime over these regions is increased in v5.1, corresponding to the fact that cloud cover is underestimated in these regions in v5.1 (Fig. 1).

**Response: Added to the text the strong correlation between the overall change in ozone in v5.1 and the change in ozone due to the WRF/CMAQ meteorological updates, along with the increase due to increased photolysis in v5.1.**

Section 5.2 ("comparison to aircraft measurements")

**C22-1:** A general comment about this part is that comparing with a single vertical profile is not enough to state an improvement or a deterioration in a model's performance. The analysis of other profiles should be included, either from the same campaign, or from routine MOZAIC measurements, which are abundant above the continental US.

**Response: The authors agree that titling this section as a comparison to aircraft measurements is a bit misleading even though it does include a comparison to a single day of aircraft measurements. Since the objective of this section was to evaluate the change in model performance for NOY, AN and PNs, the section has been retitled to reflect its purpose. In addition, several statistical metrics of NOY performance have also now been included in the section to help expand the analysis provided and highlight the greatly improved performance of NOY in CMAQv5.1. While it would be nice to be able to show additional profiles from other days and include measurements from other networks, the point of the section was to simply inform the reader of the large improvement in NOY performance and an example of the change in ANs and PNs mixing ratios that can be expected in the new model.**

**C22** Note, there is a problem in numbering, because previous section is numbered 5.2 as well.
I find it very interesting to give some comparison with aircraft measurements, even though it would be great to have it at upper altitude as well, either from this measurement campaign, or from the routine MOZAIC (or equivalent) measurements.

**Response: The section number has been corrected.**

**C23** Please provide the coordinates and altitude of "Edgewood, MD", as well as the hour (and duration, if relevant) of the considered flight, because PBL structure and the behavior of the real and modelled atmosphere depends a lot on the time of day, and if possible more meteorological context (was it a clear-sky, cloudy, rainy day at that place?). This would help a lot the reader to analyze the figures.

**Response: The location and elevation of the Edgewood site have been added to the text. The profiles themselves represent an average of several vertical spirals that took place over Edgewood that day, roughly taking place in the morning, early afternoon and late afternoon. This has now been stated in the text.**

**C24** Fig. 13: please increase the size of fonts in the panels, it is hard to read in printed version.

**Response: The font size has been increased in the figure.**

**C25** Do the authors have an idea why the Nox (and Noy) values in Fig. 13 are reduced so drastically between both model versions (about 50% for Nox)? Model simulations describe a rather young air mass with Nox/Noy ratio around 60%, while the Nox to Noy ratio in the measurement is about 30%, typical of a much more aged (and clean) air mass, suggesting different trajectories in the model than in reality. Nox level being very dependent on anthropogenic emissions, is it possible that this drastic reduction is due at least in part to the emission update? These changes between model versions seem more dramatic than the smooth statistical changes that appear in the statistical scores between v. 5.0.2. and v. 5.1.

**Response: The NOx and NOy mixing ratios decrease due to the changes made to the atmospheric chemistry in CMAQv5.1 (Section 2.4.1), while the NOx mixing ratios would be expected to decrease due to greater photolysis in v5.1 (Section 2.3). On average, the NOy mixing ratios in CMAQv5.1 decreased 21% in July and 13% in January. So, in the plots shown in Figure 13, the NOy mixing ratio decrease of about 30% seems reasonable for an isolated case. The emission platform update had little to no impact on the NOX concentrations, so the emission change does not contribute to the decrease in NOx seen in the figure.**

**Section 6**

**C26 p. 17, l. 21 :** these notions are not necessarily familiar to the reader. I think it should be precised that these notions apply to the United States of America, and possibly add a reference that explains what are the SIPs and "Federal rules".

**Response: Added text to explain that SIPs and Federal rules aim to reduce emissions through regulations in order to meet mandated air quality standards.**

**C27 p. 18, l. 2 :** Text and figure caption of Fig. 14 announce that a ratio (emission cut simulation / base simulation) will be shown, but the panels show that what is shown is a "RRF", a notion which is not defined. If RRF is to be actually used, then it should be defined, and possibly, some clues shall be given about the use of this indicator, which is not known to the entire modelling community.

**Response: Since the value presented is indeed a ratio and not a true RRF calculation, the term RRF has been removed from the plots and caption, and replaced with the explanation that a ratio of concentration has been used.**

**C28 Figure 14:** it is not clear to me what kind of samples populates the box plots. Are the samples made from model grid cells, model time series at given locations? Also, the sample size for of each bin should be precised (for example, the appearance of the rightmost box plot for the case of January suggests that the sample size may be very small. A bit more methodological precisions for this plot (as well as Fig. 15) would be welcome. I would also suggest that the format of Fig. 15 is applied to Fig. 14 as well, which avoids introducing the RRF, and permits the reader to evaluate the reduction obtained in v5.0.2, the reduction obtained in v5.1, and visualize and evaluate the difference between the responsiveness in both versions. I think Fig. 14 does not allow as to know if the difference in responsiveness between both model versions amounts to 2% or 50% of the expected model response, while Fig. 15 does.

**Response: Figure 14 has been updated to remove the term RRF to the correct description as a daily ratio (as is presented in Figure 15). The bins are populated from model grid cells, which is now stated in the caption. In addition, the number of model grid cells in each bin is now included above the x-axis.**

**Section 7**

**C29 p. 18, l. 35:** I do not agree that the model has been "evaluated in terms of operational performance" since the evaluation has been performed for year 2011, so more like a reanalysis than an operational forecast model. I suggest to replace by "evaluated by comparison of a simulation of year 2011 to routine measurements of ozone, Nox and PM25 from xxx ground stations" (or something equivalent)

**Response: Removed the word "operational" as it is a source of confusion and requires additional context. Here we use the term operational to refer to evaluation against observations, not an**

**evaluation of the model in an operational (e.g. forecast) mode. However, it seemed easier to remove the word operational and let the evaluation results speak for themselves.**

**C30 p. 19, l. 10:** "to decrease the amount of sub-grid in the photolysis calculation" : please clarify, come words seem to be missing here.

**Response: Added the word "clouds" after sub-grid.**

**C31 p. 19, l. 10-13:** it also seems that switching from WRF3.4 to WRF3.7 had a strong effect in reducing model cloudiness over the continental US (Fig. 1), in turn increasing summertime ozone levels over the concerned areas (Fig. 2b), even in the absence of update in the photolysis scheme. Therefore, I find this part of the conclusion (from "The net effect..." to "on average") a bit questionable.

**Response: Hopefully with the addition of the word "clouds" from the above comment it's clear that we are referring to the decrease in cloudiness in the model as the driving factor in increasing summertime ozone. It's actually the updates to the cloud treatment in CMAQ (to make them consistent with WRF clouds) and not the transition from WRF3.4 to WRF3.7 that drive the difference in the clouds in CMAQ.**

**C32 p. 19, l. 13-14:** if I am not wrong, these options are not really been described in the main development, neither which one of these options was chosen to obtain the results described here.

**Response: The availability of these options was mentioned briefly in Section 2.3 and is mentioned again briefly here as new option available in CMAQv5.1.**

**C33 p. 20, l. 2:** I think the authors should state explicitly which known issues they are referring to (because this may be of interest to model users)

**Response: Reworded to remove the words "known issues". The known issues being referred to was the windblown dust treatment in v5.1, which is referred to in the next sentence anyway.**

**C34 p. 20, l. 10-11:** I think the WRF website should be referred to as well since extensive use of WRF has been made and it seems critical that users use CMAQ with a recent WRF version.

**Response: The WRF website was added to the Data availability section.**

[revised manuscript text omitted]

Supplemental Material

**Section S.1: Use of satellite measured cloud albedo to evaluate cloud parameterizations in CMAQ and WRF**

Cloud albedo is a measure of the solar radiation that is reflected by a cloud and is one of the Imager products available from NASA's Geostationary Operational Environmental Satellite (GOES). The figure below shows the number of daytime hours (11:45UTC – 23:45UTC) with available GOES cloud albedo data during July 1 – July 31, 2011 for the modeling domain.

[Figure]

Although the WRF and CMAQ systems do not use cloud albedo directly in their cloud parameterizations, this variable provides a useful model diagnostic for identifying areas where the models are over or under-predicting the degree of cloudiness over a region.

WRF cloud albedo is calculated as:

CLDALB _WRF = (SWUPT-SWUPTC)/SWDNT*100%                              (1)

where SWUPT is the upwelling shortwave flux, SWUPTC is the upwelling clear sky shortwave flux and SWDNT is the downwelling shortwave flux. All three fluxes are instantaneous and at the top of the model.

Cloud albedo within the CMAQv5.1 and CMAQv5.0.2 photolysis module is calculated as:

CLDALB_NEW = $\pi$*(REFLECTION – CLR_REFLECTION)                              (2)

where REFLECTION is the shortwave reflection, CLR_REFLECTION is the clear sky shortwave reflection, both are instantaneous and at the top of the model.

Table S1 provides categorical evaluation metrics for daytime hours (11:45UTC – 23:45UTC) in July 1 – July 31, 2011 for model predicted clouds based on the CMAQv5.0.2 photolysis module, the CMAQv5.1 photolysis module and WRFv3.7 cloud parameterization compared to GOES satellite data. Cloud albedo is used to determine the presence of clouds in each grid cell for each hour. A cloud albedo > 5% (modeled or observed) is used as the threshold to indicate cloudy conditions. The Agreement Index (Biazer et al., 2014) is the fraction of correct model predictions (true clear skies OR true cloudy conditions) out of all grid cell/hours with available GOES data. Model over-predictions (3[rd] row) are the number of grid cell/hours where the model predicted cloudy conditions but the GOES product showed clear skies. Model under-predictions indicate the model predicted clear skies when the GOES product show cloudy conditions. Each metric is calculated separately for all hourly data available at land grid cells (N=25,771,567) and all hourly data available at ocean grid cells (N=12,376,594).

The relative frequency of model over-prediction of cloudy conditions over land decreased from 0.22 in CMAQv5.0.2 to 0.07 in CMAQv5.1 and is now consistent with the WRF evaluation over land.  However, CMAQv5.1 over-predicts clouds over the ocean to a greater extent than either CMAQv5.0.2 or WRF3.7 (i.e. the over prediction relative frequency is 0.33 compared to 0.27 or 0.21).  This issue will be addressed by new science updates in the CMAQ system and evaluation results are expected to improve in upcoming CMAQ releases.  WRVv3.7 and CMAQv5.1 under-predict cloudy conditions over land 26% of the time. Resolving this issue will require changes to the WRF cloud parameterization.

Table S1.

| Categorical Metric | CMAQv5.0.2 Photolysis | | CMAQv5.1 Photolysis | | WRFv3.7 | |
|---|---|---|---|---|---|---|
| | Land | Ocean | Land | Ocean | Land | Ocean |
| Agreement Index | 0.66 | 0.61 | 0.67 | 0.58 | 0.67 | 0.66 |
| Over-predict clouds | 0.22 | 0.27 | **0.07** | **0.33** | 0.07 | 0.21 |
| Under-predict clouds | 0.12 | 0.12 | **0.26** | 0.09 | 0.26 | 0.13 |

Reference:
Biazar, A.P., 2014: Cloud Correction and its Impact on Air Quality Simulations, presented at the 94[th] AMS Annual Meeting, Atlanta, GA.

**S.2 CMAQv5.1 CB05e51 AE6 Species Definitions (available with the CMAQv5.1 release code)**

! Updated AOCIJ, AOMIJ, and AORGAJ definition for CMAQv5.1 based on recommendations from Havala Pye (Dec 2014)

! The formulas used in this file implicitly assume that the model-ready emission files were prepared using a GSPRO from the 2002 emissions platform or later, in which POC emissions (hence, the CMAQ species APOCI and APOCJ) represent pure organic carbon without any scaling factor for OM:OC ratios.

! Output variables that begin with 'PM' represent those in which a size cut was applied.  For example, PM25_NA is all sodium that falls below 2.5 um diameter. These 'PM' variables are used for comparisons at IMPROVE and STN sites.

! Output variables beginning with 'A' (aside from AIR_DENS) represent a combination of aerosol species in which no size cut was applied.  For example, ASO4IJ is the sum of i-mode and j-mode sulfate.  These 'A' variables are used for comparisons at CASTNet sites.

! Output variables beginning with 'PMC' is the coarse mode of total PM, i.e., sums all modes then subtracts the fine mode (PM2.5).  These 'PMC' variables are used for comparisons at SEARCH sites.

```
/ File [1]: CMAQ conc/aconc file
/ File [2]: AEROVIS file
/ File [3]: METCRO3D file
/ File [4]: AERODIAM file
/ File [5]: METCRO2D file
/
/new species    ,units    ,expression

! Gases
ALD2        ,ppbV    ,1000.0*ALD2[1]
ALDX        ,ppbV    ,1000.0*ALDX[1]
BENZENE       ,ppbV     ,1000.0*BENZENE[1]
CO         ,ppbV    ,1000.0*CO[1]
ETH        ,ppbV    ,1000.0*ETH[1]
ETHA        ,ppbV    ,1000.0*ETHA[1]
FORM        ,ppbV    ,1000.0*FORM[1]
H2O2        ,ppbV    ,1000.0*H2O2[1]
HNO3        ,ppbV    ,1000.0*HNO3[1]
HNO3_UGM3      ,ug/m3    ,1000.0*(HNO3[1]*2.1756*DENS[3])
HONO        ,ppbV    ,1000.0*HONO[1]
CLNO2        ,ppbV    ,1000.0*CLNO2[1]
HOX        ,ppbV    ,1000.0*(OH[1]+HO2[1])
OH          ,ppbV    ,1000.0*(OH[1])
IOLE        ,ppbV    ,1000.0*IOLE[1]
ISOP        ,ppbV    ,1000.0*ISOP[1]
N2O5        ,ppbV    ,1000.0*N2O5[1]
NH3        ,ppbV    ,1000.0*NH3[1]
NH3_UGM3      ,ug/m3    ,1000.0*(NH3[1]*0.5880*DENS[3])
NHX         ,ug/m3    ,1000.0*(NH3[1]*0.5880*DENS[3])+ANH4I[1]+ANH4J[1]+ANH4K[1]
NO        ,ppbV    ,1000.0*NO[1]
NO2        ,ppbV    ,1000.0*NO2[1]
ANO3_PPB        ,ppbV         ,(ANO3I[1]+ANO3J[1]+ANO3K[1])/(DENS[3]*(62.0/28.97))
NTR        ,ppbV    ,1000.0*(NTROH[1]+NTRALK[1]+NTRCN[1]+NTRCNOH[1]+NTRM[1]+NTRI[1]+NTRPX[1])
PANS        ,ppbV    ,1000.0*(PAN[1]+PANX[1]+OPAN[1]+MAPAN[1])
NOY        ,ppbV
,1000.0*(NO[1]+NO2[1]+NO3[1]+2*N2O5[1]+HONO[1]+HNO3[1]+PNA[1]+CRON[1]+CRNO[1]+CRN2[1]+CRPX[1]+CL
NO2[1])+PANS[0]+NTR[0]+ANO3_PPB[0]
O3        ,ppbV    ,1000.0*O3[1]
OLE        ,ppbV    ,1000.0*OLE[1]
PAR        ,ppbV    ,1000.0*PAR[1]
PAN        ,ppbV    ,1000.0*PAN[1]
PANX        ,ppbV    ,1000.0*PANX[1]
```

SO2            ,ppbV     ,1000.0*SO2[1]
SO2_UGM3      ,ug/m3    ,1000.0*(SO2[1]*2.2118*DENS[3])
SULF          ,ppbV     ,1000.0*SULF[1]
TERP          ,ppbV     ,1000.0*TERP[1]
TOL           ,ppbV     ,1000.0*TOL[1]
VOC           ,ppbC
,1000.0*(PAR[1]+2.0*ETH[1]+2.0*ETOH[1]+2.0*OLE[1]+7.0*TOL[1]+8.0*XYLMN[1]+FORM[1]+2.0*ALD2[1]+5.0*ISOP
[1]+2.0*ETHA[1]+4.0*IOLE[1]+2.0*ALDX[1]+10.0*TERP[1]+10.0*NAPH[1])
XYLMN         ,ppbV     ,1000.0*XYLMN[1]

! Particles
!! crustal elements
AFEJ          ,ug/m3    ,AFEJ[1]
AALJ          ,ug/m3    ,AALJ[1]
ASIJ          ,ug/m3    ,ASIJ[1]
ATIJ          ,ug/m3    ,ATIJ[1]
ACAJ          ,ug/m3    ,ACAJ[1]
AMGJ          ,ug/m3    ,AMGJ[1]
AKJ           ,ug/m3    ,AKJ[1]
AMNJ          ,ug/m3    ,AMNJ[1]
ASOILJ        ,ug/m3    ,2.20*AALJ[1]+2.49*ASIJ[1]+1.63*ACAJ[1]+2.42*AFEJ[1]+1.94*ATIJ[1]
!! other PM species
AHPLUSIJ      ,ug/m3    ,(AH3OPI[1]+AH3OPJ[1])*1.0/19.0
ANAK          ,ug/m3    ,0.8373*ASEACAT[1]+0.0626*ASOIL[1]+0.0023*ACORS[1]
AMGK          ,ug/m3    ,0.0997*ASEACAT[1]          +0.0032*ACORS[1]
AKK           ,ug/m3    ,0.0310*ASEACAT[1]+0.0242*ASOIL[1]+0.0176*ACORS[1]
ACAK          ,ug/m3    ,0.0320*ASEACAT[1]+0.0838*ASOIL[1]+0.0562*ACORS[1]
ACLIJ         ,ug/m3    ,ACLI[1]+ACLJ[1]
AECIJ         ,ug/m3    ,AECI[1]+AECJ[1]
ANAIJ         ,ug/m3    ,ANAJ[1]+ANAI[1]
ANO3IJ        ,ug/m3    ,ANO3I[1]+ANO3J[1]
ANO3K         ,ug/m3    ,ANO3K[1]
TNO3          ,ug/m3    ,2175.6*(HNO3[1]*DENS[3])+ANO3I[1]+ANO3J[1]+ANO3K[1]
ANH4IJ        ,ug/m3    ,ANH4I[1]+ANH4J[1]
ANH4K         ,ug/m3    ,ANH4K[1]
AOCIJ         ,ugC/m3
,(AXYL1J[1]+AXYL2J[1]+AXYL3J[1])/2.0+(ATOL1J[1]+ATOL2J[1]+ATOL3J[1])/2.0+(ABNZ1J[1]+ABNZ2J[1]+ABNZ3J[
1])/2.0
+(AISO1J[1]+AISO2J[1])/1.6+AISO3J[1]/2.7+(ATRP1J[1]+ATRP2J[1])/1.4+ASQTJ[1]/2.1+AALK1J[1]/1.17+AALK2J[1]/1.1
7+AORGCJ[1]/2.0+(AOLGBJ[1]+AOLGAJ[1])/2.1+APOCI[1]+APOCJ[1]+(APAH1J[1]+APAH2J[1]+APAH3J[1])/2.03
AOMIJ         ,ug/m3
,AXYL1J[1]+AXYL2J[1]+AXYL3J[1]+ATOL1J[1]+ATOL2J[1]+ATOL3J[1]+ABNZ1J[1]+ABNZ2J[1]+ABNZ3J[1]+AISO1J
[1]+AISO2J[1]+AISO3J[1]+ATRP1J[1]+ATRP2J[1]+ASQTJ[1]+AALK1J[1]+AALK2J[1]+AORGCJ[1]+AOLGBJ[1]+AOLG
AJ[1]+APOCI[1]+APOCJ[1]+APNCOMI[1]+APNCOMJ[1]+APAH1J[1]+APAH2J[1]+APAH3J[1]
AORGAJ        ,ug/m3
,AXYL1J[1]+AXYL2J[1]+AXYL3J[1]+ATOL1J[1]+ATOL2J[1]+ATOL3J[1]+ABNZ1J[1]+ABNZ2J[1]+ABNZ3J[1]+AALK1
J[1]+AALK2J[1]+AOLGAJ[1]+APAH1J[1]+APAH2J[1]+APAH3J[1]
AORGBJ        ,ug/m3    ,AISO1J[1]+AISO2J[1]+AISO3J[1]+ATRP1J[1]+ATRP2J[1]+ASQTJ[1]+AOLGBJ[1]
AORGCJ        ,ug/m3    ,AORGCJ[1]
APOCIJ        ,ugC/m3   ,APOCI[1]+APOCJ[1]
APOAIJ        ,ug/m3    ,APOCIJ[0]+APNCOMI[1]+APNCOMJ[1]
ASO4IJ        ,ug/m3    ,ASO4I[1]+ASO4J[1]
ASO4K         ,ug/m3    ,ASO4K[1]
ATOTI         ,ug/m3
,ASO4I[1]+ANO3I[1]+ANH4I[1]+ANAI[1]+ACLI[1]+AECI[1]+APOCI[1]+APNCOMI[1]+AOTHRI[1]
ATOTJ         ,ug/m3    ,ASO4J[1]+ANO3J[1]+ANH4J[1]+ANAJ[1]+ACLJ[1]+AECJ[1]+AOMIJ[0]-
(APOCI[1]+APNCOMI[1])+AOTHRJ[1]+AFEJ[1]+ASIJ[1]+ATIJ[1]+ACAJ[1]+AMGJ[1]+AMNJ[1]+AALJ[1]+AKJ[1]
ATOTK         ,ug/m3    ,ASOIL[1]+ACORS[1]+ASEACAT[1]+ACLK[1]+ASO4K[1]+ANO3K[1]+ANH4K[1]
PMIJ          ,ug/m3    ,ATOTI[0]+ATOTJ[0]
PM10          ,ug/m3    ,PMIJ[0]+ATOTK[0]

```
AUNSPEC1IJ      ,ug/m3     ,PMIJ[0] - (ASOILJ[0] + ANO3IJ[0] + ASO4IJ[0] + ANH4IJ[0] + AOCIJ[0] + AECIJ[0] +
ANAIJ[0] + ACLIJ[0])
ANCOMIJ        ,ug/m3     ,AOMIJ[0]-AOCIJ[0]
AUNSPEC2IJ      ,ug/m3     ,AUNSPEC1IJ[0] - ANCOMIJ[0]
!! OM/OC ratios
AOMOCRAT_PRI   ,none      ,APOAIJ[0]/APOCIJ[0]
AOMOCRAT_TOT   ,none      ,AOMIJ[0]/AOCIJ[0]

!! PM2.5 sharp cutoff species
PM25_HP        ,ug/m3     ,(AH3OPI[1]*PM25AT[4]+AH3OPJ[1]*PM25AC[4]+AH3OPK[1]*PM25CO[4])*1.0/19.0
PM25_CL        ,ug/m3     ,ACLI[1]*PM25AT[4]+ACLJ[1]*PM25AC[4]+ACLK[1]*PM25CO[4]
PM25_EC        ,ug/m3     ,AECI[1]*PM25AT[4]+AECJ[1]*PM25AC[4]
PM25_NA        ,ug/m3     ,ANAI[1]*PM25AT[4]+ANAJ[1]*PM25AC[4]+ANAK[0]*PM25CO[4]
PM25_MG        ,ug/m3     ,        AMGJ[1]*PM25AC[4]+AMGK[0]*PM25CO[4]
PM25_K         ,ug/m3     ,        AKJ[1] *PM25AC[4]+AKK[0] *PM25CO[4]
PM25_CA        ,ug/m3     ,        ACAJ[1]*PM25AC[4]+ACAK[0]*PM25CO[4]
PM25_NH4       ,ug/m3     ,ANH4I[1]*PM25AT[4]+ANH4J[1]*PM25AC[4]+ANH4K[1]*PM25CO[4]
PM25_NO3       ,ug/m3     ,ANO3I[1]*PM25AT[4]+ANO3J[1]*PM25AC[4]+ANO3K[1]*PM25CO[4]
PM25_OC        ,ugC/m3    ,APOCI[1]*PM25AT[4]+(AOCIJ[0]-APOCI[1])*PM25AC[4]
PM25_SOIL      ,ug/m3     ,ASOILJ[0]*PM25AC[4]+(ASOIL[1]+ACORS[1])*PM25CO[4]
PM25_SO4       ,ug/m3     ,ASO4I[1]*PM25AT[4]+ASO4J[1]*PM25AC[4]+ASO4K[1]*PM25CO[4]
PM25_TOT       ,ug/m3     ,ATOTI[0]*PM25AT[4]+ATOTJ[0]*PM25AC[4]+ATOTK[0]*PM25CO[4]
PM25_UNSPEC1   ,ug/m3     ,PM25_TOT[0]-
(PM25_CL[0]+PM25_EC[0]+PM25_NA[0]+PM25_NH4[0]+PM25_NO3[0]+PM25_OC[0]+PM25_SOIL[0]+PM25_SO4[0])
PMC_CL         ,ug/m3     ,ACLI[1]+ACLJ[1]+ACLK[1]-PM25_CL[0]
PMC_NA         ,ug/m3     ,ANAIJ[0]+ANAK[0]*0.78-PM25_NA[0]
PMC_NH4        ,ug/m3     ,ANH4I[1]+ANH4J[1]+ANH4K[1]-PM25_NH4[0]
PMC_NO3        ,ug/m3     ,ANO3I[1]+ANO3J[1]+ANO3K[1]-PM25_NO3[0]
PMC_SO4        ,ug/m3     ,ASO4I[1]+ASO4J[1]+ASO4K[1]-PM25_SO4[0]
PMC_TOT        ,ug/m3     ,PM10[0]-PM25_TOT[0]

!Meteorology
DCV_Recon      ,deciview  ,DCV_Recon[2]
AIR_DENS       ,kg/m3     ,DENS[3]
RH         ,%         ,100.00*RH[4]
SFC_TMP        ,C         ,(TEMP2[5]-273.15)
PBLH           ,m         ,PBL[5]
SOL_RAD        ,WATTS/m2  ,RGRND[5]
precip        ,cm        ,RN[5]+RC[5]
WSPD10         ,m/s       ,WSPD10[5]
WDIR10         ,deg       ,WDIR10[5]

!FRM PM Equivalent Calculation
K          ,ppb^2   ,exp(118.87-24084/TEMP2[5]-6.025*log(TEMP2[5]))
P1         ,        ,exp(8763/TEMP2[5]+19.12*log(TEMP2[5])-135.94)
P2         ,        ,exp(9969/TEMP2[5]+16.22*log(TEMP2[5])-122.65)
P3         ,        ,exp(13875/TEMP2[5]+24.46*log(TEMP2[5])-182.61)
a          ,        ,1-RH[0]/100
K_prime        ,ppb^2   ,(P1[0]-P2[0]*a[0]+(P3[0]*a[0]*a[0]))*(a[0]^1.75)*K[0]
sqrt_Ki        ,ppb     ,sqrt(RH[0]<=61 ? K[0] : K_prime[0])
max_NO3_loss   ,ug/m3   ,745.7/TEMP2[5]*sqrt_Ki[0]
PM25_NO3_loss  ,ug/m3   ,max_NO3_loss[0]<=PM25_NO3[0] ? max_NO3_loss[0] : PM25_NO3[0]
ANO3IJ_loss    ,ug/m3   ,max_NO3_loss[0]<=ANO3IJ[0] ? max_NO3_loss[0] : ANO3IJ[0]
PM25_NH4_loss  ,ug/m3   ,PM25_NO3_loss[0]*(18/62)
ANH4IJ_loss    ,ug/m3   ,ANO3IJ_loss[0]*(18/62)
PMIJ_FRM       ,ug/m3   ,PMIJ[0]-(ANO3IJ_loss[0]+ANH4IJ_loss[0])+0.24*(ASO4IJ[0]+ANH4IJ[0]-ANH4IJ_loss[0])+0.5
PM25_FRM       ,ug/m3   ,PM25_TOT[0]-(PM25_NO3_loss[0]+PM25_NH4_loss[0])+0.24*(PM25_SO4[0]+PM25_NH4[0]-
PM25_NH4_loss[0])+0.5
```

**S.3 CMAQv5.1 CB05e51 AE6 Wet/Dry Deposition Species Definitions (available with the CMAQv5.1 release code)**

```
/ File [1]: DRYDEP
/ File [2]: WETDEP
/ File [3]: METCRO2D
/
/new species      ,units    ,expression

ANAK_D          ,kg/ha ,0.8373*ASEACAT[1]+0.0626*ASOIL[1]+0.0023*ACORS[1]
ANAK_W           ,kg/ha ,0.8373*ASEACAT[2]+0.0626*ASOIL[2]+0.0023*ACORS[2]
AMGK_D          ,kg/ha ,0.0997*ASEACAT[1]            +0.0032*ACORS[1]
AMGK_W           ,kg/ha ,0.0997*ASEACAT[2]           +0.0032*ACORS[2]
AKK_D           ,kg/ha ,0.0310*ASEACAT[1]+0.0242*ASOIL[1]+0.0176*ACORS[1]
AKK_W            ,kg/ha ,0.0310*ASEACAT[2]+0.0242*ASOIL[2]+0.0176*ACORS[2]
ACAK_D          ,kg/ha ,0.0320*ASEACAT[1]+0.0838*ASOIL[1]+0.0562*ACORS[1]
ACAK_W          ,kg/ha ,0.0320*ASEACAT[2]+0.0838*ASOIL[2]+0.0562*ACORS[2]

DDEP_NO2        ,kg/ha    ,NO2[1]
WDEP_NO2        ,kg/ha    ,NO2[2]
DDEP_NO         ,kg/ha    ,NO[1]
WDEP_NO         ,kg/ha    ,NO[2]
DDEP_FORM       ,kg/ha    ,FORM[1]
WDEP_FORM       ,kg/ha    ,FORM[2]
DDEP_H2O2       ,kg/ha    ,H2O2[1]
WDEP_H2O2        ,kg/ha    ,H2O2[2]
DDEP_N2O5       ,kg/ha    ,N2O5[1]
WDEP_N2O5        ,kg/ha    ,N2O5[2]
DDEP_HONO        ,kg/ha    ,HONO[1]
WDEP_HONO        ,kg/ha    ,HONO[2]
DDEP_HNO3        ,kg/ha    ,HNO3[1]
WDEP_HNO3        ,kg/ha    ,HNO3[2]
DDEP_ANO3IJ      ,kg/ha    ,ANO3I[1] + ANO3J[1]
DDEP_ANO3K       ,kg/ha    ,ANO3K[1]
WDEP_ANO3IJK      ,kg/ha    ,ANO3I[2] + ANO3J[2] + ANO3K[2]
DDEP_TNO3       ,kg/ha    ,ANO3I[1] + ANO3J[1] + ANO3K[1] + 0.984*HNO3[1]
WDEP_TNO3       ,kg/ha    ,ANO3I[2] + ANO3J[2] + ANO3K[2] + 0.984*HNO3[2]
DDEP_NTR        ,kg/ha    ,NTROH[1]+NTRALK[1]+NTRCN[1]+NTRPX[1]+NTRCNOH[1] + NTRM[1]+NTRI[1]
WDEP_NTR        ,kg/ha    ,NTROH[2]+NTRALK[2]+NTRCN[2]+NTRPX[2]+NTRCNOH[2] + NTRM[2]+NTRI[2]
DDEP_PANT       ,kg/ha    ,PAN[1] + PANX[1] + OPAN[1] + MAPAN[1]
WDEP_PANT        ,kg/ha    ,PAN[2] + PANX[2] + OPAN[2] + MAPAN[2]
DDEP_NH3        ,kg/ha    ,NH3[1]
WDEP_NH3        ,kg/ha    ,NH3[2]
DDEP_ANH4IJ      ,kg/ha    ,ANH4I[1] + ANH4J[1]
DDEP_ANH4K       ,kg/ha    ,ANH4K[1]
WDEP_ANH4IJK      ,kg/ha    ,ANH4I[2] + ANH4J[2] + ANH4K[2]
DDEP_NHX        ,kg/ha    ,ANH4I[1] + ANH4J[1] + ANH4K[1] + 1.059*NH3[1]
WDEP_NHX        ,kg/ha    ,ANH4I[2] + ANH4J[2] + ANH4K[2] + 1.059*NH3[2]
DDEP_SO2        ,kg/ha    ,SO2[1]
DDEP_ASO4IJ      ,kg/ha    ,ASO4I[1] + ASO4J[1]
DDEP_ASO4K       ,kg/ha    ,ASO4K[1]
WDEP_ASO4IJK      ,kg/ha    ,ASO4I[2] + ASO4J[2] + ASO4K[2]
WDEP_TSO4       ,kg/ha    ,ASO4I[2] + ASO4J[2] + ASO4K[2] + 1.5*SO2[2]
DDEP_AECIJ      ,kg/ha    ,AECI[1] +AECJ[1]
DDEP_AOCIJ       ,kg/ha
,(AXYL1J[1]+AXYL2J[1]+AXYL3J[1])/2.0+(ATOL1J[1]+ATOL2J[1]+ATOL3J[1])/2.0+(ABNZ1J[1]+ABNZ2J[1]+ABNZ3J[
1])/2.0+(AISO1J[1]+AISO2J[1])/1.6+AISO3J[1]/2.7+(ATRP1J[1]+ATRP2J[1])/1.4+ASQTJ[1]/2.1+0.64*(AALK1J[1]+AALK
2J[1])+(APAH1J[1]+APAH2J[1]+APAH3J[1])/2.03+AORGCJ[1]/2.0 +(AOLGBJ[1]+AOLGAJ[1])/2.1+APOCI[1]+APOCJ[1]
DDEP_SSSO4J      ,kg/ha    ,0.19579876*ANAJ[1]
```

```
DDEP_SSSO4K       ,kg/ha   ,0.19579876*ANAK_D[0]
WDEP_SSSO4JK      ,kg/ha   ,0.19579876*ANAJ[2] + 0.19579876*ANAK_W[0]
DDEP_ANAJ        ,kg/ha   ,ANAJ[1]
DDEP_ANAK        ,kg/ha   ,ANAK_D[0]
DDEP_ANAJK       ,kg/ha   ,ANAJ[1] + ANAK_D[0]
WDEP_ANAJK       ,kg/ha   ,ANAJ[2] + ANAK_W[0]
TDEP_ANAJK       ,kg/ha   ,DDEP_ANAJK[0] + WDEP_ANAJK[0]
DDEP_ACLJ        ,kg/ha   ,ACLJ[1]
DDEP_ACLK        ,kg/ha   ,ACLK[1]
DDEP_ACLJK       ,kg/ha   ,ACLJ[1] + ACLK[1]
WDEP_TCL         ,kg/ha   ,0.972*HCL[2]+0.435*CLNO2[2] + ACLJ[2] + ACLK[2]
TDEP_CL          ,kg/ha   ,DDEP_ACLJK[0] + WDEP_TCL[0]
DDEP_CAJ         ,kg/ha   ,ACAJ[1]
WDEP_CAJ         ,kg/ha   ,ACAJ[2]
DDEP_CAJK        ,kg/ha   ,ACAJ[1]+ACAK_D[0]
WDEP_CAJK        ,kg/ha   ,ACAJ[2]+ACAK_W[0]
DDEP_FEJ         ,kg/ha   ,AFEJ[1]
WDEP_FEJ         ,kg/ha   ,AFEJ[2]
DDEP_ALJ         ,kg/ha   ,AALJ[1]
WDEP_ALJ         ,kg/ha   ,AALJ[2]
DDEP_SIJ        ,kg/ha   ,ASIJ[1]
WDEP_SIJ        ,kg/ha   ,ASIJ[2]
DDEP_TIJ        ,kg/ha   ,ATIJ[1]
WDEP_TIJ        ,kg/ha   ,ATIJ[2]
DDEP_MGJ        ,kg/ha   ,AMGJ[1]
WDEP_MGJ         ,kg/ha   ,AMGJ[2]
DDEP_MGJK        ,kg/ha   ,AMGJ[1]+AMGK_D[0]
WDEP_MGJK        ,kg/ha   ,AMGJ[2]+AMGK_W[0]
DDEP_KJ         ,kg/ha   ,AKJ[1]
WDEP_KJ         ,kg/ha   ,AKJ[2]
DDEP_KJK         ,kg/ha   ,AKJ[1]+AKK_D[0]
WDEP_KJK         ,kg/ha   ,AKJ[2]+AKK_W[0]
DDEP_MNJ         ,kg/ha   ,AMNJ[1]
WDEP_MNJ         ,kg/ha   ,AMNJ[2]
DDEP_O3         ,kg/ha   ,O3[1]
WDEP_O3         ,kg/ha   ,O3[2]
WDEP_PNA         ,kg/ha   ,PNA[2]
RT             ,cm      ,RN[3] + RC[3]
DD_OXN_NOX       ,kg/ha   ,0.30435*NO2[1] + 0.46667*NO[1]
WD_OXN_NOX       ,kg/ha   ,0.30435*NO2[2] + 0.46667*NO[2]
DD_OXN_TNO3      ,kg/ha   ,0.22222*HNO3[1] + 0.22581*ANO3I[1] + 0.22581*ANO3J[1] + 0.22581*ANO3K[1]
WD_OXN_TNO3      ,kg/ha   ,0.22581*WDEP_TNO3[0]
DD_OXN_PANT      ,kg/ha   ,0.11570*PAN[1] + 0.11570*PANX[1] + 0.11570*OPAN[1] + 0.11570*MAPAN[1]
WD_OXN_PANT      ,kg/ha   ,0.11570*PAN[2] + 0.11570*PANX[2] + 0.11570*OPAN[2] + 0.11570*MAPAN[2]
DD_OXN_ORGN      ,kg/ha   ,0.10770*(NTROH[1]+NTRALK[1]+NTRCN[1]+NTRPX[1]+NTRCNOH[1] +
NTRM[1]+NTRI[1])
WD_OXN_ORGN      ,kg/ha   ,0.10770*(NTROH[2]+NTRALK[2]+NTRCN[2]+NTRPX[2]+NTRCNOH[2] +
NTRM[2]+NTRI[2])
/DD_OXN_OTHR     ,kg/ha   ,0.25926*N2O5[1] + 0.29787*HONO[1]+0.1717*CLNO2[1]
WD_OXN_OTHR      ,kg/ha   ,0.25926*N2O5[2] + 0.29787*HONO[2]+0.177720*PNA[2]+0.1717*CLNO2[2]
/DD_OXN_TOT      ,kg/ha   ,DD_OXN_NOX[0] + DD_OXN_TNO3[0] + DD_OXN_PANT[0] + DD_OXN_ORGN[0] +
DD_OXN_OTHR[0]
WD_OXN_TOT       ,kg/ha   ,WD_OXN_NOX[0] + WD_OXN_TNO3[0] + WD_OXN_PANT[0] + WD_OXN_ORGN[0] +
WD_OXN_OTHR[0]
/TD_OXN_TOT      ,kg/ha   ,DD_OXN_TOT[0] + WD_OXN_TOT[0]
/DD_OXN_TOTMEQ     ,meq/m2   ,7.14*DD_OXN_TOT[0]
WD_OXN_TOTMEQ     ,meq/m2   ,7.14*WD_OXN_TOT[0]
/TD_OXN_TOTMEQ     ,meq/m2   ,DD_OXN_TOTMEQ[0] + WD_OXN_TOTMEQ[0]
DD_REDN_TOT      ,kg/ha   ,0.7777*DDEP_NHX[0]
WD_REDN_TOT      ,kg/ha   ,0.7777*WDEP_NHX[0]
```

```
TD_REDN_TOT        ,kg/ha     ,DD_REDN_TOT[0] + WD_REDN_TOT[0]
DD_REDN_TOTMEQ     ,meq/m2    ,7.14*DD_REDN_TOT[0]
WD_REDN_TOTMEQ     ,meq/m2    ,7.14*WD_REDN_TOT[0]
TD_REDN_TOTMEQ     ,meq/m2    ,DD_REDN_TOTMEQ[0] + WD_REDN_TOTMEQ[0]
DD_S_TOT           ,kg/ha     ,0.5*SO2[1] + 0.33333*ASO4I[1] + 0.33333*ASO4J[1] + 0.33333*ASO4K[1]
WD_S_TOT           ,kg/ha     ,0.33333*WDEP_TSO4[0]
TD_S_TOT           ,kg/ha     ,DD_S_TOT[0] + WD_S_TOT[0]
DD_S_TOTMEQ        ,meq/m2    ,6.24*DD_S_TOT[0]
WD_S_TOTMEQ        ,meq/m2    ,6.24*WD_S_TOT[0]
TD_S_TOTMEQ        ,meq/m2    ,DD_S_TOTMEQ[0] + WD_S_TOTMEQ[0]
DD_S_SeaS          ,kg/ha     ,0.33333*DDEP_SSSO4J[0] + 0.33333*DDEP_SSSO4K[0]
WD_S_SeaS          ,kg/ha     ,0.33333*WDEP_SSSO4JK[0]
TD_S_SeaS          ,kg/ha     ,DD_S_SeaS[0] + WD_S_SeaS[0]
DD_S_SeaSMEQ       ,meq/m2    ,6.24*DD_S_SeaS[0]
WD_S_SeaSMEQ       ,meq/m2    ,6.24*WD_S_SeaS[0]
TD_S_SeaSMEQ       ,meq/m2    ,DD_S_SeaSMEQ[0] + WD_S_SeaSMEQ[0]
```

**S.4 – WRFv3.4 Namelist**

```
&time_control
start_year                  = 2011,
start_month                 = 12,
start_day                   = 27,
start_hour                  = 00,
start_minute                = 00,
start_second                = 00,
end_year                    = 2012,
end_month                   = 01,
end_day                     = 01,
end_hour                    = 00,
end_minute                  = 00,
end_second                  = 00,
interval_seconds            = 10800,
input_from_file             = .true.,
history_interval            = 60,
frames_per_outfile          = 24,
restart                     = .TRUE.,
restart_interval            = 7200,
io_form_history             = 2
io_form_restart             = 2
io_form_input               = 2
io_form_boundary            = 2
debug_level                 = 0
io_form_auxinput2           = 2
io_form_auxinput4           = 2
auxinput1_inname            = "metoa_em.d01.<date>"
auxinput4_inname            = "wrflowinp_d01"
auxinput4_interval          = 180
auxinput4_end_h             = 9001
write_hist_at_0h_rst        = .true.,
/

&domains
time_step                   = 60,
time_step_fract_num         = 0,
time_step_fract_den         = 1,
use_adaptive_time_step      = .false.
max_dom                     = 1,
s_we                        = 1,
e_we                        = 472,
s_sn                        = 1,
e_sn                        = 312,
s_vert                      = 1,
e_vert                      = 36,
p_top_requested             = 5000,
eta_levels                  = 1.000, 0.9975, 0.995, 0.990, 0.985,
                              0.980, 0.970, 0.960, 0.950,
                              0.940, 0.930, 0.920, 0.910,
                              0.900, 0.880, 0.860, 0.840,
                              0.820, 0.800, 0.770, 0.740,
                              0.700, 0.650, 0.600, 0.550,
                              0.500, 0.450, 0.400, 0.350,
                              0.300, 0.250, 0.200, 0.150,
                              0.100, 0.050, 0.000
dx                          = 12000,
dy                          = 12000,
grid_id                     = 1,
```

```
parent_id                   = 0,
i_parent_start              = 0,
j_parent_start              = 0,
parent_grid_ratio           = 1,
parent_time_step_ratio      = 1,
feedback                    = 1,
smooth_option               = 0,
/

&physics
mp_physics                  = 10,
ra_lw_physics               = 4,
ra_sw_physics               = 4,
radt                        = 20,
sf_sfclay_physics           = 7,
sf_surface_physics          = 7,
bl_pbl_physics              = 7,
bldt                        = 0,
cu_physics                  = 1,
kfeta_trigger               = 2,
cudt                        = 0,
isfflx                      = 1,
ifsnow                      = 1,
icloud                      = 1,
surface_input_source        = 1,
num_soil_layers             = 2,
sst_update                  = 1,
pxlsm_smois_init            = 0,
slope_rad                   = 1,
topo_shading                = 1,
shadlen                     = 25000.,
num_land_cat                = 40,
prec_acc_dt                 = 60,
mp_zero_out                 = 2,
fractional_seaice           = 1,
seaice_threshold            = 0.0,
/

&fdda
grid_fdda                   = 1,
grid_sfdda                  = 1,
pxlsm_soil_nudge            = 1,
sgfdda_inname               = "wrfsfdda_d01",
sgfdda_end_h                = 9001,
sgfdda_interval_m           = 180,
sgfdda_interval             = 10800,
gfdda_inname                = "wrffdda_d<domain>",
gfdda_end_h                 = 9001,
gfdda_interval_m            = 180,
fgdt                        = 0,
if_no_pbl_nudging_uv        = 1,
if_no_pbl_nudging_t         = 1,
if_no_pbl_nudging_q         = 1,
if_zfac_uv                  = 0,
 k_zfac_uv                  = 13,
if_zfac_t                   = 0,
 k_zfac_t                   = 13,
if_zfac_q                   = 0,
 k_zfac_q                   = 13,
guv                         = 0.0001,
```

```
gt                        = 0.0001,
gq                        = 0.00001,
guv_sfc                     = 0.0000,
gt_sfc                      = 0.0000,
gq_sfc                      = 0.0000,
if_ramping                  = 0,
dtramp_min                  = 60.0,
io_form_gfdda               = 2,
rinblw                    = 250.0
/

&dynamics
w_damping                   = 1,
diff_opt                    = 1,
km_opt                      = 4,
diff_6th_opt                = 2,
diff_6th_factor             = 0.12,
damp_opt                    = 3,
base_temp                   = 290.
zdamp                     = 5000.,
dampcoef                    = 0.05,
khdif                     = 0,
kvdif                     = 0,
non_hydrostatic             = .true.,
moist_adv_opt               = 2,
tke_adv_opt                 = 2,
scalar_adv_opt              = 2,
/

&dfi_control
dfi_opt                   = 0
dfi_nfilter               = 7
dfi_write_filtered_input        = .true.
dfi_write_dfi_history         = .false.
dfi_cutoff_seconds            = 60
dfi_time_dim                = 1000
dfi_bckstop_year            = 2006
dfi_bckstop_month             = 08
dfi_bckstop_day             = 04
dfi_bckstop_hour            = 12
dfi_bckstop_minute            = 00
dfi_bckstop_second            = 00
dfi_fwdstop_year            = 2006
dfi_fwdstop_month             = 08
dfi_fwdstop_day             = 04
dfi_fwdstop_hour            = 13
dfi_fwdstop_minute            = 00
dfi_fwdstop_second            = 00
/

&bdy_control
spec_bdy_width              = 5,
spec_zone                 = 1,
relax_zone                = 4,
specified                 = .true.,
nested                    = .false.,
/

&grib2
/
```

```
&namelist_quilt
nio_tasks_per_group = 0,
nio_groups = 1,
/
```

**S.5 – WRFv3.7 Namelist**

```
&time_control
start_year                   = 2011,
start_month                   = 12,
start_day                    = 27,
start_hour                   = 00,
start_minute                  = 00,
start_second                  = 00,
end_year                     = 2012,
end_month                     = 01,
end_day                      = 01,
end_hour                     = 00,
end_minute                    = 00,
end_second                    = 00,
interval_seconds              = 10800,
input_from_file               = .true.,
history_interval              = 60,
frames_per_outfile            = 24,
restart                     = .TRUE.,
restart_interval             = 7200,
io_form_history               = 2
io_form_restart               = 2
io_form_input                 = 2
io_form_boundary              = 2
debug_level                  = 0
io_form_auxinput2             = 2
io_form_auxinput4             = 2
auxinput1_inname              = "metoa_em.d01.<date>"
auxinput4_inname              = "wrflowinp_d01"
auxinput4_interval            = 180
auxinput4_end_h               = 9001
write_hist_at_0h_rst          = .true.,
/

&domains
time_step                    = 60,
time_step_fract_num            = 0,
time_step_fract_den            = 1,
use_adaptive_time_step          = .false.
max_dom                       = 1,
s_we                         = 1,
e_we                         = 472,
s_sn                         = 1,
e_sn                         = 312,
s_vert                        = 1,
e_vert                        = 36,
p_top_requested               = 5000,
eta_levels                    = 1.000, 0.9975, 0.995, 0.990, 0.985,
                             0.980, 0.970, 0.960, 0.950,
                             0.940, 0.930, 0.920, 0.910,
                             0.900, 0.880, 0.860, 0.840,
                             0.820, 0.800, 0.770, 0.740,
                             0.700, 0.650, 0.600, 0.550,
                             0.500, 0.450, 0.400, 0.350,
                             0.300, 0.250, 0.200, 0.150,
                             0.100, 0.050, 0.000
dx                          = 12000,
dy                          = 12000,
grid_id                       = 1,
```

```
parent_id                = 0,
i_parent_start           = 0,
j_parent_start           = 0,
parent_grid_ratio        = 1,
parent_time_step_ratio   = 1,
feedback                 = 1,
smooth_option            = 0,
/

&physics
mp_physics               = 10,
ra_lw_physics            = 4,
ra_sw_physics            = 4,
radt                     = 20,
sf_sfclay_physics        = 7,
sf_surface_physics       = 7,
bl_pbl_physics           = 7,
bldt                     = 0,
cu_physics               = 1,
kfeta_trigger            = 2,
cudt                     = 0,
isfflx                   = 1,
ifsnow                   = 1,
icloud                   = 1,
cu_rad_feedback          = .true.,
surface_input_source     = 1,
num_soil_layers          = 2,
sst_update               = 1,
pxlsm_smois_init         = 0,
slope_rad                = 1,
topo_shading             = 1,
shadlen                  = 25000.,
num_land_cat             = 40,
prec_acc_dt              = 60,
mp_zero_out              = 2,
fractional_seaice        = 1,
seaice_threshold         = 0.0,
/

&fdda
grid_fdda                = 1,
grid_sfdda               = 1,
pxlsm_soil_nudge         = 1,
sgfdda_inname            = "wrfsfdda_d01",
sgfdda_end_h             = 9001,
sgfdda_interval_m        = 180,
sgfdda_interval          = 10800,
gfdda_inname             = "wrffdda_d<domain>",
gfdda_end_h              = 9001,
gfdda_interval_m         = 180,
fgdt                     = 0,
if_no_pbl_nudging_uv     = 1,
if_no_pbl_nudging_t      = 1,
if_no_pbl_nudging_q      = 1,
if_zfac_uv               = 0,
 k_zfac_uv               = 13,
if_zfac_t                = 0,
 k_zfac_t                = 13,
if_zfac_q                = 0,
 k_zfac_q                = 13,
```

```
guv                      = 0.0001,
gt                       = 0.0001,
gq                       = 0.00001,
guv_sfc                   = 0.0000,
gt_sfc                    = 0.0000,
gq_sfc                    = 0.0000,
if_ramping                = 0,
dtramp_min                = 60.0,
io_form_gfdda             = 2,
rinblw                   = 250.0
/

&dynamics
w_damping                 = 1,
diff_opt                  = 1,
km_opt                    = 4,
diff_6th_opt              = 2,
diff_6th_factor           = 0.12,
damp_opt                  = 3,
base_temp                 = 290.
zdamp                     = 5000.,
dampcoef                  = 0.05,
khdif                     = 0,
kvdif                     = 0,
non_hydrostatic           = .true.,
moist_adv_opt             = 2,
tke_adv_opt               = 2,
scalar_adv_opt            = 2,
/

&dfi_control
dfi_opt                   = 0
dfi_nfilter               = 7
dfi_write_filtered_input     = .true.
dfi_write_dfi_history        = .false.
dfi_cutoff_seconds           = 60
dfi_time_dim                 = 1000
dfi_bckstop_year             = 2006
dfi_bckstop_month             = 08
dfi_bckstop_day              = 04
dfi_bckstop_hour             = 12
dfi_bckstop_minute            = 00
dfi_bckstop_second            = 00
dfi_fwdstop_year             = 2006
dfi_fwdstop_month             = 08
dfi_fwdstop_day              = 04
dfi_fwdstop_hour             = 13
dfi_fwdstop_minute            = 00
dfi_fwdstop_second            = 00
/

&bdy_control
spec_bdy_width               = 5,
spec_zone                 = 1,
relax_zone                = 4,
specified                 = .true.,
nested                    = .false.,
/

&grib2
```

```
/

&namelist_quilt
nio_tasks_per_group = 0,
nio_groups = 1,
/
```

[Figure]

**Figure S1: Difference in monthly average PM$_{2.5}$ (µgm$^{-3}$) for January (a) and July (b) and O$_3$ (ppbV) for January (c) and July (d) between the version 1 and version 2 emissions platform used.**

[Figure]

**Figure S2: CMAQv5.0.2 simulation seasonal average PM$_{2.5}$ concentrations (µgm$^{-3}$) for a) winter b) spring c) summer and d) fall.**

[Figure]

**Figure S3: CMAQv5.0.2 simulation seasonal average MDA8 O₃ mixing ratio (ppbv) for a) winter b) spring c) summer and d) fall. Note that the scales for each plot can vary.**

[Figure]

**Figure S4: Histograms of the difference in the absolute value of monthly average (2011) PM$_{2.5}$ mean bias for winter (DJF; top left), spring (MAM; top right), summer (JJA; bottom left) and fall (SON; bottom right) between CMAQ v5.0.2_Base and v5.1_Base (CMAQv5.1_Base – CMAQv5.0.2_Base). All plots are in units of µgm$^{-3}$. Cool colors indicate a reduction in PM$_{2.5}$ mean bias in CMAQv5.1_Base while warm color indicate an increase in PM$_{2.5}$ mean bias CMAQv5.1_Base.**

[Figure]

**Figure S5: Diurnal time series of winter PM$_{2.5}$ from AQS observations (grey), CMAQv5.0.2_Base (blue) and CMAQv5.1_Base (red) for concentration (top), mean bias (top middle), root mean square error (bottom middle) and correlation (bottom). All units are in μgm$^{-3}$ except for correlation.**

[Figure]

**Figure S6: Diurnal time series of spring PM$_{2.5}$ from AQS observations (grey), CMAQv5.0.2_Base (blue) and CMAQv5.1_Base (red) for concentration (top), mean bias (top middle), root mean square error (bottom middle) and correlation (bottom). All units are in μgm$^{-3}$ except for correlation.**

[Figure]

**Figure S7: Diurnal time series of summer PM₂.₅ from AQS observations (grey), CMAQv5.0.2_Base (blue) and CMAQv5.1_Base (red) for concentration (top), mean bias (top middle), root mean square error (bottom middle) and correlation (bottom). All units are in μgm⁻³ except for correlation.**

[Figure]

**Figure S8: Diurnal time series of fall PM$_{2.5}$ from AQS observations (grey), CMAQv5.0.2_Base (blue) and CMAQv5.1_Base (red) for concentration (top), mean bias (top middle), root mean square error (bottom middle) and correlation (bottom). All units are in μgm$^{-3}$ except for correlation.**

[Figure]

**Figure S9: Histograms of the difference in the absolute value of monthly average O₃ mean bias for winter (DJF; top left), spring (MAM; top right), summer (JJA; bottom left) and fall (SON; bottom right) between CMAQ v5.0.2_Base and v5.1_Base (CMAQv5.1_Base – CMAQv5.0.2_Base). All plots are in units of ppbV. Cool colors indicate a reduction in O₃ mean bias in CMAQv5.1_Base while warm color indicate an increase in O₃ mean bias CMAQv5.1_Base.**

[Figure]

**Figure S10: Diurnal time series of winter O3 from AQS observations (grey), CMAQv5.0.2_Base (blue) and CMAQv5.1_Base (red) for concentration (top), mean bias (top middle), root mean square error (bottom middle) and correlation (bottom). All units are in ppbV except for correlation.**

[Figure]

**Figure S11: Diurnal time series of winter NO$_X$ from AQS observations (grey), CMAQv5.0.2_Base (blue) and CMAQv5.1_Base (red) for concentration (top), mean bias (top middle), root mean square error (bottom middle) and correlation (bottom). All units are in ppbV except for correlation.**

[Figure]

**Figure S12: Diurnal time series of spring O₃ from AQS observations (grey), CMAQv5.0.2_Base (blue) and CMAQv5.1_Base (red) for concentration (top), mean bias (top middle), root mean square error (bottom middle) and correlation (bottom). All units are in ppbV except for correlation.**

[Figure]

**Figure S13. Diurnal time series of spring NOx from AQS observations (grey), CMAQv5.0.2_Base (blue) and CMAQv5.1_Base (red) for concentration (top), mean bias (top middle), root mean square error (bottom middle) and correlation (bottom). All units are in ppbV except for correlation.**

[Figure]

**Figure S14: Diurnal time series of summer O₃ from AQS observations (grey), CMAQv5.0.2_Base (blue) and CMAQv5.1_Base (red) for concentration (top), mean bias (top middle), root mean square error (bottom middle) and correlation (bottom). All units are in ppbV except for correlation.**

[Figure]

**Figure S15: Diurnal time series of summer NOx from AQS observations (grey), CMAQv5.0.2_Base (blue) and CMAQv5.1_Base (red) for concentration (top), mean bias (top middle), root mean square error (bottom middle) and correlation (bottom). All units are in ppbV except for correlation.**

[Figure]

**Figure S16: Diurnal time series of fall O₃ from AQS observations (grey), CMAQv5.0.2_Base (blue) and CMAQv5.1_Base (red) for concentration (top), mean bias (top middle), root mean square error (bottom middle) and correlation (bottom). All units are in ppbV except for correlation.**

[Figure]

**Figure S17: Diurnal time series of fall NO$_X$ from AQS observations (grey), CMAQv5.0.2_Base (blue) and CMAQv5.1_Base (red) for concentration (top), mean bias (top middle), root mean square error (bottom middle) and correlation (bottom). All units are in ppbV except for correlation.**

---

## Referee Report (RR1)

**Review of**

**Overview and evaluation of the Community Multiscale Air Quality (CMAQ) model version 5.1**

By *K. Wyat Appel, Sergey L. Napelenok, Kristen M. Foley, Havala O. T. Pye, Christian Hogrefe, Deborah J. Luecken, Jesse O. Bash, Shawn J. Roselle, Jonathan E. Pleim, Hosein Foroutan, William T. Hutzell, George A. Pouliot, Golam Sarwar, Kathleen M. Fahey, Brett Gantt, Robert C. Gilliam, Daiwen Kang, Rohit Mathur, Donna B. Schwede, Tanya L. Spero, David C. Wong, and Jeffrey O. Young*

**I bring here 3 points for the final review. Point 1 strikes me as critical and needs to be corrected because it leads to a (in my opinion) biased and unjustified piece of conclusion regarding the relative importance of the update in emissions and the scientific updates.**

Point 2 brings back to consideration a remark from the initial review that I think has been too overlooked by the authors,

Point 3 is a request for change in the color scale of a Figure so that the albedo is between 0 and 1

I nonetheless wish to thank the authors for the considerable work in the Review process, even though I still think that more written information about the model design could have been brough in this new CMAQ reference article.

Below, in green the Authors' text (either answers to my initial review or text from the manuscript), in black my text, in blue statements from my initial review.

**Point 1**

"Obviously it was not made clear in the manuscript that the overall impact from the emission platform change was small. Hopefully this is now made clear in the text. In addition, a figure showing the impact of the emissions platform change on ozone and PM2.5 in January and July has been added to the text to quantify to the reader the impact from the emissions platform change."

The following statement is introduced in the revised version (it would be easier to find if the Authors had indicated explicitly where they had made such an adjustement):
*"However, based on sensitivity simulations performed for January and July 2011 where the only difference was the emissions platform used, the differences in O3 30 and PM2.5 between those two simulations used were generally small and isolated, suggesting there is minimal impact to the comparison between the v5.0.2 and v5.1 simulations from the change in the emissions platform used. Figure S1 shows the impact on winter (January) and summer (July) O3 and PM2.5 between simulations using the different emission platforms."*

I have several remarks on the Author's response and the corresponding changes that have been performed :
- The Figure has not been added "in the text" but as a supplement S1
- The statement that "the differences between those two simulations were generally small and isolated" seem to me as very strange : if one looks at Fig. S1a along with Fig. 6a of the revised manuscript, it is evident that **about 100% of the PM25 difference between v5.0.2 and v5.1 is due to the change in**

**emissions !** Comparison with Figs. 5c, 4c, 2c and 1c reveal that all other causes of change in wintertime PM25 are about 1 to 2 order of magnitudes smaller than the changes in the emission dataset.

Unless the authors demonstrate in a convincing way that this remark is due to me misunderstanding the figures, which is possible, I recommend that :
- Fig. S1 is moved into the main manuscript since it reveals effects 1 order of magnitude larger than the figures shown in the main manuscript for wintertime PM2.5
- The statement that there is *"minimal impact to the comparison between the v5.0.2 and v5.1 simulations from the change in the emissions platform used."* strikes me as very biased at least regarding PM2.5 and I recommend that it is replaced by a more realistic discussion.

This failure to analyze in a realistic way the effect of emission changes on PM25 leads, in my opinion, to a biased conclusion : "the scientific updates in v5.1 resulted in relatively dramatic improvements in model performance for PM2.5 in winter and summer" while, as discussed above, comparison of Fig. S1 with Figs. 5c, 4c, 2c and 1c reveal that all other causes of change in wintertime PM25 are about 1 to 2 order of magnitudes smaller than the changes due to the upset in the emission dataset, so the effect of scientific updates alone seem to be at best marginal compared to the effects of the update of dataset. As wintertime PM25 concentrations are usually a major worry for air quality modellers (due to usually strong emissions and stable atmospheric conditions), **I consider critical that this statement is replaced by a statement telling explicitly that scientific updates brought relatively small changes to wintertime PM2.5 when compared to the emissions changes, not allowing the authors to quantify the changes in model performance regarding wintertime aerosols**. I think that this is a caveat of the study that needs to be acknowledged. The only effect of the scientific updates that seem very significant to me is the effect opf the new aerosol processes in summertime as shown in Fig. 2d.

**Point 2**

C1 p. 5, l. 31: Is this time interval valid for all the domain? Days in July should last much longer than 12 hours at least in the north of the domain, and the daytime interval must be very different from the west to the east of the simulation domain (about 5000 km, which is about 4 hours time lag in the solar time). Using points from 11:45 to 23:45 UTC from west to east would result to using data points from mid-morning to the sunset at the eastern part of the domain, and from dawn to mid-afternoon in the west of the domain, which is critical as cloudiness often has a strong diurnal cycle.

I recommend that all the available daytime data points shall be used for this comparison.

Response: All available data from the satellite product are used in the average in Figure 1a (Note this Figure has been moved and is now referred to as FigureXa). The figure in the Supplemental Information section S1 shows the number of daytime hours (11:45UTC – 23:45UTC) with available GOES cloud albedo data during July 1 – July 31, 2011 for the modeling domain. Regions in the eastern half of the US have a larger number of available satellite observations (on the order of 390hrs) compared to the western coast which has < 340hrs. Since the reference to the time window of 11:45UTC-23:45UTC caused unnecessary confusion we have removed this from the main text. We now point readers to the Supplemental Information for further description of the hours of available satellite data:
"The satellite data are available at 15 minutes prior to the top of the hour during daytime hours and were matched to model output at the top of the hour (see section S.1 in the supplemental material for further information)."

The figure that is shown in the Supplementary material seems to only confirm what I was stating, that the time window from 11:45UTC to 23:45 UTC is arbitrary and may produce problems biases : on that figure in the Supplement showing the number of available daypoints, a strong east-west gradient is visible, and while more than 400 daytime points are available for the center-east of the US, less than 340 are available for California, suggesting that late-afternoon points are missing over California and generally the west of the domain. Less daylight time on the west coast than east-coast would actually be a very surprising result...

In San Francisco in summertime, the sunset is about 20:30 local time (4:30 AM UTC), so 23:45UTC is 15:45 local time, what one would call mid-afternoon, almost 5 hours before sunset, so are 4 to 5 hours of valid data in the afternoon/evening discarded over the western US ? Why impose an arbitrary time window and not just use all available GOES data ?

While I do not consider this point critical, I consider that the figure shown in the Supplement only reveals that what I feared in my review is actually what happens, so instead of bypassing this remark **I would like to maintain the recommendation that all the daytime data points are used for the comparison, not just between 11:45 and 23:45 UTC**.

**Point 3**

C2 p. 5, l. 32: the description of Fig. 1 does not fit that in the caption of Fig. 1 (the latter one seems to be more relevant). The average cloud albedo seems not to be shown.
This should be clarified.

Response: The reviewer is correct. The wrong Figure 1 was included with the original submission of the manuscript. In the revised version of the manuscript this Figure (now called Figure 4) does show the average cloud albedo, consistent with the description in the text. The Figure caption has also been changed to say:
"Figure 4. The average cloud albedo during daytime hours in July 2011 derived from (a) the GOES satellite product (b) WRF3.7 (c) CMAQv5.1 with photolysis/cloud model treatment from v5.0.2 and WRF3.7 inputs (CMAQv5.1_RetroPhot) (d) CMAQv5.1 using WRF3.7 inputs (CMAQv5.1_Base)."

There is a problem in Fig. 4 : the albedo ranges between 0 and 40, it should be a value between 0 and 1. Actually, I had ot go through the Supplement to realize that the albedo is probably present as percentage values which is, I think, very uncommoon. I recommend that the albedo is brought to its classical dimensionless form between 0 and 1 in the Figure.

---

## Author Response (AR2)

**Overview and evaluation of the Community Multiscale Air Quality (CMAQ) model version 5.1**

By *K. Wyat Appel, Sergey L. Napelenok, Kristen M. Foley, Havala O. T. Pye, Christian Hogrefe, Deborah J. Luecken, Jesse O. Bash, Shawn J. Roselle, Jonathan E. Pleim, Hosein Foroutan, William T. Hutzell, George A. Pouliot, Golam Sarwar, Kathleen M. Fahey, Brett Gantt, Robert C. Gilliam, Daiwen Kang, Rohit Mathur, Donna B. Schwede, Tanya L. Spero, David C. Wong, and Jeffrey O. Young*

I bring here 3 points for the final review. Point 1 strikes me as critical and needs to be corrected because it leads to a (in my opinion) biased and unjustified piece of conclusion regarding the relative importance of the update in emissions and the scientific updates.

Point 2 brings back to consideration a remark from the initial review that I think has been too overlooked by the authors,

Point 3 is a request for change in the color scale of a Figure so that the albedo is between 0 and 1

I nonetheless wish to thank the authors for the considerable work in the Review process, even though I still think that more written information about the model design could have been brought in this new CMAQ reference article.

Below, in green the Authors' text (either answers to my initial review or text from the manuscript), in black my text, in blue statements from my initial review.

**Point 1**
"Obviously it was not made clear in the manuscript that the overall impact from the emission platform change was small. Hopefully this is now made clear in the text. In addition, a figure showing the impact of the emissions platform change on ozone and PM2.5 in January and July has been added to the text to quantify to the reader the impact from the emissions platform change."

The following statement is introduced in the revised version (it would be easier to find if the Authors had indicated explicitly where they had made such an adjustment):
"*However, based on sensitivity simulations performed for January and July 2011 where the only difference was the emissions platform used, the differences in O3 30 and PM2.5 between those two simulations used were generally small and isolated, suggesting there is minimal impact to the comparison between the v5.0.2 and v5.1 simulations from the change in the emissions platform used. Figure S1 shows the impact on winter (January) and summer (July) O3 and PM2.5 between simulations using the different emission platforms.*"

I have several remarks on the Author's response and the corresponding changes that have been performed:
- The Figure has not been added "in the text" but as a supplement S1
- The statement that "the differences between those two simulations were generally small and isolated" seem to me as very strange: if one looks at Fig. S1a along with Fig. 6a of the revised

manuscript, it is evident that about 100% of the PM25 difference between v5.0.2 and v5.1 is due to the change in emissions! Comparison with Figs. 5c, 4c, 2c and 1c reveal that all other causes of change in wintertime PM25 are about 1 to 2 order of magnitudes smaller than the changes in the emission dataset. Unless the authors demonstrate in a convincing way that this remark is due to me misunderstanding the figures, which is possible, I recommend that:
- Fig. S1 is moved into the main manuscript since it reveals effects 1 order of magnitude larger than the figures shown in the main manuscript for wintertime PM2.5
- The statement that there is "*minimal impact to the comparison between the v5.0.2 and v5.1 simulations from the change in the emissions platform used.*" strikes me as very biased at least regarding PM2.5 and I recommend that it is replaced by a more realistic discussion.

This failure to analyze in a realistic way the effect of emission changes on PM25 leads, in my opinion, to a biased conclusion: "the scientific updates in v5.1 resulted in relatively dramatic improvements in model performance for PM2.5 in winter and summer" while, as discussed above, comparison of Fig. S1 with Figs. 5c, 4c, 2c and 1c reveal that all other causes of change in wintertime PM25 are about 1 to 2 order of magnitudes smaller than the changes due to the upset in the emission dataset, so the effect of scientific updates alone seem to be at best marginal compared to the effects of the update of dataset. As wintertime PM25 concentrations are usually a major worry for air quality modellers (due to usually strong emissions and stable atmospheric conditions), I consider critical that this statement is replaced by a statement telling explicitly that scientific updates brought relatively small changes to wintertime PM2.5 when compared to the emissions changes, not allowing the authors to quantify the changes in model performance regarding wintertime aerosols. I think that this is a caveat of the study that needs to be acknowledged. The only effect of the scientific updates that seem very significant to me is the effect of the new aerosol processes in summertime as shown in Fig. 2d.

**Response: After considering how to address the issue raised by the reviewer, we determined the best approach was to re-run the annual CMAQv5.1 simulation using the same emissions as the CMAQv5.0.2 simulation, thereby eliminating any differences in model performance caused by differences in emissions. As such, the analysis presented in Section 5 has been updated to present results of CMAQv5.0.2 and CMAQv5.1 simulations that utilize the exact same emissions inventory, and all aspects of the analysis presented in that section have been updated correspondingly.**

**Indeed, as indicated by the reviewer some differences in model performance (particularly in the winter) were certainly attributable to differences in the emissions. Overall, the large-scale changes in model performance remain when the differences due to emissions are removed, however a majority of the more isolated differences (and some larger-scale differences) in model performance disappear once the effects of the difference in emission inputs are removed. Note that the max/min values on the scales on Figure 6 have been lowered for most panels, thereby highlighting differences between the model versions that were not apparent previously with the larger concentration scale.**

**Hopefully the reviewer will agree that the analysis now presented in Section 5 represents a true difference due to the updates to the modeling system for CMAQv5.1 without the influence from emission changes which can be considered outside the updates to the**

**modeling system. In addition, the statement in the text singled out above by the reviewer has been modified to remove the words "relatively dramatic". Finally, the supplemental Figure showing the difference in PM$_{2.5}$ and O$_3$ due to the different emissions inventories has been removed as it is no longer relevant to the analysis presented.**

**Point 2**

C1 p. 5, l. 31: Is this time interval valid for all the domain? Days in July should last much longer than 12 hours at least in the north of the domain, and the daytime interval must be very different from the west to the east of the simulation domain (about 5000 km, which is about 4 hours time lag in the solar time). Using points from 11:45 to 23:45 UTC from west to east would result to using data points from mid-morning to the sunset at the eastern part of the domain, and from dawn to mid-afternoon in the west of the domain, which is critical as cloudiness often has a strong diurnal cycle.

I recommend that all the available daytime data points shall be used for this comparison.

Response: All available data from the satellite product are used in the average in Figure 1a (Note this Figure has been moved and is now referred to as FigureXa). The figure in the Supplemental Information section S1 shows the number of daytime hours (11:45UTC – 23:45UTC) with available GOES cloud albedo data during July 1 – July 31, 2011 for the modeling domain. Regions in the eastern half of the US have a larger number of available satellite observations (on the order of 390hrs) compared to the western coast which has < 340hrs. Since the reference to the time window of 11:45UTC-23:45UTC caused unnecessary confusion we have removed this from the main text. We now point readers to the Supplemental Information for further description of the hours of available satellite data:
"The satellite data are available at 15 minutes prior to the top of the hour during daytime hours and were matched to model output at the top of the hour (see section S.1 in the supplemental material for further information)."

The figure that is shown in the Supplementary material seems to only confirm what I was stating, that the time window from 11:45UTC to 23:45 UTC is arbitrary and may produce problems biases : on that figure in the Supplement showing the number of available daypoints, a strong east-west gradient is visible, and while more than 400 daytime points are available for the center-east of the US, less than 340 are available for California, suggesting that late-afternoon points are missing over California and generally the west of the domain. Less daylight time on the west coast than east-coast would actually be a very surprising result...

In San Francisco in summertime, the sunset is about 20:30 local time (4:30 AM UTC), so 23:45UTC is 15:45 local time, what one would call mid-afternoon, almost 5 hours before sunset, so are 4 to 5 hours of valid data in the afternoon/evening discarded over the western US? Why impose an arbitrary time window and not just use all available GOES data?

While I do not consider this point critical, I consider that the figure shown in the Supplement only reveals that what I feared in my review is actually what happens, so instead of bypassing this remark. I would like to maintain the recommendation that all the daytime data points are used for the comparison, not just between 11:45 and 23:45 UTC.

**Response:**
The satellite albedo data downloaded from the NASA website is completely missing for the entire domain outside the stated time window. The authors did not select this time window in any way, but rather were trying to document when there was missing satellite data. In this way, the original plot did use all available GOES data for the month of July 2011. There was no screening involved, it was simply an average of all non-missing data points in the dataset downloaded from the web. Due to the location of the geostationary satellite, there were some hours that had data for only part of the domain, with data at either the eastern or western edge of the domain missing, depending on the time day. The figure in the supplemental material was intended to show the available data from the GOES product for July 2011 that was used in the monthly average. However, since this has caused confusion we have now computed the average using only the hours that had a complete set of satellite data, i.e. removing hours that had any missing data along the eastern or western edges.

As a result, the monthly average albedo shown in the new Figure 4 in the text is an average of 301 hours of data for every grid cell. This is the new figure used in the text and we have updated the Figure caption and the description in section 4.3 accordingly. Below we show this new figure and the original figure which was an average of 301-394 hours, depending on the grid cell. With careful inspection, some very slight differences can be seen in the color gradients across the US. However, the conclusions drawn from this figure in terms of the modeled versus observed albedo are unchanged. We regret that the language in the original text and the response to your original question did not more clearly explain how this figure was generated. Thank you for the opportunity to clarify this and improve the paper for other readers.

New text, page 12, lines 33-37:
The satellite data are available at 15 minutes prior to the top of the hour during daytime hours and were matched to model output at the top of the hour. There were 301 hours with available satellite data across the domain in July 2011. Figure 4 shows the average cloud albedo (i.e. reflectivity at the top of the atmosphere) during these 301 hours in July derived from the GOES 35 satellite product (Figure 4a), and the cloud parameterizations within: WRF3.7 (Figure 4b), CMAQv5.1_RetroPhot (Figure 4c) and CMAQv5.1_Base (Figure 4d).

New figure caption:
Figure 4. The average cloud albedo during daytime hours in July 2011 with available satellite data (n = 301 hours total) derived from (a) the GOES satellite product (b) WRF3.7 (c) CMAQv5.1 with photolysis/cloud model treatment from v5.0.2 and WRF3.7 inputs (CMAQv5.1_RetroPhot) (d) CMAQv5.1 using WRF3.7 inputs (CMAQv5.1_Base).

Supplemental Material:
The Figure in Section S.1 has been removed since the number of hours of GOES data used in the calculation of the mean albedo is now 301 for every grid cell, making a spatial plot unnecessary. The categorical metrics in Table S1 have been updated based on the new

sample size.  The numbers change slightly, however the conclusions about CMAQv5.1 performance are the same as the original text.

[Figure]

**Original Figure 4 (with new color legend showing albedo as a fraction rather than a percent): The average cloud albedo during daytime hours in July 2011 derived from (a) the GOES satellite product (b) WRF3.7 (c) CMAQv5.1 with photolysis/cloud model treatment from v5.0.2 and WRF3.7 inputs (CMAQv5.1_RetroPhot) (d) CMAQv5.1 using WRF3.7 inputs (CMAQv5.1_Base).**

[Figure]

**New Figure 4: The average cloud albedo during daytime hours in July 2011 with available satellite data (n = 301 hours total) derived from (a) the GOES satellite product (b) WRF3.7 (c) CMAQv5.1 with photolysis/cloud model treatment from v5.0.2 and WRF3.7 inputs (CMAQv5.1_RetroPhot) (d) CMAQv5.1 using WRF3.7 inputs (CMAQv5.1_Base).**

C2 p. 5, l. 32: the description of Fig. 1 does not fit that in the caption of Fig. 1 (the latter one seems to  be more relevant). The average cloud albedo seems not to be shown.
This should be clarified.

Response: The reviewer is correct. The wrong Figure 1 was included with the original submission of  the manuscript. In the revised version of the manuscript this Figure (now called Figure 4) does show   the average cloud albedo, consistent with the description in the text. The Figure caption has also been   changed to say:
"Figure 4. The average cloud albedo during daytime hours in July 2011 derived from (a) the GOES   satellite product (b) WRF3.7 (c) CMAQv5.1 with photolysis/cloud model treatment from v5.0.2 and   WRF3.7 inputs (CMAQv5.1_RetroPhot) (d) CMAQv5.1 using WRF3.7 inputs (CMAQv5.1_Base)."

**Point 3**
There is a problem in Fig. 4: the albedo ranges between 0 and 40, it should be a value between 0 and 1.  Actually, I had to go through the Supplement to realize that the albedo is probably present as percentage values which is, I think, very uncommon. I recommend that the albedo is brought to its   classical dimensionless form between 0 and 1 in the Figure.

**Response:**
**The color scale for Figure 4 has been changed to a fractional value rather than a percent as**

**recommended.**

[revised manuscript text omitted]

**2.5.4 Gravitational Settling**

Previous evaluations of the ground-level coarse particle ($PM_{10}$ - $PM_{2.5}$) concentrations in CMAQ have shown that the model
significantly underestimated the total $PM_{10}$ concentrations (Appel et al., 2012). Contributing to this underestimation is the
fact that CMAQ previously did not have a mechanism in place to allow coarse particles to settle from upper layers to lower
layers (although coarse particles in layer one can settle to the surface). As a result, large particles that would normally settle
to lower layers in the model could remain trapped in the layers in which they were emitted or formed. To account for this
deficiency in the model, the effects of gravitational settling of coarse aerosols from upper to lower layers has been added to
v5.1 to more realistically simulate the aerosol mass distribution. The net effect of this update is an increase in ground-level
$PM_{10}$ concentrations in v5.1 compared to v5.0.2, particularly near coastal areas impacted by sea-spray (Nolte et al., 2015).

[revised manuscript text omitted]

 Two sets of emission input data were utilized for the analysis presented here. Both sets of emission data were based on the 2011 NEI, with version 1 (v1) of the 2011 NEI modeling platform developed by the USEPA from regulatory applications (https://www.epa.gov/sites/production/files/2015-08/documents/lite_finalversion_ver10.pdf)  utilized for the majority of the simlations, while version 2 (v2) of the 2011 modeling platform  was utilized for one set of sensitivity simulations. However, ~~based on sensitivity simulations performed for January and July 2011 where the only difference was the emissions platform used, the differences in O$_3$ and PM$_{2.5}$ between those two simulations used were generally small and isolated, suggesting there is minimal impact to the comparison between the v5.0.2 and v5.1 simulations from the change in the emissions platform used. Figure S1 shows the impact on winter (January) and summer (July) O$_3$ and PM$_{2.5}$ between simulations using the different emission platforms~~ all the comparisons of model simulations presented here are shown with simulations that utitilzed the exact same emissions inventory, and as such any differences in model performance are not the result of differences in emissions. See Table 2 for information regarding which version of the emission inventory was utilized for each simulation.

[revised manuscript text omitted]

**5.1  $PM_{2.5}$**

Figure 6 shows the seasonal average difference in model simulated $PM_{2.5}$ between v5.0.2 and v5.1 (CMAQv5.1_Base_NEIv1 – CMAQv5.0.2_Base), with cool colors indicating a decrease in $PM_{2.5}$ in v5.1 (versus v5.0.2) and warm colors indicating an increase in $PM_{2.5}$. Figure 7 shows the seasonal mean bias (MB) for $PM_{2.5}$ for the CMAQv5.1_Base_NEIv1 simulation, while Figure 8 shows the change in the absolute value of the seasonal mean bias (|MB|) in $PM_{2.5}$ between the CMAQv5.0.2_Base and CMAQv5.1_Base_NEIv1 simulations. Cool colors indicate smaller $PM_{2.5}$ |MB| in the CMAQv5.1_Base_NEIv1 simulation (versus the CMAQv5.0.2_Base simulation), while warm colors indicate larger |MB| in the CMAQv5.1_Base_NEIv1 simulation.

During winter, v5.1  simulates lower $PM_{2.5}$ concentrations in the eastern U.S. and portions of  central Canada compared to v5.0.2,  and higher $PM_{2.5}$ concentrations in the SJV (Figure 6).  $PM_{2.5}$  is largely overestimated in the eastern U.S. and underestimated in the western U.S. (exception being portions of the Northwest) in the winter in  CMAQv5.1_Base_NEIv1 simulation (Figure 7a). The change in |MB| between v5.0.2 and v5.1 is negative (reduced MB in v5.1) across the majority of the sites, with relatively large reductions (3-5 $\mu gm^{-3}$) in |MB| in the Northeast, upper Midwest (i.e. Great Lakes region)  (and the SJV (Figure 8a).  Figure S3 presents a histogram of the change in $PM_{2.5}$ |MB| using the same data and color scale as in Figure 8. It  is clear from the histogram the large percentage (72.3%) of sites where the |MB| decreases in the

CMAQv5.1_Base_NEIv1 simulation in the winter (Figure S3a), demonstrating a  widespread improvement in the PM$_{2.5}$ performance for v5.1 versus v5.0.2.

515    The diurnal profile of PM$_{2.5}$ for winter (Figure 9a)  indicates a relatively large decrease in MB throughout most of the day  with v5.1 versus v5.0.2, particularly during the overnight, morning and late afternoon hours. A similar improvement is seen in the RMSE,  and the correlation also improves for all hours (Figure S5). Figure 10 shows seasonal and regional stacked bar plots of PM$_{2.5}$ composition (SO$_4^{2-}$, NO$_3^-$, NH$_4^+$, EC, OC, soil, NaCl, NCOM, and Other). Soil is based on the IMPROVE soil equation and contains both primary and secondary sources of soil (Appel et al., 2013), while

520   Other represents the unspeciated PM mass in the inventory (see Appel et al., 2008) The five regions shown in Figure 10 are the Northeast (Maine, New Hampshire, Vermont, Massachusetts, New York, New Jersey, Maryland, Delaware, Connecticut, Rhode Island, Pennsylvania, District of Columbia, Virginia and West Virginia), Great Lakes (Ohio, Michigan, Indiana, Illinois and Wisconsin), Atlantic (North Carolina, South Carolina, Georgia and Florida), South (Kentucky, Tennessee, Mississippi, Alabama, Louisiana, Missouri, Oklahoma and Arkansas) and West (California, Oregon, Washington, Arizona, Nevada, New

525   Mexico). These regions are derived from principle component analysis to group states with similar PM$_{2.5}$ source regions together. For winter, the total PM$_{2.5}$ high bias is reduced across all five regions, with most of the improvement coming from  reductions in OC, non-carbon organic matter (NCOM; see S.2 or S.3 for definition) and Other, indicating that improvements in the representation of mixing under stable conditions helped in reducing the high bias. Still, a large bias remains for OC, which may be due in part to an overestimation of the residential wood combustion in

530   the NEI.

   For spring, the changes in PM$_{2.5}$ are much more isolated than in winter (Figure 6b), with the largest decreases occurring around Montreal (Canada)  and portions of the Midwest and desert Southwest (lack of  windblown dust in v5.1  contributes to the decrease in the desert Southwest).  The MB for PM$_{2.5}$ in the spring

535   is relatively small, with most sites  (75%) reporting a MB between $\pm 3.0$ $\mu$gm$^{-3}$,  with some larger underestimations in Texas and larger overestimations in the Northeast, Great Lakes and Northwest  (Figure 7b). As expected with the relatively small  change in modeled PM$_{2.5}$ concentrations  with v5.1 in the spring (Figure 6b), the difference in |MB| between v5.0.2 and v5.1 is relatively small, with most differences in |MB| less than $\pm 1.0$ $\mu$gm$^{-3}$ (Figure 8b). Some slightly larger decreases in |MB| occur in the Northeast and

540   Northwest, while some larger increases in |MB| occur in the Midwest and Texas. A little more than half (53.0%) of the sites  report an improvement in |MB| (Figure S3b). The diurnal profile of PM$_{2.5}$ for spring shows a consistent underestimation of PM$_{2.5}$ throughout most the day in the v5.0.2 simulation, which becomes larger in the CMAQv5.1 Base_NEIv1 simulation, with an overall decrease in PM$_{2.5}$ in the spring (Figure 9b). However, the RMSE is lower during the overnight, morning and afternoon hours in the CMAQv5.1

545    Base_NEIv1 simulation, and the correlation improves throughout most of the day as well (Figure S5). Total PM$_{2.5}$ MB improves in three of the five regions shown in Figure 10, with

most of the improvement coming from lower concentrations of OC and NCOM.

In the summer, $PM_{2.5}$ is considerably higher ($> 5.0\ \mu gm^{-3}$) across a large portion of the eastern U.S. in the CMAQv5.1_Base_NEIv1

 simulation, particularly in Mississippi, Alabama, Georgia and portions of the Ohio Valley  (Figure 6c). The  increase in $PM_{2.5}$ is primarily due to the updates to the IEPOX-SOA chemistry in v5.1 (Figure 2), updates to BVOC emissions in BEIS v3.61 (approximately 1.0 $\mu gm^{-3}$ increase $PM_{2.5}$ in the southwest U.S.), and

 the ACM2/MOL updates in WRF and CMAQ (Figure 1), with smaller contributions from the updates in CB05e51 chemical mechanism (Figure 5) and updates to the clouds/photolysis (Figure 3). Despite the increase in $PM_{2.5}$ with v5.1, $PM_{2.5}$ still remains largely underestimated in the summer, with the largest underestimations in the southeast U.S., Texas and California (Figure 7c). However, the result of the widespread increase in $PM_{2.5}$ with v5.1  is a similar large, widespread reduction in the |MB| across the eastern U.S., particularly in the Southeast

  and the Ohio Valley, where reductions in |MB| range from 3.0 - 5.0 $\mu gm^{-3}$(Figure 8c). Smaller increases in the |MB| (typically less than 2.0 $\mu gm^{-3}$) occur in  Florida and isolated areas in the western U.S. Of all the sites,  69.8% report an improvement in |MB|, with a  number of sites showing reductions in |MB| greater than 5.0 $\mu gm^{-3}$ (Figure S3c). $PM_{2.5}$ is underestimated throughout the day in both the v5.0.2 and v5.1 simulations (Figure 9c) in summer, with the underestimation improving slightly with v5.1, particularly during the af-

 ternoon and overnight hours. RMSE improves during the daytime hours with v5.1, while correlation is considerably higher with v5.1 than v5.0.2 throughout the entire day (Figure S6). Total $PM_{2.5}$ is underestimated in the CMAQv5.0.2_Base simulation in four of the five regions (West region being the exception),  which improves in the CMAQv5.1_Base_NEIv1 simulation (Figure 10). The overestimation in the West region with v5.0.2 also improves with v5.1. Small increases in $SO_4^{2-}$ and $NH_4^+$, and larger increases in OC and

 NCOM contributing to the improvement.

For the fall, the difference in $PM_{2.5}$ between v5.0.2 and v5.1 is again small, with the largest increases occurring in isolated portions of the eastern U.S. and California, and the largest decreases occurring in Montreal  and isolated areas in the western U.S. (Figure 6d). The  MB pattern in the fall (Figure 7d) is similar

 to the one in the spring as well (Figure 7b), with relatively small MBs in the Eastern U.S. ($\pm2.0\ \mu gm^{-3}$) and larger MBs along the west coast (underestimated in California and overestimated in the Northwest). As expected, the change in the |MB| between v5.0.2 and v5.1 is also relatively small in the fall, with the majority of the sites reporting a change in |MB| of less than $\pm2.0\ \mu gm^{-3}$ (Figure 8d), and 68.1% of the sites reporting a reduction in |MB| (Figure S3d). The average diurnal profile of $PM_{2.5}$ in the fall (Figure 9d) is similar to the spring, with improved MB with v5.1 during the overnight,

 morning and late afternoon/evening hours and reduced RMSE and improved correlation throughout the entire day (Figure S8). Total $PM_{2.5}$ is overestimated in all five regions in the fall (Figure 10), but improves with v5.1

in  all of those regions ( albeit only very slightly for the South region), with decreases in  EC and OC responsible for most of the improvement.

**5.2 Ozone**

[revised manuscript text omitted]

760  Conversely, during the summer when $PM_{2.5}$ is  larely underestimated by CMAQ over the U.S., $PM_{2.5}$ concentrations are typically higher with v5.1 versus v5.0.2, particularly in the southeastern U.S. The change in $O_3$ mixing ratios in v5.1 resulted in mixed improvement in MB, both spatially and temporally, with the summer showing the largest increase in MB. However, RMSE largely improved regardless of season and showed a larger improvement spatially across the sites than MB, and the correlation  was almost always higher with v5.1. Comparisons of vertical profiles of several

765  species taken over Edgewood, MD during the DISCOVER-AQ campaign showed improved performance with v5.1 throughout the PBL for $O_3$, $NO_2$, $NO_y$, ANs and CO, with the PNs being the only species to show degraded performance on that day.

The response of the model to changes in emission inputs was examined by comparing the ratio of the base v5.0.2 and v5.1 simulations to sensitivity simulations with 50% cuts each to anthropogenic $NO_x$, VOC and $SO_x$ emissions. CMAQv5.1 simulated MDA8 $O_3$ exhibited more responsiveness (greater reduction) to the 50% $NO_x$ cut in January and July than v5.0.2,

770  which is considered an improvement as previous studies suggested CMAQ $O_3$ to be under-responsive to large changes in emissions. The responsiveness of $PM_{2.5}$ to the emission cuts is more complicated than for $O_3$ since there are many more species comprising $PM_{2.5}$ and some of those have greater or smaller response with v5.1. However, the new pathways of formation for several $PM_{2.5}$ components in v5.1 generally result in greater responsiveness in v5.1 compared to v5.0.2 for the various emission cut scenarios.

[revised manuscript text omitted]
  ppbv and $PM_{2.5}$ plots are in units of $\mu gm^{-3}$. Note that the scales  between each plot  may vary.

[Figure]

**Figure 4.** The average cloud albedo during daytime hours in July 2011 with available satellite data (n = 301 hours total) derived from (a) the GOES satellite product (b) WRF3.7 (c) CMAQv5.1 with photolysis/cloud model treatment from v5.0.2 and WRF3.7 inputs (CMAQv5.1_RetroPhot) (d) CMAQv5.1 using WRF3.7 inputs (CMAQv5.1_Base_NEIv2).

[Figure]

**Figure 5.** Difference in the monthly average $O_3$ for a) January and b) July and $PM_{2.5}$ (with organic matter mass removed) for c) January and d) July between CMAQ v5.1_Base_NEIv2 and v5.1_TUCL (CMAQv5.1_Base_NEIv2 - CMAQv5.1_TUCL). $O_3$ plots are in units of  ppbv and $PM_{2.5}$ plots are in units of $\mu gm^{-3}$. Note that the scales for each plot can vary.

[Figure]

**Figure 6.** Difference in the seasonal average $PM_{2.5}$ for a) winter (DJF) b) spring (MAM) c) summer (JJA) and d) fall (SON) between CMAQv5.0.2_Base and CMAQv5.1_Base_NEIv1 (CMAQv5.1_Base_NEIv1 - CMAQv5.0.2_Base). All plots are in units of $\mu gm^{-3}$.

[Figure]

**Figure 7.** Seasonal average PM$_{2.5}$ mean bias ($\mu$gm$^{-3}$) at IMPROVE (circles), CSN (triangles), AQS Hourly (squares) and AQS Daily (diamonds) sites for a) winter (DJF) b) spring (MAM) c) summer (JJA) and d) fall (SON) for the CMAQ v5.1_Base simulation.

[Figure]

**Figure 8.** Difference in the absolute value of seasonal average PM$_{2.5}$ mean bias for a) winter (DJF) b) spring (MAM) c) summer (JJA) and d) fall (SON) between CMAQ v5.0.2_Base and v5.1_Base_NEIv1 (CMAQv5.1_Base_NEIv1 - CMAQv5.0.2_Base). All plots are in units of $\mu$gm$^{-3}$. Cool colors indicate a reduction in PM$_{2.5}$ mean bias in v5.1 while warm color indicate an increase in PM$_{2.5}$ mean bias v5.1.

[Figure]

**Figure 9.** Diurnal time series of seasonal PM$_{2.5}$ ($\mu$gm$^{-3}$) for AQS observations (grey), CMAQv5.0.2_Base simulation (blue) and CMAQv5.1_Base_NEIv1 simulation (red) for a) winter b) spring c) summer and d) fall.

[Figure]

**Figure 10.** Regional and seasonal stacked bar plots of PM$_{2.5}$ composition at the CSN sites (left), CMAQv5.0.2_Base simulation (middle) and CMAQv5.1_Base_NEIv1 simulation (right). In order from top to bottom are spring, summer, fall and winter seasons and left to right the Northeast, Great Lakes, Atlantic, South and West regions. The individual PM$_{2.5}$ components (in order from bottom to top) are SO$_4^{2-}$ (yellow), NO$_3^-$ (red), NH$_4^+$ (orange), EC (black), OC (light gray), Soil (brown), NaCl (green), NCOM (pink), other (white), blank adjustment (dark gray) and H2O/FRM adjustment (blue).

[Figure]

**Figure 11.** Difference in the monthly average hourly O$_3$ (ppbv) for winter (DJF; top left), spring (MAM; top right), summer (JJA; bottom left) and fall (SON; bottom right) between CMAQ v5.0.2_Base and v5.1_Base_NEIv1 (CMAQv5.1_Base_NEIv1 - CMAQv5.0.2_Base). Note that the scales  between each plot  may vary.

[Figure]

**Figure 12.** Seasonal average hourly O$_3$ (ppbv) mean bias at AQS sites for a) winter (DJF) b) spring (MAM) c) summer (JJA) and d) fall (SON) for the CMAQ v5.1_Base_NEIv1 simulation.

[Figure]

**Figure 13.** Difference in the absolute value of monthly average $O_3$ (ppbv) mean bias for a) winter (DJF) b) spring (MAM) c) summer (JJA) and d) fall (SON) between CMAQ v5.0.2_Base and v5.1_Base_NEIv1 (CMAQv5.1_Base_NEIv1 - CMAQv5.0.2_Base). Cool colors indicate a reduction in $O_3$ mean bias in v5.1 while warm color indicate an increase in $O_3$ mean bias v5.1.

[Figure]

**Figure 14.** Diurnal time series of seasonal O₃ ( ppbv)  for AQS observations (grey), CMAQv5.0.2_Base simulation (blue) and CMAQv5.1_Base_NEIv1 simulation (red) for a) winter b) spring c) summer and d) fall.

[Figure]

**Figure 15.** Diurnal time series of seasonal NO$_x$ (ppbv)  for AQS observations (grey), CMAQv5.0.2_Base simulation (blue) and CMAQv5.1_Base_NEIv1 simulation (red) for a) winter b) spring c) summer and d) fall.

[Figure]

**Figure 16.** Observed (black) and CMAQ simulated vertical profiles of a) $O_3$ b) $NO_2$ c) $NO_y$ d) alkyl nitrates (ANs) e) peroxy nitrates (PNs) and f) $HNO_3$ for the Edgewood site in Baltimore, MD on July 5, 2011. CMAQv502_Base simulation profiles are shown in green and CMAQv51_Base_NEIv1 simulation profiles are shown in red. Altitude (km) is given on the y-axis, while mixing ratio (ppbv) is given on the x-axis.

[Figure]

**Figure 17.** Difference in MDA8 O₃ daily ratios (Cut Scenario / Base) for CMAQv5.0.2 and v5.1 (v5.0.2 - v5.1) for a 50% cut in anthropogenic NOx NO$_x$ (top) and VOC (bottom) for January (left) and July (right) binned by the modeled MDA8 O₃ mixing ratio (ppbV ppbv). Values greater than one indicate v5.1 is more responsive than v5.0.2 to the emissions cut, while values less than one indicate v5.0.2 is more responsive. Given above the x-axis is the number of model grid cells in each bin.

[Figure]

**Figure 18.** Box plots of monthly average ratio values (Cut /Base) of PMIJ (total PM$_{2.5}$), ASO4IJ, ANO3IJ, ANH4IJ, AECIJ, ANCOMIJ, AUNSPECIJ, AOMIJ, APOAIJ, AORGAJ, AORGBJ, and AORGCJ for v5.0.2 (blue) and v5.1 (red) for a 50% cut in anthropogenic NO$_x$ (left), VOC (middle) and SO$_x$ (right) for January (top) and July (bottom).

**Table 1.** New/revised SOA species in the CMAQv5.1 AERO6 mechanism.

| Aerosol Species | Change since v5.0.2 | Applicable Mechanism | Description of Modification |
|---|---|---|---|
| AH3OP | added | all | Hydronium ion (predicted by ISOR-ROPIA for I+J modes); used for IEPOX uptake |
| APAH1,2 | added | cb05e51, saprc07tb, saprc07tc, saprc07tic, racm | Naphthalene aersol from $RO_2+NO$ reactions |
| APAH3 | added | cb05e51, saprc07tb, saprc07tc, saprc07tic, racm | Naphthalene aersol from $RO_2+HO_2$ reactions |
| AISO1,2 | updated | cb05e51, saprc07tb, saprc07tc*, racm | Aerosol from isoprene reactions $NO_3$ added to existing OH (all yields follow the OH pathway) |
| AISO3 | updated | cb05e51, saprc07tb, saprc07tc*, racm | Aerosol from reactive uptake of IEPOX on aqueous aerosol particles. Specifically intended to be the sum of 2-methyltetrols and IEPOX-derived organosulfates |
| AALK1,2 | added | cb05e51, saprc07tb, saprc07tc, saprc07tic, racm | Alkane aerosol |
| AALK | removed | all | deprecated alkane aerosol |

* AERO6i does not include SOA from isoprene+$NO_3$ in AISO1,2 (it is included in AISOPNNJ). AERO6i does not include IEPOX SOA in AISO3 (it is included in AITETJ, AIEOSJ, AIDIMJ, etc.). AISO3 is approximately zero in AERO6i.

**Table 2.** Description of the CMAQ model simulations utilized.

| CMAQ Simulation Name | CMAQ Version | WRF Version | NEI Version | Photolysis Scheme | Chemical Mechanism | Simulation Period (all 2011) |
|---|---|---|---|---|---|---|
| CMAQv5.0.2_Base | v5.0.2 | v3.4 | v1 | v5.0.2 | CB05TULC | Annual |
| CMAQv5.0.2_WRFv3.7 | v5.0.2 | v3.7 | v1 | v5.0.2 | CB05TUCL | January and July |
| CMAQv5.1_Base_NEIv1 | v5.1 | v3.7 | v1 | v5.1 | CB05e51 | Annual |
| CMAQv5.1_Base_NEIv2 | v5.1 | v3.7 | v2 | v5.1 | CB05e51 | Annual |
| CMAQv5.1_Retrophot | v5.1 | v3.7 | v2 | v5.0.2 | CB05e51 | January and July |
| CMAQv5.1_TUCL | v5.1 | v3.7 | v2 | v5.1 | CB05e51 | January and July |